# PROBABILISTIC DECOMPOSABLE EMBEDDINGS: UNCERTAINTY-AWARE COMPOSITION IN VISION-LANGUAGE MODELS

## ABSTRACT

Vision-Language Models (VLMs) organize concepts into shared embedding spaces, enabling compositional reasoning across modalities. Prior works demonstrated that composite concepts can be constructed by combining "ideal words" derived from attribute–object pairs. However, they rely solely on mean representations, neglecting the uncertainty inherent in these embeddings. In this work, we introduce Probabilistic Decomposable Embeddings (PDE), a framework that explicitly models ideal words as distribution. Instead of simply averaging attribute and object vectors, PDE formulates composition as a maximum a posteriori (MAP) estimation problem, producing composite embeddings biased toward concepts with lower variance. This probabilistic treatment yields partner-aware, precision-weighted composites with a simple count-based scale recovery. We first visualize PDE, showing that it reorients composite directions toward higher-precision axes while decoupling direction from scale. On compositional classification, PDE often matches or surpasses linear decomposable embeddings and geodesically decomposable embeddings in both modalities—improving harmonic mean and AUC. These results highlight *compositional pliability* as a useful inductive bias for uncertainty-aware composition in VLM embeddings.

## 1 INTRODUCTION

The concept of compositionality lies at the heart of human cognition. Complex meanings can be expressed as systematic combinations of simpler constituents such as attributes and objects. Vision-Language Models (VLMs) such as Contrastive Language-Image Pretraining (CLIP) (Radford et al., 2021) have shown that this principle also emerges in representation learning, where textual and visual signals are aligned within a shared embedding space. A central question is how such compositional structure should be explicitly modeled in order to construct embeddings of novel concepts from known ones.

Prior works have approached this question through the notion of an *ideal word*. In Linear Decomposable Embeddings (LDE) (Trager et al., 2023), the ideal word was defined as a context-adjusted representation of an attribute or object, obtained by subtracting a shared context vector from observed embeddings; compositions were then formed by linear addition. Geodesically Decomposable Embeddings (GDE) (Berasi et al., 2025b) extended this idea by accounting for the non-Euclidean geometry of $\ell_2$-normalized embeddings. In our interpretation, GDE can be viewed as defining a *geodesic ideal word*, computed through geometry-aware operations on the hypersphere via tangent-space addition and exponential mapping. Both perspectives emphasize that the building blocks of compositionality are ideal words, whose definition depends on the context and geometry of the embedding space.

However, an important limitation persists. Existing formulations treat ideal words as deterministic points, represented solely by their means. In reality, the underlying world states that give rise to observations are inherently stochastic. Attributes and objects denote *families of states* with intrinsic variability (Gardenfors, 2004; Rosch & Mervis, 1975; Murphy, 2004; Vilnis & McCallum, 2015; Kendall & Gal, 2017; Nagarajan & Grauman, 2018). When such variability is projected into a latent space, concepts are naturally expressed as *distributions*—often anisotropic—rather than single

vectors. Beyond central tendency, these *distributions* encode a concept's *compositional pliability* (or, dually, *compositional stiffness*): the degree to which a concept is expected to yield to, or dominate over, another during composition. Ignoring this probabilistic structure discards the information that governs which component should be emphasized when two concepts are combined.

Therefore, in this work, we introduce **Probabilistic Decomposable Embeddings (PDE)**, a framework that *explicitly* models ideal words as probability distributions. Rather than averaging mean vectors, PDE casts composition as a maximum a posteriori (MAP) estimation under Gaussian concept distributions (optionally with an isotropic prior), yielding a covariance-weighted fusion that *naturally* biases the composite toward lower-variance components while respecting anisotropy. This probabilistic treatment reflects the inherently stochastic, many-to-many mapping from world states to latent representations for each concepts—each concept induces a distribution over embeddings—and yields more faithful, reliable compositional structures.

**Contributions.** Our contributions are summarized as follows:

- **Conceptual reframing:** We recast ideal words from deterministic points to *probability distributions* and formalize *compositional pliability* (the inverse, *compositional stiffness*) as variance-informed behavior during composition.

- **Method:** We propose PDE, a MAP-based covariance-weighted composition rule that incorporates uncertainty and anisotropy of concept distributions, providing a principled alternative to mean-only addition.

- **Practice and evidence:** We present a practical estimation pipeline with count-adaptive regularization, and demonstrate consistent gains on CLIP-based compositional classification task, with qualitative ablation assessing the role of uncertainty modeling.

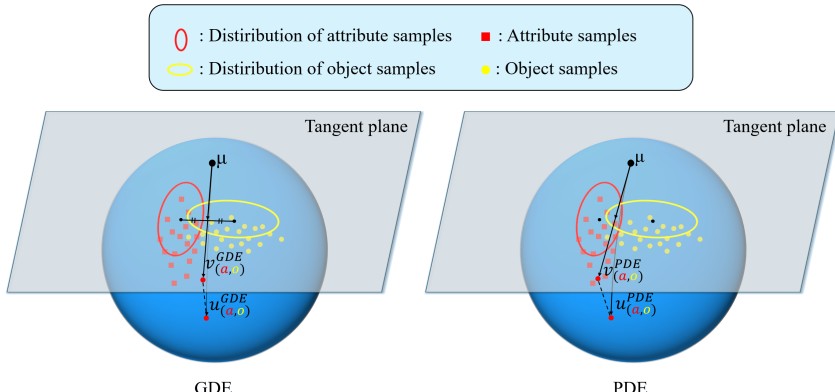

Figure 1: In the GDE method (left), the composite word is defined as the simple sum of the ideal words, whereas in our PDE method (right), it is defined as the overlapping direction of the distribution formed by the samples that generated the ideal words. Here, $\mu$ denotes the *intrinsic mean* on the sphere (the base point of the tangent plane); $\mathbf{v}(a, o)$ is the *composed embedding* in the tangent space; and $\mathbf{u}(a, o) = \mathrm{Exp}_\mu\big(\mathbf{v}(a, o)\big)$ is the denoised embedding mapped back onto the sphere.

## 2 PRELIMINARIES

We begin by reviewing CLIP (Sec. 2.1). We then reinterpret and consolidate two compositional schemes—Linear Decomposition of Embeddings (Sec. 2.2) and Geodesic Decomposition of Embeddings (Sec. 2.3)—placing them in unified notation and clarifying their geometric assumptions.

### 2.1 CONTRASTIVE LANGUAGE-IMAGE PRETRAINING (CLIP)

CLIP (Radford et al., 2021) consists of a pretrained image encoder $\phi_{\mathrm{im}} : \mathcal{X} \to \mathbb{R}^d$ and a text encoder $\phi_t : \mathcal{Y} \to \mathbb{R}^d$, which jointly embed images $x \in \mathcal{X}$ and texts $y \in \mathcal{Y}$ into a shared vision–language

space. Given an image embedding $\phi_{\text{im}}(x)$ and a text embedding $\phi_t(y)$, their normalized representations are defined as

$$u_x = \frac{\phi_{\text{im}}(x)}{\|\phi_{\text{im}}(x)\|}, \quad u_y = \frac{\phi_t(y)}{\|\phi_t(y)\|}. \tag{1}$$

The similarity between $x$ and $y$ is measured by the cosine similarity $\langle u_x, u_y \rangle$. The encoders are trained on large-scale image–text pairs by optimizing a contrastive objective that aligns matched pairs and separates mismatched ones.

## 2.2 Linear Decomposition of Embeddings (LDE)

The work of Trager et al. (2023) was the first to identify latent compositional structure in CLIP embeddings, showing that composite concepts can be approximated as linear combinations of a small set of *ideal words*. Building on this idea, LDE provides a simple linear factorization in the Euclidean space of embeddings. Let $Z = Z_1 \times \cdots \times Z_s$ index composite concepts, and let $\{u_z \in \mathbb{R}^d : z \in Z\}$ be their normalized CLIP text embeddings (e.g., $u_z = \phi_t(\texttt{prompt}(z))/\|\phi_t(\texttt{prompt}(z))\|$).

**Context.** The global context is defined as the mean over all composites, *denoted by $u_0$*:

$$u_0 = \frac{1}{|Z|} \sum_{z \in Z} u_z. \tag{2}$$

**Ideal words.** For each primitive value $z_i \in Z_i$, the representative mean over the slice $Z(z_i) = \{z \in Z : z_i \text{ appears in } z\}$ is *denoted by $\bar{u}_{z_i}$* and given by

$$\bar{u}_{z_i} = \frac{1}{|Z(z_i)|} \sum_{z \in Z(z_i)} u_z. \tag{3}$$

The ideal word is the context-removed vector, *denoted by $v_{z_i}$*:

$$v_{z_i} = \bar{u}_{z_i} - u_0, \qquad \sum_{z_i \in Z_i} v_{z_i} = 0 \ (\forall i). \tag{4}$$

**Composite.** A composite embedding is obtained by adding the ideal words of its factors together with the context, *denoted by $u_z^{\text{LDE}}$*:

$$u_z^{\text{LDE}} = u_0 + \sum_{i=1}^{s} v_{z_i}, \qquad z = (z_1, \ldots, z_s). \tag{5}$$

**Attribute/object case.** Let $Z = \mathcal{A} \times \mathcal{O}$ and $\mathcal{Z} \subseteq \mathcal{A} \times \mathcal{O}$ be the observed pairs with embeddings $u_{(a,o)}$. Then:

$$\textit{Context:} \quad u_0 = \frac{1}{|\mathcal{Z}|} \sum_{(a,o) \in \mathcal{Z}} u_{(a,o)}. \tag{6}$$

$$\textit{Ideal word:} \quad v_a = \bar{u}_a - u_0 \quad \text{where} \quad \bar{u}_a = \frac{1}{|\{\, o : (a,o) \in \mathcal{Z}\}|} \sum_{\substack{o \\ (a,o) \in \mathcal{Z}}} u_{(a,o)}, \tag{7}$$

$$v_o = \bar{u}_o - u_0 \quad \text{where} \quad \bar{u}_o = \frac{1}{|\{\, a : (a,o) \in \mathcal{Z}\}|} \sum_{\substack{a \\ (a,o) \in \mathcal{Z}}} u_{(a,o)}. \tag{8}$$

$$\textit{Composite:} \quad u_{(a,o)}^{\text{LDE}} = u_0 + v_a + v_o. \tag{9}$$

## 2.3 Geodesic Decomposition of Embeddings (GDE)

Like LDE, GDE (Berasi et al., 2025b) builds a context, ideal words, and a composite vector—now on the manifold where normalized CLIP embeddings live. Let $M \subset \mathbb{R}^d$ be the unit hypersphere with geodesic distance $d_M$, and let $\{u_z \in M : z \in Z\}$ be normalized embeddings for $Z = Z_1 \times \cdots \times Z_s$.

**Context.** The context is the intrinsic mean on the manifold, *denoted by* $\mu$:

$$\mu = \arg\min_{u \in M} \frac{1}{|Z|} \sum_{z \in Z} d_M(u, u_z)^2. \tag{10}$$

**Geodesic ideal words.** For each primitive value $z_i \in Z_i$, the geodesic ideal word is defined as the average in the tangent space at $\mu$ over the slice $Z(z_i)$, *denoted by* $v_{z_i}$:

$$v_{z_i} = \frac{1}{|Z(z_i)|} \sum_{z \in Z(z_i)} \mathrm{Log}_\mu(u_z), \qquad \sum_{z_i \in Z_i} v_{z_i} = 0 \; (\forall i). \tag{11}$$

**Composite.** A composite embedding is formed by summing the ideal words in $T_\mu M$ and mapping back to the manifold via the exponential map, *denoted by* $u_z^{\mathrm{GDE}}$:

$$u_z^{\mathrm{GDE}} = \mathrm{Exp}_\mu\Big(v_{z_1} + \cdots + v_{z_s}\Big), \qquad z = (z_1, \ldots, z_s). \tag{12}$$

**Attribute/object case.** Let $Z = \mathcal{A} \times \mathcal{O}$ with observed pairs $\mathcal{Z}$ and pair embeddings $u_{(a,o)} \in M$, and let $\mu$ be the intrinsic mean of $\{u_{(a,o)}\}$. Then:

$$\textit{Context:} \quad \mu = \arg\min_{u \in M} \frac{1}{|\mathcal{Z}|} \sum_{(a,o) \in \mathcal{Z}} d_M(u, u_{(a,o)})^2. \tag{13}$$

$$\textit{Ideal word:} \quad v_a = \frac{1}{|\{o : (a,o) \in \mathcal{Z}\}|} \sum_{\substack{o \\ (a,o) \in \mathcal{Z}}} \mathrm{Log}_\mu(u_{(a,o)}), \tag{14}$$

$$v_o = \frac{1}{|\{a : (a,o) \in \mathcal{Z}\}|} \sum_{\substack{a \\ (a,o) \in \mathcal{Z}}} \mathrm{Log}_\mu(u_{(a,o)}). \tag{15}$$

$$\textit{Composite:} \quad u_{(a,o)}^{\mathrm{GDE}} = \mathrm{Exp}_\mu(v_a + v_o). \tag{16}$$

**Constructing ideal words from images.** Ideal words can also be constructed from *images* rather than text. For a composite label $z$ (e.g., "red dog"), suppose we have multiple images indexed by a finite set $E_z$ with normalized image embeddings $u_{(z,e)} = \phi_{\mathrm{im}}(x_{(z,e)})/\|\phi_{\mathrm{im}}(x_{(z,e)})\|$. We first compute a *denoised* tangent representative of the composite $v_z$, via a weighted log-map average:

$$p(z, e) = \frac{\exp\big(u_{(z,e)}^\top u_{y(z)}/t\big)}{\sum_{e' \in E_z} \exp\big(u_{(z,e')}^\top u_{y(z)}/t\big)}, \qquad v_z = \sum_{e \in E_z} p(z, e) \, \mathrm{Log}_\mu\big(u_{(z,e)}\big), \tag{17}$$

where $u_{y(z)} = \phi_t(y(z))/\|\phi_t(y(z))\|$ is the normalized text embedding of a prompt for $z$ and $t$ is a temperature. The geodesic ideal words for primitives are then obtained by averaging these denoised representatives over slices,

$$v_{z_i} = \frac{1}{|Z(z_i)|} \sum_{z \in Z(z_i)} v_z, \qquad \sum_{z_i \in Z_i} v_{z_i} = 0 \; (\forall i), \tag{18}$$

and the composite remains *denoted by* $u_z^{\mathrm{GDE}} = \mathrm{Exp}_\mu\big(\sum_{i=1}^s v_{z_i}\big)$.

## 3 METHOD

### 3.1 COMPOSITIONAL PLIABILITY

Existing decompositions (LDE/GDE; Secs. 2.2–2.3) treat each primitive's *ideal word* as partner-agnostic. When two primitives are combined, their contributions are fixed regardless of which partner they meet. In practice, however, contributions should adapt to *compositional pliability*—the tendency of a primitive to yield or dominate depending on its uncertainty structure (including potential anisotropy) in the embedding space. To model this partner-aware behavior, we *redefine the ideal word as a distribution* whose covariance encodes compositional pliability (and whose inverse encodes stiffness), enabling composites to naturally weight primitives by their uncertainty.

**Setup (attribute–object; notational simplification).**   We focus on the attribute–object (A–O) case for clarity and generalize to $s$-way composition later. Let $\mathcal{A}$ and $\mathcal{O}$ be the sets of attributes and objects, and let $\mathcal{Z} \subseteq \mathcal{A} \times \mathcal{O}$ be the observed pairs with normalized embeddings $u_{(a,o)}$ (text or image). Denote the set of composing primitives by $\mathcal{P} = \{a, o\}$ so that $s = |\mathcal{P}| = 2$.

**Redefinition of *Ideal Word*.**   We replace point ideal words by *distributions* in the context-centered space (Euclidean or $T_\mu M$). For each primitive $p \in \{a, o\}$,

$$v_p \sim \mathcal{N}(m_p, \Sigma_p), \qquad P_p \equiv \Sigma_p^{-1} \succ 0, \tag{19}$$

with $\Sigma_p$ estimated using shrinkage and a small $\varepsilon I$ for conditioning.

**Definition of Compositional Pliability.**   For a unit direction $w \in \mathbb{R}^d$ with $\|w\| = 1$, the compositional pliability of a primitive $p$ is

$$\pi_p(w) = w^\top \Sigma_p w, \tag{20}$$

and we define the corresponding directional stiffness by $\kappa_p(w) = 1/\pi_p(w)$, so that $\pi_p(w)\,\kappa_p(w) = 1$. Given a unit direction $w$, the *directional pliability* is the quadratic form $\pi_p(w) = w^\top \Sigma_p w$ and the corresponding *directional stiffness* is $\kappa_p(w) = 1/\pi_p(w)$. (Equivalently, when parameterized by the stiffness tensor $P_p := \Sigma_p^{-1}$, one writes $\kappa_p(w) = w^\top P_p w$, in which case $\pi_p(w)\,\kappa_p(w) \geq 1$ with equality on eigen-directions.)

### 3.2 IDEAL COMPOSITIONAL VECTOR

We introduce the *ideal compositional vector*, the composite that considers compositional pliability of the primitives. We mathematically define this vector as the MAP solution obtained by fusing the distributions of composing primitives, followed by a scale factor $s$ (the number of primitives). This precision-weighted fusion *automatically* reflects each concept's *compositional pliability*.

**Definition of *Ideal Compositional Vector* (Composite).**   Given $(a, o)$ with ideal-word distributions $v_a \sim \mathcal{N}(m_a, \Sigma_a)$ and $v_o \sim \mathcal{N}(m_o, \Sigma_o)$, first compute the MAP fusion

$$v_{(a,o)}^\star = \arg\max_v\ \mathcal{N}(v; m_a, \Sigma_a)\,\mathcal{N}(v; m_o, \Sigma_o)\,\mathcal{N}(v; 0, \lambda^{-1}I) = (P_a + P_o + \lambda I)^{-1}\big(P_a m_a + P_o m_o\big). \tag{21}$$

We then define the *ideal compositional vector* by applying the primitive-count scale:

$$v_{(a,o)}^{\mathrm{comp}} = s\, v_{(a,o)}^\star, \qquad s = \#\ \text{of composing primitives (here } s = 2). \tag{22}$$

(General $s$-way composition $p_1, \ldots, p_s$ is obtained by summing the corresponding precisions and means in the same form; scale restoration by $s$ is discussed in Sec. 3.3.)

**Why does the MAP solution encode compositional pliability?**   For 1D intuition, let $v_a \sim \mathcal{N}(m_a, \sigma_a^2)$ and $v_o \sim \mathcal{N}(m_o, \sigma_o^2)$ (set $\lambda = 0$). The MAP of the product is the inverse-variance average as

$$v^\star = \frac{\sigma_a^{-2}}{\sigma_a^{-2} + \sigma_o^{-2}}\, m_a + \frac{\sigma_o^{-2}}{\sigma_a^{-2} + \sigma_o^{-2}}\, m_o, \tag{23}$$

which lies closer to the mean with *smaller variance* (higher precision). In multiple dimensions, projecting onto any unit $w$ yields the same inverse-variance principle with directional precisions $w^\top P_p w$, so the MAP solution bends toward the partner that is stiffer along $w$, and the post-factor $s$ does not change this behavior (it only restores scale). (See Appx. B.2 for more details.)

## 3.3 PDE: PROBABILISTICALLY DECOMPOSABLE EMBEDDING

*Goal.* PDE instantiates the ideal compositional vector under a *carrier* × *scale* decomposition: (i) determine a partner-aware *carrier* by precision-weighted MAP fusion, and (ii) set the *scale* as the primitive count $s$. Both the carrier and the scaling are defined in this subsection.

**Equal-weight baseline (LDE/GDE) as *carrier* × *scale*.** Both LDE and GDE form an equal-weight *carrier* $c^{\mathrm{eq}}$ and restore magnitude by the primitive count $s$ (i.e., $s = |\mathcal{P}|$, where $\mathcal{P}$ is the set of composing primitives; e.g., $\mathcal{P} = \{a, o\}$ in the A–O case).

$$\text{Euclidean (LDE)}: \quad c_{(a,o)}^{\mathrm{eq}} = \tfrac{1}{s} \sum_{p \in \{a,o\}} m_p \in \mathbb{R}^d. \tag{24}$$

$$\text{Geometric (GDE)}: \quad c_{(a,o)}^{\mathrm{eq}} = \tfrac{1}{s} \sum_{p \in \{a,o\}} \log_\mu(m_p) \in T_\mu M. \tag{25}$$

$$\textit{Equal-weight composite carrier:} \quad v_{(a,o)}^{\mathrm{EW}} = s\, c_{(a,o)}^{\mathrm{eq}}. \tag{26}$$

$$\textit{Composites:} \quad u_{(a,o)}^{\mathrm{LDE}} = u_0 + v_{(a,o)}^{\mathrm{EW}}, \quad u_{(a,o)}^{\mathrm{GDE}} = \exp_\mu\big(v_{(a,o)}^{\mathrm{EW}}\big). \tag{27}$$

**Precision-weighted carrier via MAP; scale recovery.** Using the shrinkage precisions $P_a, P_o$, define the MAP *carrier*

$$c_{(a,o)}^{\mathrm{map}} := \big(P_a + P_o + \lambda I\big)^{-1}\big(P_a m_a + P_o m_o\big), \tag{28}$$

computed in the chosen linearization (Euclidean or $T_\mu M$). The ideal compositional vector is then simply

$$v_{(a,o)}^{\mathrm{PDE}} = s\, c_{(a,o)}^{\mathrm{map}}. \tag{29}$$

This formulation shows that the LDE/GDE carrier can be viewed as a special case of the PDE carrier. When all primitive Gaussians share a common isotropic covariance $\Sigma_p = \sigma^2 I$ and no shrinkage is used ($\lambda = 0$), the MAP carrier $c_{\mathcal{P}}^{\mathrm{map}}$ reduces to the simple mean of the primitive means, recovering the equal-weight carrier used in LDE/GDE. See Appx. C.5 for a formal derivation and Appx. B.2 for quantitative evidence on scale recovery.

**Reliability-blended PDE (carrier blending; $\eta$ as evidence gate).** We interpolate between the equal-weight carrier and the precision-weighted MAP carrier using a *reliability* gate $\eta \in [0, 1]$:

$$u_{(a,o)} = (1 - \eta_{(a,o)})\, c_{(a,o)}^{\mathrm{eq}} + \eta_{(a,o)}\, c_{(a,o)}^{\mathrm{map}}, \quad v_{(a,o)}^{\mathrm{PDE}} = s\, u_{(a,o)}. \tag{30}$$

*Philosophy.* $\eta$ quantifies how much we *trust* the covariance-driven direction (MAP) over the equal-weight carrier. Therefore, it should rise when covariance estimates are well-supported.

*One automatic choice.* As a practical default, we set $\eta$ from the *evidence* in the *sample covariance* $\widehat{\Sigma}_p$ for each primitive $p$. Let

$$\delta_p = \frac{\left(1 - \frac{2}{D}\right) t_2 + t_1^2}{\left(n_p + 1 - \frac{2}{D}\right)\left(t_2 - \frac{t_1^2}{D}\right)}, \quad t_1 = \mathrm{tr}(\widehat{\Sigma}_p),\ t_2 = \mathrm{tr}(\widehat{\Sigma}_p^2), \tag{31}$$

be the OAS coefficient for embedding dimension $D$ and effective sample size $n_p$ (Chen et al., 2010). Interpreting $1 - \delta_p$ as covariance *evidence*, we combine attribute/object evidence geometrically:

$$\eta_{(a,o)} = \sqrt{\big(1 - \delta_a\big)\big(1 - \delta_o\big)} \in [0, 1]. \tag{32}$$

We clip $\delta_p$ to $[0, 1]$ before forming $\eta$, and use pair-level $n_p$ when covariances are pair-conditioned (otherwise raw counts).

**General $s$-way composition.** For $p_1, \ldots, p_s$, set $P_\Sigma = \sum_{i=1}^{s} P_{p_i}$ and $b = \sum_{i=1}^{s} P_{p_i} m_{p_i}$. Then $v^\star = (P_\Sigma + \lambda I)^{-1} b$. Let $c^{\text{eq}} = \frac{1}{s} \sum_{i=1}^{s} m_{p_i}$ (or its tangent analogue), and define $\eta = \sqrt{\prod_{i=1}^{s}(1 - \delta_{p_i})}$. Blend carriers as $u = (1 - \eta) c^{\text{eq}} + \eta v^\star$, and set $v^{\text{PDE}} = s\, u$.

## 4 EXPERIMENTS

We begin by *visualizing* PDE to show how partner-aware precision reorients composite directions while preserving scale. Unless otherwise specified, we instantiate PDE with the *GDE backbone* ("Probabilistic GDE"). For completeness, we also implement the Euclidean instantiation with an LDE backbone ("Probabilsitic LDE"), and its results are reported in Appx. B.4. We then evaluate PDE on *compositional classification* in both closed- and open-world settings across two benchmarks (UT-Zappos (Yu & Grauman, 2014), MIT-States (Isola et al., 2015)), comparing against CLIP, LDE, and GDE. We use pretrained SIGLIP (Zhai et al., 2023) as default CLIP model. Experiments regarding other models like (RN50 (Radford et al., 2021), ViT/L-14 (Radford et al., 2021), SIGLIP2 (Tschannen et al., 2025)) are in Appx. B.1.

### 4.1 VISUALIZING PROBABILISTIC DECOMPOSABLE EMBEDDINGS

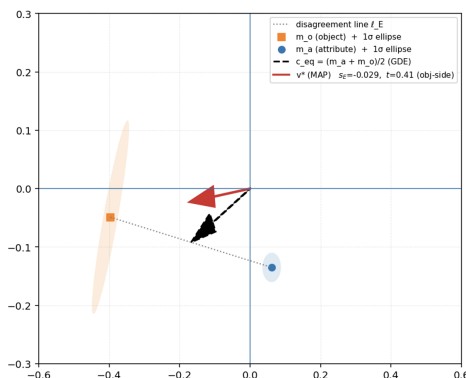

Figure 2: **Visualising PDE with tilt metrics.** Dataset: *MIT-States*; ideal-word modality: *image*. We fix the object (*apple*) and attribute (*mossy*). We draw the equal-weight carrier $c^{\text{eq}}$ (black) and the precision-weighted MAP carrier $c^{\text{map}}$ (red) as arrows from the origin in $T_\mu$ with $s{=}2$ (two primitives). On-figure we report only the *tilt certificate* $s_E$ and the projection coefficient $t$ (equation 33–equation 34): $s_E > 0$ ($t > \frac{1}{2}$) indicates the attribute side along the disagreement line, $s_E < 0$ ($t < \frac{1}{2}$) indicates the object side.

**PCA-based visualization.** All geometry is drawn in the context-centered tangent space $T_\mu M$, where the context vector is the origin. For each fixed object $o$, we collect the tangent vectors $\{m_o, m_a, c^{\text{eq}}, c^{\text{map}}\}$ over all attributes paired with $o$ and build a *two-dimensional, uncentered PCA* basis $W \in \mathbb{R}^{d \times 2}$. The first principal axis is oriented to align with the average disagreement direction. We then project by $x \mapsto xW$, draw the disagreement line $\ell_E = \{m_o + t(m_a - m_o) : t \in \mathbb{R}\}$, and plot both carriers from the origin: the equal-weight carrier $c^{\text{eq}} := m_a + m_o$ (dashed) and the precision-weighted MAP carrier $c^{\text{map}}$ (solid). Arrow lengths are made comparable via the primitive-count rule with $s{=}2$. To convey uncertainty, we render $1\sigma$ covariance ellipses at $m_o$ and $m_a$ using $\Sigma_{\text{2D}} = W^\top \Sigma W$. All diagnostics are computed *only* along $\ell_E$; the legend reports the tilt certificate $s_E$ and the projection coefficient $t$.

**Tilt metrics ($s_E$ and $t$).** Let the *disagreement vector* be $d := m_a - m_o$ and the *disagreement line* $\ell_E = \{m_o + t\,d : t \in \mathbb{R}\}$. We define the scalar

$$s_E := d^\top(v^\star - c_{\text{eq}}), \quad \text{where} \quad c_{\text{eq}} := \tfrac{1}{2}(m_a + m_o), \;\; d := m_a - m_o \tag{33}$$

and the orthogonal projection coefficient of $v^\star$ onto $\ell_E$:

$$t := \frac{d^\top(v^\star - m_o)}{\|d\|^2}. \tag{34}$$

They satisfy the exact equivalence

$$s_E > 0 \iff t > \tfrac{1}{2} \quad \text{(attribute-side)},$$
$$s_E < 0 \iff t < \tfrac{1}{2} \quad \text{(object-side)}, \tag{35}$$
$$s_E = 0 \iff t = \tfrac{1}{2} \quad \text{(midpoint)}.$$

Thus, $\text{sign}(s_E)$ indicates *dominance label* of attribute, and we additionally display $t$ to convey the *amount* of displacement from the midpoint along $\ell_E$. See Appx. C. for detailed explanation.

*Interpretation.* Because PDE composes by a precision-weighted MAP rather than a mean, the composite $v^\star$ is *tilted* toward the primitive with higher directional precision along $\ell_E$. The sign of $s_E$ *certifies* the dominance side (attribute vs. object), while $t$ *quantifies* displacement from the midpoint.

### 4.2 COMPOSITIONAL CLASSIFICATION

Table 1: Compositional classification on UT-Zappos and MIT-States using **SigLIP (ViT-SO400M/14, patch of 224**. The highest values within each modality are in **bold**. The second-highest values are underlined.

| Dataset | Method | Closed-World | | | | | | Open-World | | | | | |
|---|---|---|---|---|---|---|---|---|---|---|---|---|---|
| | | Attr | Obj | Seen | Unseen | HM | AUC | A | O | S | U | H | AUC |
| | CLIP | 52.5 | 74.5 | 44.9 | 68.1 | 39.2 | 24.6 | 47.2 | 70.1 | 44.9 | 49.9 | 32.5 | 16.5 |
| UT-Zappos | LDE (*text*) | 46.8 | 73.6 | 41.3 | 70.9 | 33.2 | 20.5 | 45.1 | 72.1 | 41.3 | 55.2 | 31.1 | 16.6 |
| | GDE (*text*) | 47.8 | **74.9** | 43.7 | **71.4** | 34.5 | 21.9 | 46.3 | **73.0** | 43.7 | **56.3** | 32.7 | 17.8 |
| | PDE (*text*) | **54.8** | 72.7 | **46.9** | 68.1 | **39.5** | **24.9** | **52.0** | 69.4 | **46.9** | 54.0 | **35.2** | **19.0** |
| | LDE (*image*) | 21.3 | 50.2 | 7.0 | 42.0 | 8.2 | 1.6 | 18.8 | 44.6 | 7.0 | 27.1 | 7.3 | 1.0 |
| | GDE (*image*) | **48.4** | 72.4 | 43.1 | 68.5 | 42.1 | 25.2 | **41.0** | 70.4 | 41.3 | **49.6** | **31.2** | 15.3 |
| | PDE (*image*) | 47.6 | **74.7** | **45.4** | **69.0** | **42.8** | **26.7** | 39.0 | **71.7** | **45.4** | 48.5 | 31.0 | **16.2** |
| MIT-States | CLIP | 45.8 | 61.3 | 43.7 | 57.8 | 39.3 | 22.0 | 24.3 | 54.5 | 43.7 | 16.2 | 16.6 | 5.4 |
| | LDE (*text*) | 38.9 | 58.5 | 34.2 | 51.5 | 30.0 | 14.1 | **29.3** | 56.9 | 34.2 | **21.7** | 18.2 | 5.6 |
| | GDE (*text*) | 39.5 | 58.8 | 34.9 | 52.1 | 30.6 | 14.8 | **29.3** | 56.8 | 34.9 | 21.6 | 18.4 | 5.8 |
| | PDE (*text*) | **46.5** | **61.7** | **43.6** | **58.9** | **40.0** | **22.5** | 28.0 | 56.4 | **43.6** | 20.8 | **19.8** | **6.9** |
| | LDE (*image*) | 19.0 | 34.4 | 18.8 | 27.5 | 14.8 | 3.6 | 13.8 | 37.1 | 18.8 | 6.9 | 6.0 | 0.7 |
| | GDE (*image*) | 32.1 | **50.1** | 36.2 | **40.5** | 27.0 | 11.7 | 21.9 | **49.1** | 35.9 | **11.8** | **12.5** | 3.0 |
| | PDE (*image*) | **33.5** | 49.5 | **46.1** | 39.9 | **30.8** | **14.6** | **22.3** | 48.1 | **46.1** | 10.5 | 11.9 | **3.3** |

**Protocol.** We follow the generalized zero-shot setup of prior work (e.g., GDE) and evaluate *compositional zero-shot classification*: given an image, the model predict a *composite text label* formed by an attribute–object pair. We consider two regimes: (i) **Closed-world**, where the candidate label set is restricted to the attribute–object pairs that appear as test pairs in the dataset; (ii) **Open-world**, where all valid attribute–object combinations in the dataset are candidates. For each modality (text/image), we build composite prototypes in a CLIP-based embedding space with four methods: *CLIP* (naïve pairing), *LDE* (equal-weight Euclidean), *GDE* (equal-weight geodesic), and *PDE* (precision-weighted MAP with evidence blending). Scoring uses cosine similarity in the appropriate linearization (tangent space for GDE/PDE). We report attribute accuracy (Attr), object accuracy (Obj), seen/unseen pair accuracies (Seen/Unseen), their best harmonic mean (HM), and the area under the seen–unseen curve (AUC), following the same calibration sweep protocol as GDE.

**Modalities and data usage.** The only difference between modalities is how the *ideal words* (primitive prototypes) are estimated: in the **text** modality, primitives come from CLIP text embeddings of attribute/object prompts; in the **image** modality, primitives are estimated from training images (pair-level embeddings) and then decomposed/combined into attribute/object and composites. Unlike the GDE image setting—which leverages a validation split to tune hyperparameters—we *do not use validation images*; all statistics (means, covariances/precisions) is used at test time.

**Hyperparameters.** Unless otherwise noted, PDE uses a fixed configuration with ridge $\lambda = 0.15$ for covariance/precision stabilization, while the evidence coefficient $\eta$ is set by an OAS-style rule

(Sec. 3.3). We keep $\lambda$ fixed because performance was empirically insensitive to moderate changes. Additional hyperparamters and analyses appear in Appx. B.3.

**Results.** Table 1 reports results on UT-Zappos, MIT-States. PDE attains the best or tied-best HM/AUC across modalities and regimes. In image modality, PDE raises HM over GDE in both closed-world and open-world, with consistent AUC gains; text shows similar trends. PDE consistently improves HM and AUC over LDE/GDE in closed-world for both modalities and remains competitive in open-world. We also report additional experiment on relational compositional classification in Appx. B.9.

**Findings.** Across benchmarks, PDE's *partner-aware* direction and *count-based* scale help in closed-world and frequently in open-world as well, especially when primitive reliabilities differ.

## 5 RELATED WORK

**Compositionality.** Compositionality refers to the principle that the meaning of a complex concept can be built by combining the meanings of simpler parts, enabling generalization to unseen combinations. This view—central in perception and cognition—motivates building representations that can be decomposed and recomposed systematically (Lake & Baroni, 2018; Keysers et al., 2020; Lake et al., 2017). Such structure is especially useful for vision problems, where modular representations of primitives (e.g., attributes and objects) facilitate recombination and transfer to novel compositions (Naeem et al., 2021; Nagarajan & Grauman, 2018; Karthik et al., 2022).

**Compositionality in Vision–Language Models.** Modern VLMs such as CLIP learn a shared image–text embedding space via contrastive training, without explicitly enforcing compositional structure (Radford et al., 2021). Nevertheless, emergent compositional behavior has been documented across modalities. On the *text* side, LDE approximate composite prompts by sums of a small set of "ideal words," enabling linear, interpretable manipulations (Trager et al., 2023). On the *image* side, GDE argue that spherical (Riemannian) geometry matters: visual composites can be modeled via geodesic operations—computing intrinsic means, adding in the tangent space, and mapping back by the exponential map (Berasi et al., 2025b). Taken together, these findings indicate that VLMs exhibit latent compositional structure and that non-Euclidean (geodesic) composition can outperform linear addition in text and image spaces (Trager et al., 2023; Berasi et al., 2025b). This structure is typically evaluated with *compositional classification*, which in the broader literature is studied as compositional zero-shot learning (CZSL): given *seen* attribute–object primitives, the goal is to recognize *unseen* compositions on benchmarks such as MIT-States and UT-Zappos (Isola et al., 2015; Yu & Grauman, 2014).

Within this CZSL setting, a rich line of work builds task-specific architectures on top of VLMs, for example by constructing language-informed class distributions or progressive language observations over primitives (Bao et al., 2024; Li et al., 2023), designing prompt-based or multi-path heads with separate attribute/object/composition prompts (Huang et al., 2024; Zhou et al., 2022), learning graph or prototype representations over state–object–composition nodes (Mancini et al., 2024; Qu et al., 2025), or disentangling visual attribute and object factors (Saini et al., 2022). These approaches typically modify prompts, introduce new classifiers, or fine-tune the backbone for CZSL. In contrast, PDE does not introduce any additional trainable modules: it is a post-hoc, probabilistic refinement of LDE/GDE that operates entirely inside the fixed decomposed VLM latent space, and can in principle be plugged into many of these CZSL pipelines as an uncertainty-aware composition rule.

**Probabilistic Interpretation and Precision-Weighted Composition.** A complementary line of work represents lexical or multimodal meaning as *probability distributions* rather than point vectors, modeling uncertainty and asymmetry (e.g., Gaussian word embeddings) (Vilnis & McCallum, 2015; Athiwaratkun et al., 2018) and formalizing composition probabilistically (Emerson & Copestake, 2016). In vision–language, *probabilistic cross-modal embeddings* embed images/texts as distributions and compare them via probabilistic similarities, leveraging uncertainty for one-to-many correspondences (Chun et al., 2021; Chun, 2024). Closest in spirit to our approach is the work of Neculai et al. (2022), who encode each image/text query as a Gaussian and compose multiple queries via a product-of-Gaussians "composer" in a task-specific embedding space for multimodal

retrieval. In contrast, PDE is a probabilistic refinement of LDE/GDE defined on top of a *decomposed* CLIP latent space: we first obtain disentangled attribute/object directions via ideal words and then perform precision-weighted fusion of Gaussian primitives in this decomposed space, yielding an interpretable carrier–scale composite that remains comparable to LDE/GDE. Moreover, while Neculai et al. (2022) focus on multi-query retrieval, we study general attribute–object composition (and its extensions to relational composition) in closed- and open-world *compositional classification* benchmarks.

## 6 CONCLUSION

We introduced *Probabilistic Decomposable Embeddings (PDE)*, which redefines ideal words as distributions whose covariances encode *compositional pliability*. PDE composes primitives via a precision-weighted MAP *carrier* (optionally blended by an evidence gate $\eta$) and a simple count-based *scale*, decoupling direction from magnitude in either Euclidean space or $T_\mu M$.

Across UT-Zappos and MIT-States, PDE matches or surpasses LDE/GDE on HM and AUC, and yields tighter, better-oriented composites in visualization. While PDE assumes shrinkage-stabilized Gaussian structure, future work includes richer priors, low-rank covariance modeling, multi-way grammars, task-adaptive $\eta$, and retrieval/generation evaluations. In sum, uncertainty is a useful inductive bias for compositional reasoning in VLM embeddings.

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

# Appendix

## A  IMPLEMENTATION DETAILS

Our implementation builds on the official code release of Berasi et al. (2025a).

### A.1  ESTIMATION OF SAMPLE COVARIANCE.

Let $p \in \{a, o\}$ index attribute and object. In the context-centered space (Euclidean or $T_\mu M$), estimate each primitive's mean $m_p$ and sample covariance $C_p$. To stabilize $C_p$ in the small-$n$ regime, we use a shrinkage estimator

$$\widehat{\Sigma}_p = (1 - \alpha) C_p + \alpha \tau_p^2 I + \varepsilon I, \qquad \tau_p^2 = \tfrac{1}{D} \operatorname{tr}(C_p), \ \ \alpha = 0.15,$$

and define the precision $P_p \equiv \widehat{\Sigma}_p^{-1}$. The small ridge $\varepsilon I$ improves conditioning; $\alpha$ controls shrinkage toward the spherical target (Ledoit & Wolf, 2004; Chen et al., 2010).

### A.2  COMPUTATIONAL COMPLEXITY AND RUNTIME COMPARISON

From a complexity viewpoint, PDE introduces two sources of additional cost compared to GDE, assuming that CLIP embeddings have already been extracted.

**Per-concept covariance and precision estimation.**  For each attribute or object primitive $p$, we estimate a covariance matrix in the decomposed CLIP space and compute its inverse (precision). Let $d$ be the CLIP embedding dimension (fixed across all experiments), and let $N_a, N_o$ denote the numbers of attributes and objects, respectively. Forming one covariance matrix costs $O(d^2)$ and inverting it costs $O(d^3)$, so the total offline complexity is

$$O\big((N_a + N_o)d^3\big).$$

Since $d$ is fixed and the number of concepts is in the low hundreds, this step is inexpensive and grows only linearly with the number of attributes/objects. The resulting precision matrices are cached and reused throughout training and evaluation.

**Per-composition MAP carrier.**  Given an attribute–object pair $(a, o)$, the PDE carrier is the MAP estimate under the corresponding Gaussians (plus a weak isotropic prior), which amounts to solving

$$(P_a + P_o + \lambda I)z = P_a\mu_a + P_o\mu_o, \tag{36}$$

where $P_a, P_o$ are the precomputed precision matrices and $\mu_a, \mu_o$ are the ideal means. In practice, we instantiate this MAP update once per composition class: we construct $A_{a,o} = P_a + P_o + \lambda I$ and solve $A_{a,o}z_{a,o} = P_a\mu_a + P_o\mu_o$ using a standard linear solver, which is $O(d^3)$ per prediction class. We then cache the resulting MAP carrier $z_{a,o}$ and use it as a fixed prototype for that composition. During evaluation (both in the closed- and open-world settings), all images associated with $(a, o)$ are scored simply by computing similarities (e.g., cosine similarity) between their CLIP embeddings and $z_{a,o}$. Consequently, the one-off $O(d^3)$ cost is paid only once per composition, and the per-image overhead of PDE scoring is dominated by matrix–vector and vector–vector products, which are small compared to the inverse procedure.

**Wall-clock runtime.**  To make these trade-offs concrete, we measured wall-clock time on a single GPU (with CLIP features precomputed) for fitting and evaluating GDE versus PDE on all UT-Zappos and MIT-States splits. Table 2 summarizes the results.

| Dataset / modality | Method | Closed (s) | Open (s) |
|---|---|---|---|
| UT-Zappos / text | GDE | 0.71 | 0.74 |
| | PDE | 3.81 | 4.59 |
| UT-Zappos / image | GDE | 238.79 | 248.37 |
| | PDE | 7.12 | 7.93 |
| MIT-States / text | GDE | 6.05 | 58.39 |
| | PDE | 49.82 | 370.96 |
| MIT-States / image | GDE | 356.06 | 1682.55 |
| | PDE | 47.10 | 371.16 |

Table 2: Wall-clock time (seconds) to fit and evaluate GDE vs. PDE on UT-Zappos and MIT-States, using precomputed CLIP embeddings. PDE is somewhat slower than GDE in the text-based configuration due to the extra covariance and MAP computations, but the absolute overhead remains modest. In the image-based configuration, PDE is substantially faster overall because it avoids the validation-time grid search over shrinkage/regularization parameters required by GDE.

# B ADDITIONAL RESULTS

We show additional results for this paper. We add cgqa (Naeem et al., 2021) dataset for additional results.

## B.1 MODEL BIAS EXPERIMENT.

Table 3: Ablation on backbone architecture in compositional classification with closed-world scenario. The highest values within each modality are in **bold**; The second-highest values are underlined.

| Dataset | Method | CLIP, RN50 | | | | | | CLIP, ViT-L/14 | | | | | | SigLIP, ViT-SO400M/14 | | | | | |
|---|---|---|---|---|---|---|---|---|---|---|---|---|---|---|---|---|---|---|---|
| | | ATTR | OBJ | SEEN | UNSEEN | HM | AUC | ATTR | OBJ | SEEN | UNSEEN | HM | AUC | ATTR | OBJ | SEEN | UNSEEN | HM | AUC |
| UT-Zappos | CLIP [59] | 24.4 | 40.5 | 4.8 | 42.0 | 6.7 | 1.5 | 24.1 | 58.3 | 11.9 | 45.7 | 15.3 | 4.4 | 52.5 | 74.5 | 44.9 | 68.1 | 39.2 | 24.6 |
| | LDE (*text*) | 18.3 | 41.5 | **5.7** | 32.6 | 6.3 | 1.1 | 24.1 | 58.8 | 11.9 | 45.7 | 14.1 | 4.0 | 46.8 | 73.6 | 41.3 | 70.9 | 33.2 | 20.5 |
| | GDE (*text*) | **23.2** | 42.1 | 5.2 | **38.8** | 6.3 | **1.2** | **25.3** | **60.0** | **17.0** | **48.2** | **18.9** | **6.4** | 47.8 | **74.9** | 43.7 | **71.4** | 34.5 | **21.9** |
| | PDE (*text*) | 21.0 | **47.0** | 5.1 | 37.5 | **6.5** | **1.2** | 22.1 | 56.1 | 15.8 | 43.8 | 18.1 | 5.6 | **54.8** | 72.7 | **46.9** | 68.1 | **39.5** | **24.9** |
| | LDE (*image*) | 15.1 | 44.6 | 3.2 | 21.7 | 4.7 | 0.5 | 13.9 | 52.6 | 5.6 | 32.1 | 6.6 | 0.9 | 21.3 | 50.2 | 7.0 | 42.0 | 8.2 | 1.6 |
| | GDE (*image*) | 28.1 | 56.1 | 23.9 | 43.7 | **23.7** | 8.6 | 36.3 | 64.1 | 31.4 | 55.9 | 29.3 | **13.9** | **48.4** | 72.4 | 43.1 | 68.5 | **42.1** | 25.2 |
| | PDE (*image*) | **29.4** | **62.9** | **24.0** | **48.5** | 22.0 | **8.7** | **36.8** | **66.3** | **32.8** | **57.1** | **32.2** | **15.5** | 47.6 | **74.7** | **45.4** | **69.0** | 42.8 | **26.7** |
| MIT-States | CLIP [59] | 27.4 | 42.5 | 24.0 | 35.8 | 19.9 | 6.6 | 33.0 | 52.1 | 30.6 | 45.3 | 26.3 | 11.1 | 45.8 | 61.3 | 43.7 | 57.8 | 39.3 | 22.0 |
| | LDE (*text*) | 24.3 | 40.4 | 17.9 | 30.8 | 15.0 | 3.9 | 30.6 | 51.2 | 24.7 | 43.0 | 21.9 | 8.2 | 43.0 | 58.5 | 34.2 | 51.5 | 30.0 | 14.1 |
| | GDE (*text*) | 25.6 | 41.0 | 20.0 | 32.4 | 17.0 | 4.9 | **32.6** | **51.7** | 27.8 | **45.2** | 24.5 | 10.0 | 39.5 | 58.8 | 34.9 | 52.1 | 30.6 | 14.8 |
| | PDE (*text*) | **26.4** | **42.2** | **24.1** | **34.7** | **19.7** | **6.4** | 31.6 | 51.5 | **29.2** | 45.0 | **26.2** | **10.7** | **46.5** | **61.7** | **43.6** | **58.9** | **40.0** | **22.5** |
| | LDE (*image*) | 13.8 | 24.3 | 10.3 | 16.3 | 7.9 | 1.1 | 15.3 | 30.5 | 15.0 | 20.9 | 11.1 | 2.0 | 19.0 | 34.4 | 18.8 | 27.5 | 14.8 | 3.6 |
| | GDE (*image*) | 20.8 | **34.1** | 19.1 | **24.9** | 14.4 | 3.4 | 28.1 | 45.3 | 30.7 | **36.1** | 23.4 | 8.6 | **40.5** | **50.1** | 36.2 | **40.5** | 27.0 | 11.7 |
| | PDE (*image*) | **21.5** | 33.2 | **28.2** | 23.5 | **17.2** | **4.8** | **30.0** | **45.6** | **40.5** | 35.0 | **26.9** | **11.1** | 33.5 | 49.5 | **46.1** | 39.9 | **30.8** | **14.6** |
| CGQA | CLIP [59] | 14.9 | 25.5 | 6.5 | 18.7 | 6.7 | 0.8 | 15.2 | 30.2 | 8.5 | 25.7 | 8.9 | 1.6 | 16.9 | 38.6 | 9.9 | 34.8 | 11.7 | 2.7 |
| | LDE (*text*) | 9.1 | 26.0 | 4.3 | 15.3 | 4.2 | 0.4 | 10.4 | 30.2 | 5.8 | 23.7 | 6.4 | 0.9 | 10.5 | 37.0 | 7.2 | 27.7 | 7.7 | 1.3 |
| | GDE (*text*) | 11.7 | 25.7 | 5.5 | 17.2 | 5.7 | 0.6 | 13.4 | 30.3 | 7.5 | 26.1 | 8.0 | 1.3 | 10.9 | **37.4** | 7.2 | 28.1 | 7.8 | 1.4 |
| | PDE (*text*) | **15.0** | **26.6** | **7.6** | **18.1** | **7.1** | **1.0** | **14.4** | **31.2** | **8.0** | **26.1** | **8.8** | **1.5** | **15.3** | 36.5 | **9.3** | **33.0** | **10.5** | **2.3** |
| | LDE (*image*) | 1.4 | 15.5 | 0.6 | 5.1 | 0.8 | 0.0 | 1.6 | 16.0 | 0.7 | 7.1 | 0.9 | 0.03 | 2.6 | 20.6 | 1.3 | 10.2 | 1.4 | 0.1 |
| | GDE (*image*) | **11.7** | 26.0 | 4.9 | **11.7** | 4.4 | 0.4 | **13.4** | 30.3 | 7.5 | **25.4** | 8.0 | **1.3** | **18.2** | 37.4 | 12.0 | **21.7** | 10.5 | 1.9 |
| | PDE (*image*) | 11.3 | **27.8** | **8.2** | 11.3 | **5.4** | **0.5** | 11.8 | **32.3** | **9.7** | 15.9 | **6.4** | 0.8 | 17.9 | **40.8** | **16.5** | 21.0 | **11.1** | **2.2** |

This experiment evaluates whether our method, PDE, consistently performs well regardless of the backbone. Table 3 reports results in the closed-world setting, while Table 4 presents results in the open-world setting across three backbones—CLIP (RN50), CLIP (ViT-L/14), and SigLIP (ViT-SO400M/14). Table 5 presents results in SigLIP2 (ViT-SO400M/14, patch size of 378) backbone.

## B.2 ABLATION ON SCALING FACTOR S

As described in Section 3.3, our compositional method can be written as a product of a *carrier* (direction) and a *scale* (magnitude). Table 6 summarizes an ablation on how to choose this scale factor, while keeping the number of composing primitives fixed to two (one attribute and one object) in all cases. We first compare fixed global scales $s \in \{1, 1.5, 2, 2.5\}$ to assess sensitivity to the

Table 4: Ablation on backbone architecture in compositional classification, open-world scenario. The highest values within each modality are in **bold**; The second-highest values are underlined.

| Dataset | Method | CLIP, RN50 | | | | | | CLIP, ViT-L/14 | | | | | | SigLIP, ViT-SO400M/14 | | | | | |
|---|---|---|---|---|---|---|---|---|---|---|---|---|---|---|---|---|---|---|---|
| | | ATTR | OBJ | SEEN | UNSEEN | HM | AUC | ATTR | OBJ | SEEN | UNSEEN | HM | AUC | ATTR | OBJ | SEEN | UNSEEN | HM | AUC |
| UT-Zappos | CLIP [59] | 18.8 | 38.4 | 4.8 | 22.3 | 5.6 | 0.7 | 18.8 | 57.4 | 11.9 | 23.8 | 12.0 | 2.3 | 47.2 | 70.1 | 44.9 | 49.9 | 32.5 | 16.5 |
| | LDE (*text*) | 15.1 | 35.0 | **5.7** | 18.5 | **5.7** | **0.7** | **19.2** | 57.2 | 11.9 | 20.0 | 11.1 | 1.9 | 45.1 | 72.1 | 41.3 | 55.2 | 31.1 | 16.6 |
| | GDE (*text*) | **19.9** | 37.8 | 5.2 | 20.3 | 5.7 | 0.7 | 18.7 | 59.9 | 17.0 | 21.4 | 12.2 | 2.5 | 46.3 | **73.0** | 43.7 | **56.3** | 32.7 | 17.8 |
| | PDE (*text*) | 17.4 | **44.0** | 5.0 | 20.0 | 5.4 | 0.6 | 17.3 | 57.5 | 15.8 | **23.4** | 11.9 | 2.6 | **52.0** | 69.4 | **46.9** | 54.0 | **35.2** | **19.0** |
| | LDE (*image*) | 12.3 | 41.3 | 3.2 | 10.4 | 3.2 | 0.2 | 9.8 | 48.0 | 5.6 | 14.9 | 2.3 | 0.2 | 18.8 | 44.6 | 7.0 | 27.1 | 7.3 | 1.0 |
| | GDE (*image*) | 21.1 | 53.9 | 23.9 | 22.4 | 13.6 | 3.4 | **28.6** | 61.7 | 31.3 | **33.3** | 19.0 | 6.7 | **41.0** | 70.4 | 41.3 | **49.6** | 31.2 | 15.3 |
| | PDE (*image*) | **21.4** | **59.2** | **24.0** | **25.3** | **15.2** | **4.1** | 28.1 | **62.7** | **32.8** | 32.7 | **21.8** | **7.9** | 39.0 | **71.7** | **45.4** | 48.5 | **31.0** | **16.2** |
| MIT-States | CLIP [59] | 13.8 | 38.8 | 24.0 | 6.3 | 6.2 | 1.0 | 15.6 | 47.7 | 30.6 | 8.3 | 8.4 | 1.7 | 24.3 | 54.5 | 43.7 | 16.2 | 16.6 | 5.4 |
| | LDE (*text*) | **16.8** | **39.8** | 17.9 | **8.2** | 6.9 | 0.9 | 21.1 | **50.7** | 24.7 | 13.8 | 11.9 | 2.5 | **29.3** | **56.9** | 34.2 | **21.7** | 18.2 | 5.6 |
| | GDE (*text*) | **16.8** | 39.5 | 20.0 | 8.0 | 7.3 | 1.1 | **21.3** | 49.9 | 27.8 | 13.0 | 12.1 | 2.6 | **29.3** | 56.8 | 34.9 | 21.6 | 18.4 | 5.8 |
| | PDE (*text*) | 13.8 | 39.3 | 24.1 | 6.9 | 6.9 | 1.1 | 14.8 | 48.8 | 29.2 | 8.4 | 8.6 | 1.7 | 28.0 | 56.4 | 43.6 | 20.8 | **19.8** | **6.9** |
| | LDE (*image*) | 10.1 | 28.3 | 10.3 | 3.6 | 3.0 | 0.2 | 11.0 | 34.8 | 15.0 | 5.6 | 4.6 | 0.4 | 13.8 | 37.1 | 18.8 | 6.9 | 6.0 | 0.7 |
| | GDE (*image*) | **13.4** | 32.6 | 19.1 | **3.9** | 4.2 | 0.4 | 18.5 | 43.6 | 29.7 | **8.5** | 9.3 | 1.8 | 21.9 | **49.1** | 35.9 | **11.8** | **12.5** | 3.0 |
| | PDE (*image*) | 12.3 | 32.5 | **28.2** | 3.8 | 4.6 | 0.6 | **19.1** | **43.8** | **40.5** | 8.1 | **9.5** | **2.2** | **22.3** | 48.1 | **46.1** | 10.5 | 11.9 | **3.3** |
| CGQA | CLIP [59] | 5.3 | 20.6 | 6.4 | 1.0 | 1.2 | 0.1 | 4.7 | 24.4 | 8.4 | 2.3 | 2.5 | 0.1 | 4.1 | 31.5 | 9.8 | 2.4 | 2.7 | 0.2 |
| | LDE (*text*) | 3.5 | 22.2 | 3.1 | 0.6 | 0.7 | 0.0 | 3.8 | **27.7** | 5.8 | 1.92 | 1.66 | 0.06 | 3.9 | 35.2 | 7.0 | **3.3** | 2.6 | 0.1 |
| | GDE (*text*) | 4.6 | 21.6 | 4.5 | 1.0 | 1.1 | 0.0 | 5.2 | 26.2 | 7.4 | **2.4** | **2.7** | 0.12 | 3.9 | **35.3** | 7.1 | 3.2 | 2.6 | 0.1 |
| | PDE (*text*) | **5.9** | **23.1** | 7.6 | **1.7** | **1.9** | 0.09 | 4.4 | 27.2 | 7.8 | 2.2 | 2.6 | **0.13** | 4.3 | 31.7 | 9.3 | 2.8 | **3.2** | **0.19** |
| | LDE (*image*) | 0.5 | 6.4 | 0.0 | 0.0 | 0.0 | 0.0 | 0.00 | 0.00 | 0.00 | 0.00 | 0.00 | 0.00 | 0.6 | 14.3 | 1.0 | 0.1 | 0.0 | 0.0 |
| | GDE (*image*) | **6.0** | 23.3 | 4.7 | 1.4 | 1.1 | 0.0 | 5.2 | 26.2 | 7.4 | **2.4** | **2.7** | 0.12 | **12.1** | 36.0 | 12.0 | 4.5 | **4.6** | **0.4** |
| | PDE (*image*) | 4.8 | **23.9** | 8.2 | 0.4 | 0.6 | 0.0 | 5.3 | **30.1** | 9.6 | 1.2 | 1.1 | 0.04 | 10.2 | **37.5** | 16.3 | 25.3 | 2.9 | 0.3 |

Table 5: Compositional classification results on the UT-Zappos, MIT-States, and CGQA datasets, in ViT-SO400M-14-SigLIP2-378. The highest values within each modality are in **bold**. The second-highest values are underlined.

| Dataset | Method | Closed-World | | | | | | Open-World | | | | | |
|---|---|---|---|---|---|---|---|---|---|---|---|---|---|
| | | Attr | Obj | Seen | Unseen | HM | AUC | A | O | S | U | H | AUC |
| UT-Zappos | CLIP | 51.5 | 80.2 | 42.7 | 75.4 | 38.9 | 24.8 | 45.6 | 72.3 | 42.7 | 50.1 | 30.2 | 15.0 |
| | LDE (text) | 46.1 | 78.9 | 37.6 | 77.3 | 34.7 | 21.2 | 42.6 | 77.8 | 37.6 | 56.8 | 31.3 | 15.8 |
| | GDE (text) | 47.2 | 79.2 | 39.8 | 77.3 | 36.3 | 22.5 | 43.7 | **78.0** | 39.8 | **57.0** | 32.7 | 16.9 |
| | PDE (text) | **52.8** | **81.0** | **43.6** | **78.1** | **40.6** | **26.3** | **47.2** | 77.1 | **43.6** | 56.8 | **34.6** | **18.6** |
| | LDE (image) | 25.5 | 56.8 | 8.9 | 45.6 | 10.4 | 2.2 | 20.5 | 51.3 | 8.9 | 26.0 | 7.8 | 1.2 |
| | GDE (image) | **49.1** | 71.4 | 41.9 | 66.7 | **42.1** | 24.2 | **40.1** | 69.8 | **45.3** | 46.5 | 31.3 | 16.0 |
| | PDE (image) | 48.3 | **75.0** | **44.0** | **68.5** | **42.1** | **25.7** | 40.0 | **71.6** | 44.0 | **48.8** | 31.3 | **16.4** |
| MIT-States | CLIP | 47.0 | 60.9 | 42.3 | 59.0 | 39.5 | 21.9 | 25.2 | 52.8 | 42.3 | 15.2 | 15.9 | 4.9 |
| | LDE (text) | 40.6 | 59.0 | 34.3 | 53.1 | 31.5 | 15.0 | **30.5** | 56.9 | 34.3 | **21.9** | 18.8 | 5.8 |
| | GDE (text) | 41.0 | 59.2 | 35.2 | 53.8 | 32.3 | 15.7 | 30.4 | **57.0** | 35.2 | **21.9** | **19.1** | 6.0 |
| | PDE (text) | **47.8** | **61.7** | **43.4** | **59.9** | **40.2** | **22.8** | 27.8 | 55.7 | **43.4** | 19.2 | 18.9 | **6.4** |
| | LDE (image) | 21.9 | 38.8 | 21.6 | 30.9 | 17.4 | 4.9 | 14.4 | 40.5 | 22.1 | 6.5 | 6.8 | 1.0 |
| | GDE (image) | 34.3 | **52.6** | 38.7 | **44.5** | 30.1 | 14.0 | 23.9 | **50.4** | 38.6 | **13.0** | **14.0** | 3.7 |
| | PDE (image) | **35.4** | 52.2 | **49.1** | 43.6 | **33.5** | **17.2** | **24.3** | 49.7 | **49.1** | 12.2 | 13.9 | **4.2** |
| CGQA | CLIP | 14.0 | 31.9 | 7.9 | 34.2 | 9.9 | 9.1 | 3.2 | 26.8 | 7.9 | 2.0 | 2.1 | 0.09 |
| | LDE (text) | 8.6 | 30.2 | 5.8 | 27.1 | 6.7 | 1.0 | 3.2 | 29.1 | 5.7 | 2.3 | 2.0 | 0.1 |
| | GDE (text) | 8.7 | 30.8 | 5.8 | 27.0 | 6.8 | 1.0 | 3.0 | 29.5 | 5.7 | 2.3 | 1.9 | 0.1 |
| | PDE (text) | **13.2** | **31.0** | **7.9** | **31.6** | **9.2** | **1.8** | **4.3** | 2.7 | **7.9** | **2.9** | **2.7** | **0.15** |
| | LDE (image) | 2.1 | 18.1 | 0.8 | 9.2 | 1.0 | 0.0 | 0.7 | 8.1 | 0.5 | 0.0 | 0.0 | 0.0 |
| | GDE (image) | **15.0** | 33.6 | 9.5 | **20.1** | **8.6** | 1.3 | **8.2** | 27.2 | 7.7 | **3.3** | **2.9** | **0.2** |
| | PDE (image) | 13.8 | **36.8** | **12.6** | 18.9 | 8.5 | **1.4** | 6.9 | **33.1** | **12.6** | 1.8 | 1.9 | 0.11 |

overall scale magnitude. Our default choice in the main experiments is $s = 2$, which roughly restores the norm of an equal-weight GDE/LDE composition. We then consider two data-driven variants: (i) $s = \text{all}$ assigns a separate scale $s_{a,o}$ to each attribute–object pair so that the norm of the PDE composite matches that of the corresponding GDE composite, and (ii) $s = \text{median}$ takes the median of these pair-specific scales $\{s_{a,o}\}$ over the dataset and uses this single global value for all compositions. These results indicate that PDE is robust to different ways of restoring the composite norm, and that the simple choice $s = 2$ already achieves performance close to the best of these variants.

Table 6: Compositional classification with PDE variants only (SigLIP backbone). We compare six strategies for setting the scale factor $s$ in the *carrier* $\times$ *scale* decomposition: fixed global scales $s \in \{1, 1.5, 2, 2.5\}$, a pair-wise data-driven scale $s = $ all that assigns a separate $s_{a,o}$ to each attribute–object pair so that the norm of the PDE composite matches that of the corresponding GDE composite, and a global data-driven scale $s = $ median equal to the dataset-wise median of $\{s_{a,o}\}$. The number of composing primitives is always two (one attribute and one object); $s$ only controls the composite norm. The highest values within each modality are in bold, and the second-highest values are underlined.

| Dataset | Method | Closed-World | | | | | | Open-World | | | | | |
|---|---|---|---|---|---|---|---|---|---|---|---|---|---|
| | | Attr | Obj | Seen | Unseen | HM | AUC | A | O | S | U | H | AUC |
| UT-Zappos | PDE (text; $s=1$) | 51.9 | 75.5 | 43.3 | **71.9** | 38.3 | 24.0 | 49.1 | 72.7 | 43.3 | 56.4 | 34.9 | 18.4 |
| | PDE (text; $s=1.5$) | 53.6 | 74.5 | 45.7 | 70.7 | 39.3 | **25.1** | 50.9 | 71.1 | 45.7 | 56.0 | **35.7** | **19.2** |
| | PDE (text; $s=2$) | 54.8 | 72.7 | 46.9 | 68.1 | 39.5 | 24.9 | 52.0 | 69.4 | 46.9 | 54.0 | 35.2 | 19.0 |
| | PDE (text; $s=2.5$) | **56.4** | 71.1 | **47.9** | 66.9 | **39.8** | 25.0 | **53.3** | 67.8 | **47.9** | 52.5 | 35.3 | 18.8 |
| | PDE (text; $s=$all) | 50.9 | 75.2 | 41.7 | 71.3 | 36.9 | 22.8 | 48.3 | 72.4 | 41.7 | 55.5 | 33.6 | 17.4 |
| | PDE (text; $s=$median) | 51.6 | **75.6** | 42.8 | **71.9** | 38.1 | 23.7 | 49.0 | **73.0** | 42.8 | **56.6** | 34.7 | 18.2 |
| | PDE (image; $s=1$) | 44.1 | 71.8 | 42.0 | 67.2 | 40.5 | 23.9 | 37.0 | 68.8 | 42.0 | 45.2 | 27.1 | 13.4 |
| | PDE (image; $s=1.5$) | 46.0 | 73.5 | 44.0 | 67.9 | 41.8 | 25.5 | 38.1 | 70.8 | 44.0 | 46.3 | 28.8 | 14.8 |
| | PDE (image; $s=2$) | 47.6 | 74.7 | 45.4 | 69.0 | **42.8** | **26.7** | 39.0 | 71.7 | 45.4 | 48.5 | 31.0 | 16.2 |
| | PDE (image; $s=2.5$) | **49.6** | **75.7** | 44.6 | **69.6** | 42.6 | 26.6 | **39.7** | **72.1** | 44.6 | **48.8** | **32.0** | **16.4** |
| | PDE (image; $s=$all) | 47.0 | 72.3 | 43.4 | 68.0 | 42.4 | 25.5 | 39.2 | 69.7 | 43.4 | 46.2 | 28.9 | 14.8 |
| | PDE (image; $s=$median) | 44.5 | 72.2 | 42.6 | 67.3 | 40.7 | 24.3 | 37.2 | 69.1 | 42.6 | 45.3 | 27.4 | 13.7 |
| MIT-States | PDE (text; $s=1$) | 46.2 | 61.4 | 42.4 | 58.6 | 38.9 | 21.6 | 29.4 | 56.4 | 42.4 | 21.7 | 20.4 | **7.1** |
| | PDE (text; $s=1.5$) | **46.6** | **61.7** | 43.2 | 58.9 | 39.4 | 22.2 | 28.7 | 56.4 | 43.2 | 21.1 | 20.0 | 7.0 |
| | PDE (text; $s=2$) | 46.5 | 61.7 | 43.6 | 58.9 | **40.0** | **22.5** | 28.0 | 56.4 | 43.6 | 20.8 | 19.8 | 6.9 |
| | PDE (text; $s=2.5$) | 46.1 | 61.6 | 43.2 | **59.0** | 39.7 | 22.4 | 26.7 | 56.4 | 43.2 | 20.1 | 19.0 | 6.6 |
| | PDE (text; $s=$all) | 43.5 | 61.0 | 38.8 | 57.2 | 36.0 | 18.9 | **30.3** | **56.8** | 38.8 | **23.1** | **20.6** | 7.0 |
| | PDE (text; $s=$median) | 46.2 | 61.5 | 42.4 | 58.5 | 38.8 | 21.6 | 29.3 | 56.3 | 42.4 | 21.7 | 20.4 | 7.1 |
| | PDE (image; $s=1$) | 30.8 | 49.1 | 46.3 | **41.6** | **31.6** | 15.4 | **25.4** | 48.1 | 46.3 | **14.0** | **14.9** | **4.5** |
| | PDE (image; $s=1.5$) | 31.9 | 49.4 | **46.8** | 41.2 | 31.5 | 15.4 | 24.7 | 48.1 | **46.8** | 12.7 | 13.9 | 4.1 |
| | PDE (image; $s=2$) | 33.5 | **49.5** | 46.1 | 39.9 | 30.8 | 14.6 | 22.3 | 48.1 | 46.1 | 10.5 | 11.9 | 3.3 |
| | PDE (image; $s=2.5$) | **34.8** | 49.1 | 43.9 | 38.3 | 29.4 | 13.4 | 18.8 | 47.5 | 43.9 | 8.4 | 10.1 | 2.5 |
| | PDE (image; $s=$all) | 31.2 | 48.5 | 46.0 | 40.3 | 31.1 | 14.8 | 24.5 | 47.9 | 46.0 | 11.8 | 13.1 | 3.8 |
| | PDE (image; $s=$median) | 32.8 | 49.3 | 46.7 | 40.4 | 31.3 | 15.0 | 23.4 | 48.1 | 46.7 | 11.5 | 12.8 | 3.7 |
| CGQA | PDE (text; $s=1$) | 14.9 | 36.5 | 9.2 | 32.7 | 10.5 | 2.3 | 4.6 | 31.4 | 9.2 | **3.1** | 3.3 | 0.2 |
| | PDE (text; $s=1.5$) | **15.5** | 36.2 | 9.4 | 32.5 | 10.6 | 2.3 | 4.5 | 31.3 | 9.4 | **3.1** | **3.3** | **0.2** |
| | PDE (text; $s=2$) | 15.3 | 36.5 | 9.3 | **33.0** | 10.5 | 2.3 | 4.3 | 31.7 | 9.3 | 2.8 | 3.2 | 0.19 |
| | PDE (text; $s=2.5$) | 14.9 | **37.2** | 9.0 | **33.0** | 10.3 | 2.2 | 4.1 | 32.3 | 9.0 | 2.6 | 2.9 | 0.2 |
| | PDE (text; $s=$all) | 15.1 | 36.9 | **9.6** | 31.6 | **10.8** | 2.2 | **5.1** | **32.8** | **9.6** | 2.8 | 2.8 | 0.2 |
| | PDE (text; $s=$median) | 15.4 | 36.2 | 9.4 | 32.5 | 10.7 | **2.3** | 4.5 | 31.2 | 9.3 | 3.1 | 3.3 | 0.2 |
| | PDE (image; $s=1$) | 14.5 | 37.7 | 10.3 | **22.4** | 8.4 | 1.4 | 10.3 | 34.9 | 10.2 | 2.9 | 2.6 | 0.2 |
| | PDE (image; $s=1.5$) | 17.8 | 39.4 | 14.6 | 22.2 | 10.8 | 2.0 | **11.6** | 36.5 | 14.5 | 2.9 | 2.9 | 0.2 |
| | PDE (image; $s=2$) | 17.9 | 40.8 | 16.5 | 21.0 | 11.1 | 2.2 | 10.2 | 37.5 | 16.3 | **25.3** | 2.9 | **0.3** |
| | PDE (image; $s=2.5$) | 15.9 | **41.3** | **17.0** | 20.3 | 11.1 | 2.2 | 7.4 | **37.7** | **16.9** | 2.0 | 2.9 | 0.2 |
| | PDE (image; $s=$all) | 16.2 | 37.8 | 12.5 | 22.3 | 9.9 | 1.8 | 11.3 | 35.7 | 12.4 | 3.1 | **3.1** | 0.2 |
| | PDE (image; $s=$median) | **18.1** | 40.5 | 16.2 | 21.8 | **11.2** | 2.2 | 10.8 | 37.2 | 16.1 | 2.4 | 2.8 | 0.3 |

## B.3 HYPERPARAMETER ROBUSTNESS

We assessed robustness of the PDE compositional factorizer across datasets (UT-Zappos, MIT-States, CGQA), modalities (image/text), and evaluation setups (closed/open world). All experiments were conducted with the SigLIP(ViT-SO400M/14) vision–language encoder. For each dataset–modality–setup triplet, we conducted three 1D sweeps: (i) covariance shrinkage on Figure 3, (ii) the mixture weight $\eta$ under `eta_mode=fixed` on Figure 4, Figure 5, Figure 6, and (iii) the prior precision scale $\lambda$ used as the isotropic prior precision term in Eq. (28) on Figure 7. Other hyperparameters were held at defaults (`shrinkage= 0.15`, `eta_mode= auto_oas`, `cov_mode= full`, $s = 2$). As shown in Figure 7, PDE performance is relatively insensitive to $\lambda$ around the chosen default, exhibiting a broad plateau across datasets, modalities, and evaluation setups. Performance was summarized by the AUC.

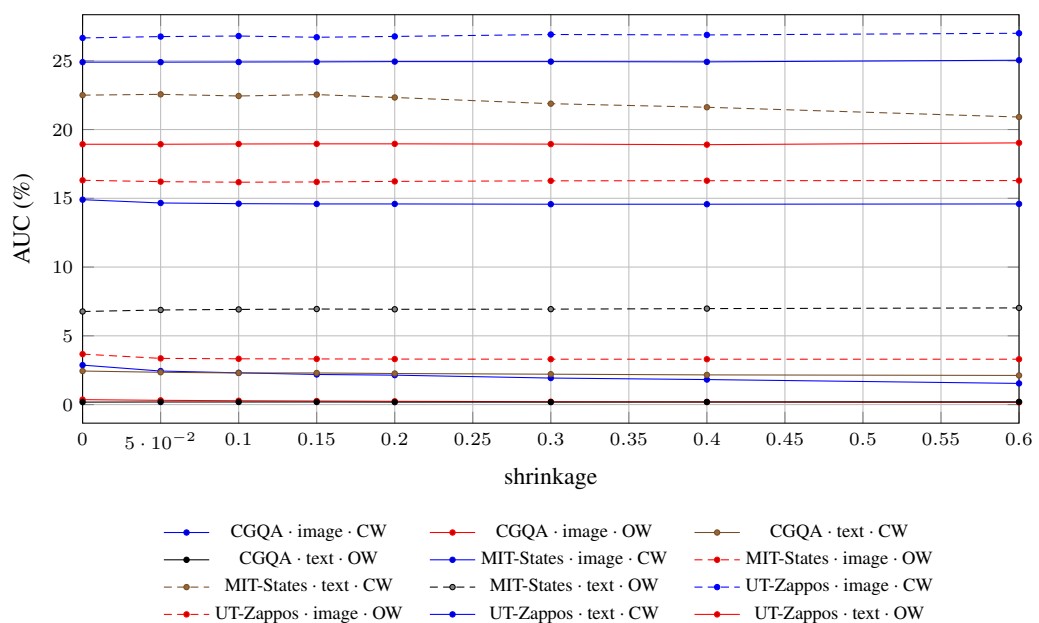

Figure 3: AUC vs Shrinkage for each DATASET·MODALITY·OW.

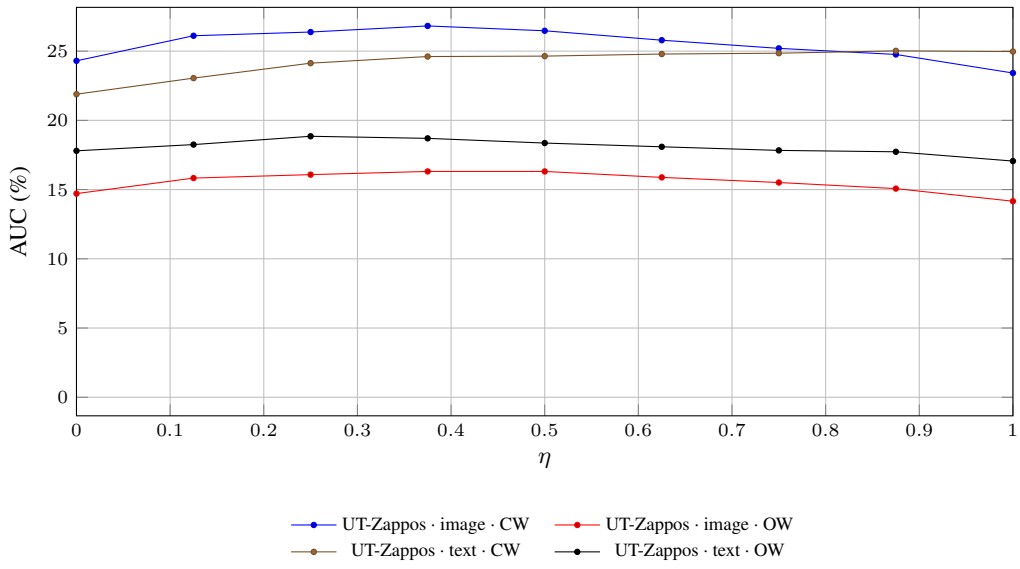

Figure 4: AUC(%) vs $\eta$ on UT-Zappos.

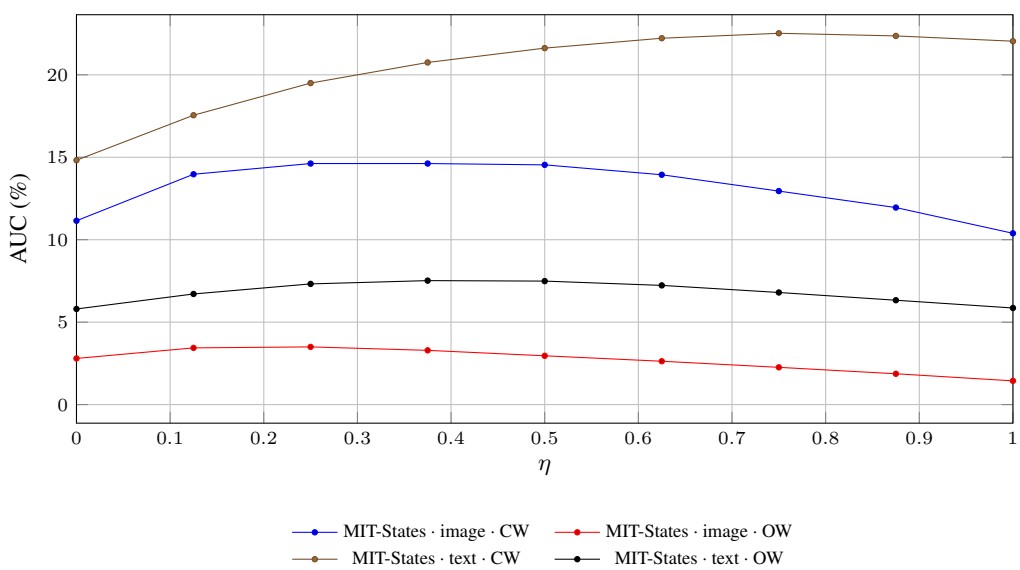

Figure 5: AUC(%) vs $\eta$ on MIT-States.

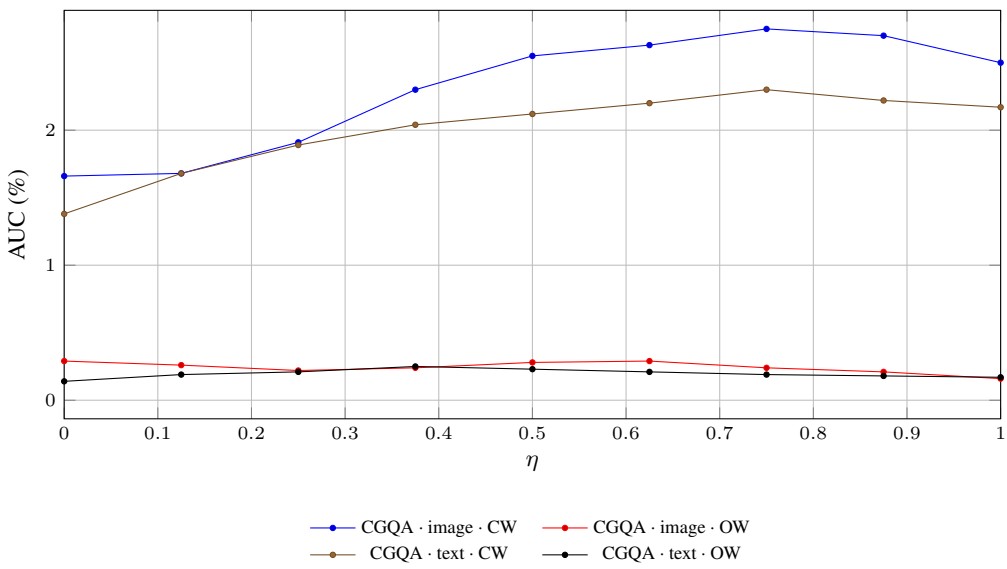

Figure 6: AUC(%) vs $\eta$ on CGQA.

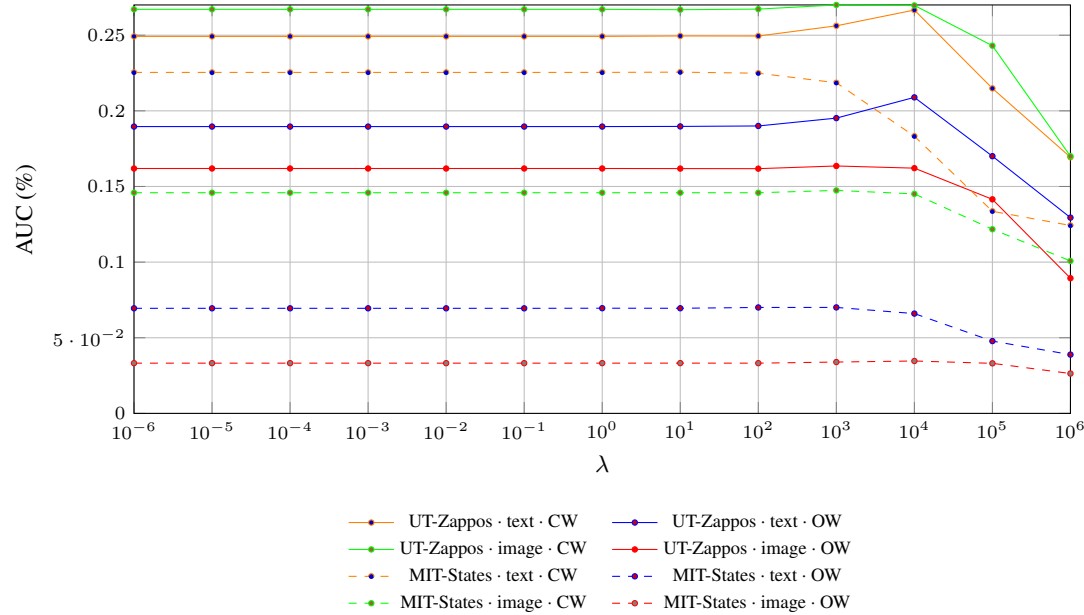

Figure 7: AUC (%) vs $\lambda$ on UT-Zappos and MIT-States.

## B.4 PE AND PLDE

To mirror our uncertainty-aware extension of GDE, we provide a Euclidean counterpart for LDE, termed **PLDE**. Unlike PDE, PLDE operates entirely in Euclidean space (no $\log / \exp$ maps); implementation reuses the same caching and hyperparameter choices as PDE. Also, we implement E and PE; E is comprising composite with addition of two primitive vectors, and PE is our approach applied in E. See results in Table 7.

## B.5 EXPERIMENT OF ETA WITHOUT ROOT

We show experiment results on ViT-L/14, with eta of

$$\eta_{(a,o)} = \left(1 - \delta_a\right)\left(1 - \delta_o\right) \in [0, 1].$$

In Table 8. The philosophy of this design is to decrease confidence when multiple primitives are combined.

## B.6 PERFORMANCE ON FIXED HYPERPARAMTER SETTINGS.

Table 9 reports an *upper bound* achievable via lightweight hyperparameter tuning under the same datasets, splits, backbone (ViT-L/14), scoring, and metrics as the main setup. We evaluated 22 LLM-recommended configurations that vary only three hyperparameters—shrinkage, $\lambda$, and $\eta$—and report the *best-of-22*, i.e., the single configuration achieving the highest performance among these candidates on the **test** set. Because selection was performed on the test set, these numbers should be interpreted as an optimistic upper bound rather than model performance under a standard hold-out validation protocol.

## B.7 EMPIRICAL VALIDATION OF THE GAUSSIAN APPROXIMATION (SHAPIRO–WILK)

PDE assumes that the ideal-word distributions in the decomposed CLIP space are reasonably close to multivariate Gaussian. This assumption is consistent with prior works Lu et al. (2022); Wang et al. (2024); Zhu et al. (2024) that treats CLIP embeddings as approximately gaussian. However, to check how well this approximation holds in practice, we perform a normality check using the Shapiro–Wilk test on the GDE representations.

Table 7: Compositional classification with **E**, **PE**, **PLDE**, and **PDE** on SigLIP (ViT-SO400M/14). E denotes just simple addition of two CLIP embeddings. PE denotes probabilistic approach for simple addition of two CLIP embeddings. PLDE denotes PDE approach for LDE. The highest values within each modality are in **bold**; The second-highest values are underlined.

| Dataset | Method | Closed-World | | | | | | Open-World | | | | | |
|---|---|---|---|---|---|---|---|---|---|---|---|---|---|
| | | Attr | Obj | Seen | Unseen | HM | AUC | A | O | S | U | H | AUC |
| UT-Zappos | E (text) | 46.0 | **76.8** | 41.1 | **72.3** | 33.7 | 21.0 | 44.0 | **74.9** | 41.1 | 56.6 | 31.6 | 16.6 |
| | PE (text) | 53.5 | 75.4 | 45.4 | 71.8 | 39.9 | 25.8 | 51.0 | 71.9 | 45.4 | 56.6 | 35.8 | 19.5 |
| | PLDE (text) | 53.9 | 75.3 | 45.6 | 71.8 | **40.2** | **26.1** | 51.2 | 71.9 | 45.6 | **56.7** | **36.0** | **19.7** |
| | PDE (text) | **54.8** | 72.7 | **46.9** | 68.1 | 39.5 | 24.9 | **52.0** | 69.4 | **46.9** | 54.0 | 35.2 | 19.0 |
| | E (image) | 39.4 | 67.4 | 31.6 | 61.6 | 31.0 | 14.8 | 32.9 | 63.9 | 31.6 | 36.0 | 21.1 | 7.6 |
| | PE (image) | 36.8 | 69.2 | 39.3 | 62.8 | 35.0 | 19.2 | 33.1 | 67.4 | 39.3 | 41.2 | 24.3 | 10.8 |
| | PLDE (image) | 21.4 | 52.5 | 15.1 | 45.7 | 11.9 | 3.1 | 20.8 | 51.6 | 15.1 | 25.4 | 10.1 | 1.9 |
| | PDE (image) | **47.6** | **74.7** | **45.4** | **69.0** | **42.8** | **26.7** | **39.0** | **71.7** | **45.4** | **48.5** | **31.0** | **16.2** |
| MIT-States | E (text) | 39.3 | 58.6 | 34.4 | 51.5 | 30.3 | 14.3 | **29.3** | **56.9** | 34.4 | **21.6** | 18.2 | 5.7 |
| | PE (text) | 46.3 | 61.5 | 42.6 | **58.9** | 39.1 | 21.9 | 28.4 | 56.1 | 42.6 | 20.7 | 19.5 | 6.8 |
| | PLDE (text) | 46.3 | 61.5 | 42.7 | 58.7 | 38.9 | 21.8 | 28.0 | 55.9 | 42.7 | 20.0 | 19.1 | 6.5 |
| | PDE (text) | **46.5** | **61.7** | **43.6** | **58.9** | **40.0** | **22.5** | 28.0 | 56.4 | **43.6** | 20.8 | **19.8** | **6.9** |
| | E (image) | 29.8 | 49.1 | 32.9 | **40.2** | 26.6 | 10.5 | 23.0 | **48.3** | 32.9 | 13.6 | 12.8 | 3.1 |
| | PE (image) | 28.1 | 47.1 | **47.9** | 40.7 | **31.0** | **15.2** | **26.2** | 46.9 | **47.9** | **15.5** | **16.1** | **5.1** |
| | PLDE (image) | 19.4 | 33.6 | 29.7 | 26.6 | 18.7 | 5.7 | 19.1 | 34.0 | 29.7 | 7.1 | 7.5 | 1.3 |
| | PDE (image) | **33.5** | **49.5** | 46.1 | 39.9 | 30.8 | 14.6 | 22.3 | 48.1 | 46.1 | 10.5 | 11.9 | 3.3 |
| CGQA | E (text) | 11.3 | **37.1** | 7.8 | 28.1 | 8.4 | 1.5 | 4.3 | **34.8** | 7.7 | 3.0 | 2.7 | 0.1 |
| | PE (text) | **16.2** | 36.1 | **10.2** | 32.5 | **11.2** | **2.5** | **5.0** | 31.1 | **10.1** | 3.8 | 3.9 | 0.3 |
| | PLDE (text) | 16.0 | 36.1 | 9.8 | 32.3 | 11.0 | 2.4 | 4.9 | 30.7 | 9.7 | 3.4 | 3.7 | 0.2 |
| | PDE (text) | 15.3 | 36.5 | 9.3 | **33.0** | 10.5 | 2.3 | 4.3 | 31.7 | 9.3 | 2.8 | 3.2 | 0.2 |
| | E (image) | 9.3 | 35.1 | 4.6 | **22.0** | 5.3 | 0.7 | 3.6 | 29.3 | 4.2 | 1.5 | 1.2 | 0.0 |
| | PE (image) | 12.7 | 35.1 | 8.9 | 21.9 | 7.7 | 1.2 | 8.7 | 31.6 | 8.5 | 1.7 | 1.7 | 0.1 |
| | PLDE (image) | 4.0 | 22.2 | 1.5 | 12.0 | 1.7 | 0.1 | 1.9 | 17.6 | 1.0 | 0.1 | 0.0 | 0.0 |
| | PDE (image) | **17.9** | **40.8** | **16.5** | 21.0 | **11.1** | **2.2** | **10.2** | **37.5** | **16.3** | **25.3** | **2.9** | **0.3** |

Table 8: Experiment on OAS-style automatic eta method without square root. The experiment is held on ViT-L/14. The highest values within each modality are in **bold**; The second-highest values are underlined.

| Dataset | Method | Closed-World | | | | | | Open-World | | | | | |
|---|---|---|---|---|---|---|---|---|---|---|---|---|---|
| | | Attr | Obj | Seen | Unseen | HM | AUC | A | O | S | U | H | AUC |
| UT-Zappos | CLIP | 24.1 | 58.3 | 11.9 | 45.7 | 15.3 | 4.4 | 18.8 | 57.4 | 11.9 | 23.8 | 12.0 | 2.3 |
| | LDE (text) | 24.1 | 58.8 | 11.9 | 45.7 | 14.1 | 4.0 | **19.2** | 57.2 | 11.9 | 20.0 | 11.1 | 1.9 |
| | GDE (text) | **25.3** | **60.0** | **17.0** | **48.2** | **18.9** | **6.4** | 18.7 | 59.9 | **17.0** | 21.4 | 12.2 | 2.5 |
| | PDE (text) | 24.9 | 60.8 | **17.0** | 47.7 | 18.6 | 6.3 | 19.1 | **61.1** | **17.0** | **23.8** | **12.5** | **2.9** |
| | LDE (image) | 13.9 | 52.6 | 5.6 | 32.1 | 6.6 | 0.9 | 9.8 | 48.0 | 5.6 | 14.9 | 2.3 | 0.2 |
| | GDE (image) | **36.3** | 64.1 | 31.4 | 55.9 | 29.3 | 13.9 | 28.6 | 61.7 | 31.3 | 33.3 | 19.0 | 6.7 |
| | PDE (image) | 36.1 | **66.6** | **33.1** | **57.2** | **32.7** | **15.8** | **29.8** | **64.0** | **33.1** | **33.9** | **21.9** | **8.3** |
| MIT-States | CLIP | 33.0 | 52.1 | 30.6 | 45.3 | 26.3 | 11.1 | 15.6 | 47.7 | 30.6 | 8.3 | 8.4 | 1.7 |
| | LDE (text) | 30.6 | 51.2 | 24.7 | 43.0 | 21.9 | 8.2 | 21.1 | **50.7** | 24.7 | **13.8** | 11.9 | 2.5 |
| | GDE (text) | 32.6 | 51.7 | 27.8 | 45.2 | 24.5 | 10.0 | **21.3** | 49.9 | 27.8 | 13.0 | 12.1 | 2.6 |
| | PDE (text) | **32.8** | **52.2** | **30.2** | **45.9** | **26.9** | **11.4** | 17.4 | 49.2 | **30.2** | 10.2 | 10.1 | 2.1 |
| | LDE (image) | 15.3 | 30.5 | 15.0 | 20.9 | 11.1 | 2.0 | 11.0 | 34.8 | 15.0 | 5.6 | 4.6 | 0.4 |
| | GDE (image) | 28.1 | **45.3** | 30.7 | **36.1** | 23.4 | 8.6 | 18.5 | 43.6 | 29.7 | 8.5 | 9.3 | 1.8 |
| | PDE (image) | **28.7** | 45.1 | **38.5** | 35.4 | **26.2** | **10.7** | **19.7** | 43.6 | **38.5** | **8.9** | **10.1** | **2.3** |
| CGQA | CLIP | 15.2 | 30.2 | 8.5 | 25.7 | 8.9 | 1.6 | 4.7 | 24.4 | 8.4 | 2.3 | 2.5 | 0.1 |
| | LDE (text) | 10.4 | 30.2 | 5.8 | 23.7 | 6.4 | 0.9 | 3.8 | **27.7** | 5.8 | 1.92 | 1.66 | 0.06 |
| | GDE (text) | 13.4 | 30.3 | 7.5 | 25.4 | 8.0 | 1.3 | **5.2** | 26.2 | 7.4 | **2.4** | **2.7** | 0.12 |
| | PDE (text) | **15.0** | **31.2** | **8.4** | **27.0** | **9.1** | **1.6** | 4.9 | 27.4 | **8.3** | 2.2 | **2.7** | **0.14** |
| | LDE (image) | 1.6 | 16.0 | 0.7 | 7.1 | 0.9 | 0.03 | 0.04 | 0.02 | 0.00 | 0.00 | 0.00 | 0.00 |
| | GDE (image) | **13.4** | 30.3 | **7.5** | **25.4** | **8.0** | **1.3** | 5.2 | 26.2 | **7.4** | **2.4** | **2.7** | **0.12** |
| | PDE (image) | 11.0 | **31.6** | 7.4 | 16.3 | 5.9 | 0.7 | **6.9** | **29.7** | 6.6 | 1.5 | 1.4 | 0.05 |

Table 9: Compositional classification results on the UT-Zappos and MIT-States datasets, in ViT-L/14. The highest values within each modality are in **bold**; The second-highest values are underlined. *best-of-22* denotes the best-performing hyperparameter setting selected from 22 randomly sampled configurations.

| Dataset | Method | Closed-World | | | | | | Open-World | | | | | |
|---|---|---|---|---|---|---|---|---|---|---|---|---|---|
| | | Attr | Obj | Seen | Unseen | HM | AUC | A | O | S | U | H | AUC |
| UT-Zappos | CLIP | 24.1 | 58.3 | 11.9 | 45.7 | 15.3 | 4.4 | 18.8 | 57.4 | 11.9 | 23.8 | 12.0 | 2.3 |
| | LDE (text) | 24.1 | 58.8 | 11.9 | 45.7 | 14.1 | 4.0 | **19.2** | 57.2 | 11.9 | 20.0 | 11.1 | 1.9 |
| | GDE (text) | 25.3 | 60.0 | **17.0** | 48.2 | **18.9** | **6.4** | 18.7 | 59.9 | **17.0** | 21.4 | 12.2 | 2.5 |
| | PDE (text) | 22.1 | 56.11 | 15.8 | 43.84 | 18.1 | 5.6 | 17.3 | 57.5 | 15.8 | **23.4** | 11.9 | 2.6 |
| | PDE (text, best-of-22) | **25.5** | **60.3** | 16.9 | **48.6** | 18.6 | **6.37** | 19.1 | **60.4** | 15.7 | 22.8 | **12.6** | **2.7** |
| | LDE (image) | 13.9 | 52.6 | 5.6 | 32.1 | 6.6 | 0.9 | 9.8 | 48.0 | 5.6 | 14.9 | 2.3 | 0.2 |
| | GDE (image) | 36.3 | 64.1 | 31.4 | 55.9 | 29.3 | 13.9 | 28.6 | 61.7 | 31.3 | **33.3** | 19.0 | 6.7 |
| | PDE (image) | **36.8** | **66.3** | 32.8 | **57.1** | **32.2** | **15.5** | 28.1 | **62.7** | **32.8** | 32.7 | **21.8** | **7.9** |
| | PDE (image, best-of-22) | 36.6 | 64.5 | **33.0** | 56.0 | 30.6 | 14.7 | **29.8** | 63.2 | 32.0 | 32.7 | 21.1 | 7.7 |
| MIT-States | CLIP | 33.0 | 52.1 | 30.6 | 45.3 | 26.3 | 11.1 | 15.6 | 47.7 | 30.6 | 8.3 | 8.4 | 1.7 |
| | LDE (text) | 30.6 | 51.2 | 24.7 | 43.0 | 21.9 | 8.2 | 21.1 | **50.7** | 24.7 | **13.8** | 11.9 | 2.5 |
| | GDE (text) | 32.6 | 51.7 | 27.8 | 45.2 | 24.5 | 10.0 | **21.3** | 49.9 | 27.8 | 13.0 | 12.1 | 2.6 |
| | PDE (text) | 31.6 | 51.5 | 29.2 | 45.0 | 26.2 | 10.7 | 14.8 | 48.8 | 29.2 | 8.4 | 8.6 | 1.7 |
| | PDE (text, best-of-22) | **34.1** | **52.3** | 31.0 | **46.3** | 27.2 | **11.7** | 21.2 | 50.1 | 29.1 | 13.2 | **12.3** | **2.8** |
| | LDE (image) | 15.3 | 30.5 | 15.0 | 20.9 | 11.1 | 2.0 | 11.0 | 34.8 | 15.0 | 5.6 | 4.6 | 0.4 |
| | GDE (image) | 28.1 | 45.3 | 30.7 | **36.1** | 23.4 | 8.6 | 18.5 | 43.6 | 29.7 | 8.5 | 9.3 | 1.8 |
| | PDE (image) | **30.0** | **45.6** | 40.5 | 35.0 | 26.9 | 11.1 | 19.1 | 43.8 | 40.5 | 8.1 | 9.5 | 2.2 |
| | PDE (image, best-of-22) | 28.8 | 45.3 | **41.3** | 36.0 | **27.2** | 11.5 | **20.4** | 43.8 | **41.3** | 9.0 | **10.5** | 2.5 |

Table 10: Primitive-wise Shapiro–Wilk statistics in the GDE space. For each primitive class, we compute the Shapiro–Wilk $p$-value for marginal normality in each GDE dimension. "Avg. dim. rejection ratio ($p \leq 0.05$)" reports the average (over primitives) fraction of dimensions whose $p$-value is at most 0.05. "Avg. $p$-value (mean/median)" report the averages (over primitives) of the mean and median Shapiro–Wilk $p$-values across dimensions.

| Dataset | Modality | Group-by | Avg. dim. rejection ratio ($p \leq 0.05$) | Avg. $p$-value (mean) | Avg. $p$-value (median) |
|---|---|---|---|---|---|
| MIT-States | image | attr | 0.110 | 0.449 | 0.422 |
| MIT-States | image | obj | 0.090 | 0.463 | 0.447 |
| UT-Zappos | image | attr | 0.069 | 0.469 | 0.451 |
| UT-Zappos | image | obj | 0.097 | 0.456 | 0.431 |
| MIT-States | text | attr | 0.153 | 0.487 | 0.479 |
| MIT-States | text | obj | 0.162 | 0.489 | 0.482 |
| UT-Zappos | text | attr | 0.061 | 0.494 | 0.486 |
| UT-Zappos | text | obj | 0.072 | 0.494 | 0.488 |

**Setup.** For each dataset (UT-Zappos, MIT-States), modality (image or text), and grouping mode (attribute or object), we first map CLIP embeddings into the context-centered tangent space used by GDE. To reduce within-class noise and give each semantic pair equal weight, we average image embeddings over identical attribute–object pairs and treat these averaged embeddings as the samples for that pair.

For each primitive class $c$ (attribute or object) and each GDE dimension $j$, we then collect the one-dimensional samples $\{x_{i,j} : i \in c\}$ and apply the Shapiro–Wilk test for marginal normality at significance level $\alpha = 0.05$. For each primitive, we compute a *rejection ratio*, defined as the fraction of dimensions whose Shapiro–Wilk $p$-value satisfies $p \leq 0.05$. We also record, for each primitive, the mean and median of the Shapiro–Wilk $p$-values across dimensions. Table 10 reports these quantities averaged over primitives for each dataset/modality/grouping.

**Interpretation.** Under exact multivariate normality, with $\alpha = 0.05$ we would expect about $5\%$ of the dimensions to yield $p \leq 0.05$ purely by chance. In our experiments, the observed rejection ratios in the text GDEs are only slightly above this level (around $5.7\%$–$7.0\%$ for UT-Zappos and MIT-States), and the corresponding mean/median $p$-values remain close to 0.5, suggesting that a

Gaussian approximation is statistically reasonable for these distributions. Overall, these results support the use of Gaussian ideal-word distributions in PDE, while acknowledging that higher-order departures from normality are present but moderate.

### B.8 WORDNET-BASED SEMANTIC-HIERARCHY ANALYSIS

To probe how interpretable PDE's covariance structure is, we analyze the relationship between the PDE carrier and an external notion of semantic specificity derived from WordNet(Miller, 1995). Concretely, we ask whether the position of the carrier along the attribute–object disagreement axis is consistent with whether the object is more general or more specific than the attribute.

**Setup.** We focus on the MIT-States dataset and fix an attribute $a$ (here *melted* and *huge*). For each attribute–object pair $(a, o)$ that appears in the dataset, we construct two scalar quantities.

First, we compute WordNet subtree sizes for the synsets corresponding to $a$ and $o$. Let $\mathrm{subtree}(c)$ denote the size of the WordNet subtree rooted at concept $c$, so that concepts with larger subtrees are more general and those with smaller subtrees are more specific. We define the semantic-hierarchy difference

$$\Delta_{\mathrm{subtree}} = -\log \mathrm{subtree}(o) - \big(-\log \mathrm{subtree}(a)\big), \tag{37}$$

which serves as the x-axis coordinate. Positive values of $\Delta_{\mathrm{subtree}}$ indicate that the object is more specific than the attribute, while negative values indicate the opposite.

Second, we compute the PDE MAP carrier $v^{(a,o)}$ for the pair $(a, o)$ and project it onto the disagreement axis between the attribute and object ideals, obtaining a scalar position $t \in [0, 1]$. Here $t = 0$ corresponds to the object mean, $t = 1$ corresponds to the attribute mean, and intermediate values indicate how strongly the composite is pulled towards the attribute versus the object. This position $t$ is used as the y-axis coordinate. For each attribute we also report the Pearson correlation between $\Delta_{\mathrm{subtree}}$ and $t$, as well as the fraction of pairs that fall on the "object side" of the axis ($t < 0.5$).

**Qualitative examples.** Figure 8 shows the resulting scatter plots for two MIT-States attributes, *melted* and *huge*. Each point corresponds to an attribute–object pair $(a, o)$ with x-axis $\Delta_{\mathrm{subtree}}$ and y-axis carrier position $t$; the dashed horizontal line marks the boundary between the object side ($t < 0.5$) and the attribute side ($t > 0.5$).

For *melted*, we obtain a negative correlation between $\Delta_{\mathrm{subtree}}$ and $t$ (approximately $-0.41$), and about $62\%$ of the points lie on the object side. This means that when the object becomes more specific than the attribute (moving to the right on the x-axis), the carrier tends to move towards the object. A similar but stronger pattern appears for the very broad attribute *huge*: the correlation is around $-0.50$, and roughly $93\%$ of the pairs satisfy $t < 0.5$. In this case PDE almost always keeps the composite close to the object, which matches the intuition that a generic size attribute should modulate the object representation without overriding its identity.

These examples illustrate that PDE's precision-weighted carriers induce interpretable dominance relationships in the decomposed latent space: more specific concepts tend to "win" along the disagreement axis, in a manner that aligns with an external semantic hierarchy.

### B.9 VISUAL SPATIAL REASONING WITH THREE PRIMITIVES

To test whether PDE extends beyond attribute–object pairs, we evaluate it on a relational composition task derived from the Visual Spatial Reasoning (VSR) Liu et al. (2023) benchmark. This experiment uses *three* primitives per example and therefore instantiates the general $k$-way MAP carrier described in Appendix C.3.

**Setup.** Each positive triple in VSR consists of a caption and a spatial relation label (e.g., "The dog is on the person."). We parse this annotation into three textual primitives $(\mathrm{obj}_1, \mathrm{rel}, \mathrm{obj}_2)$ (e.g., *dog*, *on*, *person*) and encode each primitive with the CLIP text encoder. Following our main compositional classification experiments (Sec. 4), we use the pretrained SigLIP (ViT-SO400M/14) backbone and keep all PDE hyperparameters at their default values, without any additional tuning for VSR. For every primitive we estimate an ideal-word Gaussian, and PDE additionally computes a precision

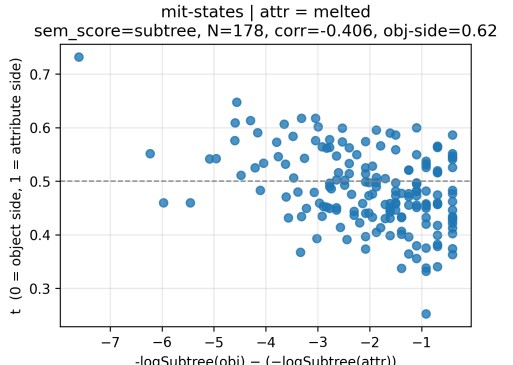
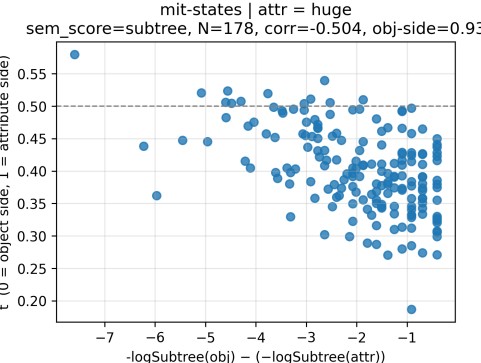

Figure 8: WordNet-based semantic-hierarchy analysis on MIT-States for two attributes, *melted* (left) and *huge* (right). Each point corresponds to an attribute–object pair $(a, o)$. The x-axis shows the semantic-hierarchy difference $\Delta_{\text{subtree}} = -\log \text{subtree}(o) - (-\log \text{subtree}(a))$, and the y-axis shows the PDE carrier position $t \in [0, 1]$ along the disagreement axis (0 = object side, 1 = attribute side). The dashed horizontal line marks $t = 0.5$. For both attributes we observe a negative correlation between $\Delta_{\text{subtree}}$ and $t$, and a majority of pairs lie on the object side, indicating that more specific objects tend to pull the composite closer to the object.

| Model | O1 | R | O2 | S | U | H | AUC |
|-------|------|------|------|-----|------|-----|-----|
| GDE | 39.0 | 9.1 | 37.6 | 4.9 | 21.5 | 5.6 | 0.7 |
| PDE | 43.6 | 12.2 | 37.3 | 8.1 | 28.3 | 8.8 | 1.6 |

Table 11: Three-primitive composition on the Visual Spatial Reasoning (VSR) benchmark. O1/R/O2 are top-1 accuracies for predicting the first object, relation, and second object, respectively. S/U/H denote triple accuracy on seen and unseen compositions and their harmonic mean. AUC is the area under the seen–unseen accuracy curve induced by sweeping a global unseen bias. PDE consistently improves over GDE on this relational, three-primitive task.

matrix for each class. Given a triple $(\text{obj}_1, \text{rel}, \text{obj}_2)$, we compose its three primitive distributions using the $k$-way PDE carrier

$$v^*_{(\text{obj}_1, \text{rel}, \text{obj}_2)} = \left( \sum_i P_{p_i} + \lambda I \right)^{-1} \left( \sum_i P_{p_i} m_{p_i} \right),$$

where the sum runs over $p_1 = \text{obj}_1$, $p_2 = \text{rel}$, and $p_3 = \text{obj}_2$. The resulting composite embedding is used in the same way as in our attribute–object experiments: we score images according to cosine similarity with the triple carrier and evaluate under closed- and open-world protocols adapted to triples.

We report accuracies for predicting the first object (O1), relation (R), and second object (O2), as well as triple-level accuracy on seen (S) and unseen (U) compositions and their harmonic mean (H). As in the main experiments, AUC denotes the area under the seen–unseen accuracy curve obtained by sweeping a global bias on unseen logits.

**Interpretation.** Despite being tuned for attribute–object composition, PDE also yields gains on this spatial-relational benchmark, improving O1 and R accuracy, unseen triple accuracy U, and AUC over GDE while maintaining comparable O2 performance. This suggests that modeling primitives as Gaussians and composing them via the MAP carrier provides a generally useful probabilistic layer for multi-primitive composition, not only for attribute–object pairs but also for more complex relational structures.

Table 12: Covariance traces of the attribute and object primitives in CLIP text and image space. For each dataset, we report $\text{Tr}(\boldsymbol{\Sigma}_{\text{attr}})$ and $\text{Tr}(\boldsymbol{\Sigma}_{\text{obj}})$ in both modalities, along with the object-to-attribute trace ratio $r_{\text{obj/attr}} = \text{Tr}(\boldsymbol{\Sigma}_{\text{obj}})/\text{Tr}(\boldsymbol{\Sigma}_{\text{attr}})$. These statistics reveal a systematic imbalance between attribute and object dispersion that differs across text and image space, which in turn affects the effective precision weighting used by PDE.

| | Text space | | | Image space | | |
|---|---|---|---|---|---|---|
| Dataset | Attribute trace | Object trace | Obj/Attr ratio | Attribute trace | Object trace | Obj/Attr ratio |
| UT-Zappos | 0.242 | 0.235 | 0.971 | 0.116 | 0.074 | 0.638 |
| MIT-States | 0.323 | 0.240 | 0.743 | 0.222 | 0.142 | 0.640 |
| C-GQA | 0.349 | 0.271 | 0.779 | 0.438 | 0.318 | 0.726 |

## B.10 COVARIANCE STATISTICS IN TEXT VS IMAGE SPACE

To better understand how attribute and object primitives are distributed in the CLIP embedding spaces, we summarize their covariance structure in the text and image branches. For each dataset and modality, we reuse the primitive Gaussian covariances $\hat{\boldsymbol{\Sigma}}_{\text{attr}}$ and $\hat{\boldsymbol{\Sigma}}_{\text{obj}}$ defined in Appendix A.1.

To obtain a scalar summary of each primitive's dispersion, we compute the trace of its covariance matrix, i.e., the sum of its diagonal entries:

$$\text{Tr}(\hat{\boldsymbol{\Sigma}}) = \sum_{d=1}^{D} \hat{\boldsymbol{\Sigma}}_{dd}, \tag{38}$$

where $D$ is the embedding dimension. We report $\text{Tr}(\hat{\boldsymbol{\Sigma}}_{\text{attr}})$ and $\text{Tr}(\hat{\boldsymbol{\Sigma}}_{\text{obj}})$ separately for attributes and objects in text and image space.

In addition to the raw traces, we also consider the object-to-attribute trace ratio

$$r_{\text{obj/attr}} = \frac{\text{Tr}(\hat{\boldsymbol{\Sigma}}_{\text{obj}})}{\text{Tr}(\hat{\boldsymbol{\Sigma}}_{\text{attr}})}, \tag{39}$$

computed separately in text and image space. Values $r_{\text{obj/attr}} < 1$ indicate that the object primitive is more concentrated (higher precision) than the attribute primitive, whereas $r_{\text{obj/attr}} > 1$ indicates the opposite.

The resulting statistics for UT-Zappos, MIT-States and C-GQA are summarized in Table 12. In all datasets and in both modalities, the object primitives exhibit a smaller trace than the attribute primitives, implying that the composite representation is slightly biased toward the object prototype. Moreover, the object-to-attribute trace ratio is consistently smaller in the image space than in the text space, indicating that this bias toward the object primitive is stronger when distributions are instantiated in the visual embedding space. Note that The trace values in Table 12 serve as a simple and concise diagnostic of the relative pliability of attribute versus object primitives in the two modalities, as it ignores the full anisotropic covariance. Importantly, the model does not impose an arbitrary dependence on modality; instead, it automatically adapts to the geometry of the underlying text and image embedding spaces.

In the image modality, these traces increase monotonically from UT-Zappos to MIT-States to C-GQA, reflecting that C-GQA has noisier and more ambiguous labels (each image often containing multiple objects). Yet PDE still outperforms GDE on C-GQA (closed world setting, SIGLIP1, SIGLIP2), suggesting that modeling per-concept covariance still brings the composed embeddings closer, on average, to the intended to "ideal composite" than GDE's uncertainty-unaware construction.

## B.11 EFFECT OF PDE ON STRONGER BASE MODELS

To summarize how the performance gap between PDE and GDE evolves with backbone strength, we compute for each dataset, modality, evaluation setup, and backbone the AUC difference

$$\Delta_{\text{PDE–GDE}} = \text{AUC(PDE)} - \text{AUC(GDE)},$$

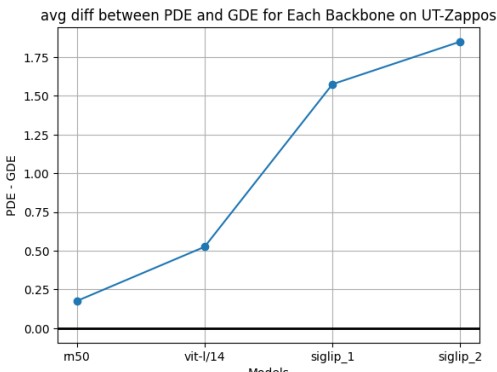 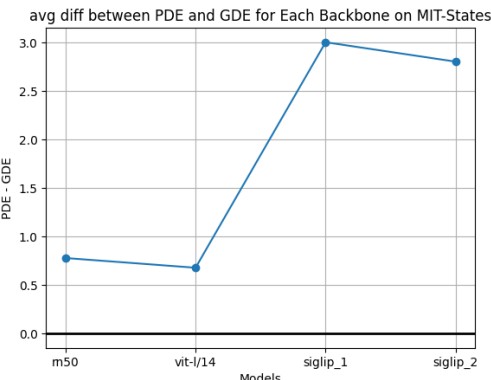

Figure 9: Backbone-wise performance margins of PDE over GDE. For UT-Zappos (left) and MIT-States (right), and for each backbone(RN50 (CLIP), ViT-L/14 (CLIP), SIGLIP1, SIGLIP2), we plot the mean AUC difference $\Delta_{\text{PDE–GDE}} = \text{AUC(PDE)} - \text{AUC(GDE)}$ in compositional classification, averaged over the four dataset configurations (text/image $\times$ closed/open world) and based on the results reported in Tables 3–5. The margins tend to grow with backbone strength, showing that PDE can better leverage stronger vision–language encoders than GDE.

using the same closed- and open-world AUC metrics as in Tables 3–5. For the visualization in Figure 9, we focus on UT-Zappos and MIT-States and, for each backbone, average $\Delta_{\text{PDE–GDE}}$ over the four dataset configurations (text/image $\times$ closed/open world). Thus, each point in the plots corresponds to the mean AUC margin across these four settings for a given dataset and backbone.

We observe that the PDE–GDE margin generally increases as we move from RN50 (CLIP) to ViT-L/14 (CLIP) and further to SigLIP and SigLIP2, meaning that PDE yields larger gains when the underlying representation space is better trained. This suggests that stronger baselines induce representation spaces whose compositional structure is more clearly captured by PDE as a compositional embedding.

## C  MATHEMATICAL DERIVATION

**Setup.**  Let $m_a, m_o \in \mathbb{R}^d$ be primitive means with covariances $\Sigma_a, \Sigma_o \succ 0$ and precisions $P_p := \Sigma_p^{-1}$. Given a ridge $\lambda \geq 0$, define

$$A := P_a + P_o + \lambda I, \qquad v^\star = A^{-1}(P_a m_a + P_o m_o), \qquad d := m_a - m_o. \tag{40}$$

### C.1  MAP IN THE POSTERIOR METRIC

Introduce the $A$-inner product $\langle x, y \rangle_A := x^\top A y$ and the whitening

$$\bar{v} := A^{1/2} v, \quad \bar{m}_p := A^{1/2} m_p, \quad \tilde{S} := A^{-1/2} P_a A^{-1/2}.$$

Then $\tilde{S}$ is symmetric with spectrum in $(0, 1)$ and the whitened MAP takes the *translated mode-wise convex* form

$$\boxed{\bar{v}^\star = \tilde{S} \bar{m}_a + (I - \tilde{S}) \bar{m}_o \ - \ \lambda A^{-1/2} m_o.} \tag{41}$$

Equivalently, let $\gamma := \lambda A^{-1/2} m_o$ and $\bar{m}_p^\circ := \bar{m}_p - \gamma$, we have

$$\bar{v}^\star = \tilde{S} \bar{m}_a^\circ + (I - \tilde{S}) \bar{m}_o^\circ, \qquad 0 \prec \tilde{S} \prec I.$$

Thus, in any eigenbasis of $\tilde{S}$, the MAP is a coordinate-wise convex average of the *translated* endpoints.

## C.2 A WHITENING-FREE SCALAR THAT GUARANTEES TILT ALONG THE DISAGREEMENT AXIS

We certify on which side of the midpoint the MAP lies *along the line through $m_o$ and $m_a$* by a single scalar computed in the original coordinates. Let $c_{eq} := \frac{1}{2}(m_a + m_o)$ and define

$$s_E := d^\top(v^\star - c_{eq}) = d^\top A^{-1} P_a d - \tfrac{1}{2}\|d\|^2 - \lambda d^\top A^{-1} m_o. \qquad (42)$$

Let $t$ be the orthogonal projection coefficient of $v^\star$ onto the line through $m_o$ in direction $d$:

$$t := \frac{d^\top(v^\star - m_o)}{\|d\|^2} = \tfrac{1}{2} + \frac{s_E}{\|d\|^2}. \qquad (43)$$

Then, *necessarily and sufficiently*,

$$s_E > 0 \iff t > \tfrac{1}{2} \quad (a\text{-side}), \qquad s_E < 0 \iff t < \tfrac{1}{2} \quad (o\text{-side}), \qquad s_E = 0 \iff t = \tfrac{1}{2} \quad (\text{midpoint}). \qquad (44)$$

**Proof of equation 44.** Write $c_{eq} = m_o + \frac{1}{2}d$ with $d = m_a - m_o$. By definition,

$$s_E = d^\top(v^\star - c_{eq}) = d^\top(v^\star - m_o - \tfrac{1}{2}d) = d^\top(v^\star - m_o) - \tfrac{1}{2}\|d\|^2.$$

By equation 43, the orthogonal projection coefficient of $v^\star$ onto the line through $m_o$ in direction $d$ is

$$t = \frac{d^\top(v^\star - m_o)}{\|d\|^2}.$$

Combining the two displays yields the identity

$$s_E = \|d\|^2\Big(t - \tfrac{1}{2}\Big).$$

If $d \neq 0$, then $\|d\|^2 > 0$, hence

$$s_E > 0 \iff t > \tfrac{1}{2}, \qquad s_E < 0 \iff t < \tfrac{1}{2}, \qquad s_E = 0 \iff t = \tfrac{1}{2},$$

which proves equation 44. □

*Consistency check.* Using $v^\star = A^{-1}(P_a m_a + P_o m_o)$ and $m_a = m_o + d$,

$$v^\star - m_o = A^{-1}(P_a(m_o + d) + P_o m_o) - m_o = A^{-1} P_a d - \lambda A^{-1} m_o,$$

so

$$s_E = d^\top(A^{-1} P_a d - \lambda A^{-1} m_o) - \tfrac{1}{2}\|d\|^2 = d^\top A^{-1} P_a d - \tfrac{1}{2}\|d\|^2 - \lambda d^\top A^{-1} m_o,$$

which matches equation 42.

## C.3 MAP ENCODES COMPOSITIONAL PLIABILITY (MULTI-DIMENSIONAL CASE)

**Pliability and stiffness.** For any unit direction $w$, define directional pliability $\pi_p(w) := w^\top \Sigma_p w$ and stiffness $\kappa_p(w) := w^\top P_p w = 1/\pi_p(w)$.

**Direction-wise inverse-variance rule (scope).** If one *first* projects the objective onto a fixed unit $w$ (reducing to a 1D problem) and solves that 1D MAP, the solution along $w$ is the inverse-variance average

$$\langle v_w^\star, w \rangle = \alpha(w)\langle m_a, w \rangle + (1 - \alpha(w))\langle m_o, w \rangle, \qquad \alpha(w) := \frac{w^\top P_a w}{w^\top(P_a + P_o + \lambda I)w} \in (0, 1). \qquad (45)$$

By contrast, the *post hoc* projection of the full $d$-dimensional MAP satisfies

$$\langle v^\star, w \rangle = w^\top A^{-1} P_a m_a + w^\top A^{-1} P_o m_o,$$

and equation 45 need *not* hold unless the problem decouples along $w$ (e.g., commuting $P_a, P_o$ with $m_a, m_o$ mass concentrated on common eigen-directions).

**Tilt relative to equal-weight composition.** Let $\bar{c}_{\text{eq}} := \frac{1}{2}(\bar{m}_a + \bar{m}_o)$ and $\bar{d} := \bar{m}_a - \bar{m}_o$. Then

$$\boxed{\bar{v}^\star - \bar{c}_{\text{eq}} = \left(\tilde{S} - \tfrac{1}{2}I\right)\bar{d} \, - \, \lambda\, A^{-1/2}m_o.} \tag{46}$$

For $\lambda = 0$, the Rayleigh-quotient criterion recovers the familiar form

$$\text{sign}\left(\left\langle \bar{v}^\star - \bar{c}_{\text{eq}}, \widehat{\bar{d}} \right\rangle\right) = \text{sign}\left(\frac{\bar{d}^\top \tilde{S}\, \bar{d}}{\|\bar{d}\|^2} - \tfrac{1}{2}\right),$$

and, under commuting $P_a, P_o$ with $\bar{d}$ aligned to a common eigenvector,

$$\frac{\bar{d}^\top \tilde{S}\, \bar{d}}{\|\bar{d}\|^2} = \frac{\kappa_a(w)}{\kappa_a(w) + \kappa_o(w) + \lambda},$$

which tilts toward the stiffer partner along $w$.

**Takeaway.** MAP composition *encodes pliability*: directions with higher variance (lower stiffness) concede more in the composition. The whitening-free scalar $s_E$ in equation 42 provides an exact, assumption-free certificate of tilt along the disagreement axis, while the $A$-whitened view equation 41–equation 46 reveals the translated mode-wise convex structure that underlies the same dominance logic.

C.4   EXTENSION TO $k$-WAY COMPOSITION

In the main text we derive the PDE carrier for the composition of two primitives (attribute and object). Here we show that the same derivation extends directly to an arbitrary number $k \geq 2$ of primitives.

**Generative view.** Let $\{p_1, \ldots, p_k\}$ be a set of primitives to be composed, and let each primitive $p_i$ be modeled as a Gaussian in the decomposed CLIP space,

$$z \mid p_i \sim \mathcal{N}(m_{p_i}, P_{p_i}^{-1}), \tag{47}$$

where $m_{p_i} \in \mathbb{R}^d$ is the ideal-word mean and $P_{p_i} \succ 0$ is the precision (inverse covariance) associated with primitive $p_i$. As in the two-way case, we place an isotropic Gaussian prior on the composite representation,

$$z \sim \mathcal{N}(0, (\lambda I)^{-1}), \tag{48}$$

with $\lambda > 0$ controlling the strength of the prior.

Assuming conditional independence of the primitive-specific views given $z$, the (unnormalized) posterior over $z$ given the set of primitives is

$$p(z \mid p_1, \ldots, p_k) \propto \left[\prod_{i=1}^{k} \mathcal{N}(z; m_{p_i}, P_{p_i}^{-1})\right] \mathcal{N}(z; 0, (\lambda I)^{-1}). \tag{49}$$

Taking the negative log of this expression and discarding constants yields the quadratic objective

$$\mathcal{E}(z) = \tfrac{1}{2}\sum_{i=1}^{k}(z - m_{p_i})^\top P_{p_i}(z - m_{p_i}) + \tfrac{\lambda}{2}\, z^\top z. \tag{50}$$

**MAP carrier.** Minimizing $\mathcal{E}(z)$ with respect to $z$ gives the MAP carrier for the $k$-way composition. Taking the gradient and setting it to zero,

$$\nabla_z \mathcal{E}(z) = \left(\sum_{i=1}^{k} P_{p_i} + \lambda I\right)z - \sum_{i=1}^{k} P_{p_i} m_{p_i} \; = \; 0, \tag{51}$$

so the optimal carrier $v^*_{(p_1,\ldots,p_k)}$ is

$$v^*_{(p_1,\ldots,p_k)} = \left(\sum_{i=1}^{k} P_{p_i} + \lambda I\right)^{-1}\left(\sum_{i=1}^{k} P_{p_i} m_{p_i}\right). \tag{52}$$

Thus, the composite representation is a precision-weighted average of the primitive means, shrunk towards the origin by the isotropic prior.

When $k = 2$ and $(p_1, p_2) = (a, o)$ correspond to an attribute and an object, respectively, Eq. equation 52 reduces exactly to the two-way PDE carrier used in the main text. In our VSR experiments with triples $(\mathrm{obj}_1, \mathrm{rel}, \mathrm{obj}_2)$ (Appendix B.9), we instantiate Eq. equation 52 with $k = 3$ and $(p_1, p_2, p_3) = (\mathrm{obj}_1, \mathrm{rel}, \mathrm{obj}_2)$.

## C.5 PDE REDUCES TO LDE/GDE UNDER ISOTROPIC SHARED COVARIANCE

We briefly show that LDE/GDE arises as a special case of PDE when all primitives share a common isotropic covariance and no additional shrinkage is applied.

Let $\mathcal{P} = \{p_1, \ldots, p_k\}$ be a finite set of primitives with "ideal-word" Gaussians

$$p_i \sim \mathcal{N}(m_{p_i}, \Sigma_{p_i}), \qquad P_{p_i} := \Sigma_{p_i}^{-1}, \tag{53}$$

where $m_{p_i} \in \mathbb{R}^d$ (or a tangent space $T_\mu M$ in the geometric case) and $P_{p_i}$ is the precision. The PDE MAP carrier over $\mathcal{P}$ is

$$c_{\mathcal{P}}^{\mathrm{map}} := \Big(\sum_{i=1}^{k} P_{p_i} + \lambda I\Big)^{-1} \Big(\sum_{i=1}^{k} P_{p_i} m_{p_i}\Big). \tag{54}$$

**Isotropic shared covariance.** Assume that all primitives share a common isotropic covariance,

$$\Sigma_{p_i} = \sigma^2 I \quad \text{for all } i \in \{1, \ldots, k\}. \tag{55}$$

Then each precision is

$$P_{p_i} = \Sigma_{p_i}^{-1} = \frac{1}{\sigma^2} I =: \alpha I, \tag{56}$$

so that

$$\sum_{i=1}^{k} P_{p_i} = \sum_{i=1}^{k} \alpha I = k\alpha I. \tag{57}$$

Substituting into equation 54 gives

$$c_{\mathcal{P}}^{\mathrm{map}} = (k\alpha I + \lambda I)^{-1} \Big(\sum_{i=1}^{k} \alpha I \, m_{p_i}\Big) \tag{58}$$

$$= \frac{1}{k\alpha + \lambda} I \cdot \Big(\alpha \sum_{i=1}^{k} m_{p_i}\Big) \tag{59}$$

$$= \frac{\alpha}{k\alpha + \lambda} \sum_{i=1}^{k} m_{p_i}. \tag{60}$$

Define the equal-weight mean of primitive means

$$\bar{m}_{\mathcal{P}} := \frac{1}{k} \sum_{i=1}^{k} m_{p_i}. \tag{61}$$

Then we can rewrite

$$c_{\mathcal{P}}^{\mathrm{map}} = \frac{\alpha}{k\alpha + \lambda} \cdot k\bar{m}_{\mathcal{P}} = \frac{k}{k + \lambda\sigma^2} \, \bar{m}_{\mathcal{P}}. \tag{62}$$

**LDE/GDE limit.** In the idealized case with no shrinkage ($\lambda = 0$), equation 62 reduces to

$$c_{\mathcal{P}}^{\mathrm{map}} = \bar{m}_{\mathcal{P}} = \frac{1}{k} \sum_{i=1}^{k} m_{p_i} =: c_{\mathcal{P}}^{\mathrm{eq}}, \tag{63}$$

which is exactly the equal-weight carrier used in LDE/GDE. The corresponding LDE/GDE composite can be written as

$$v_{\mathcal{P}}^{\text{LDE/GDE}} = \sum_{i=1}^{k} m_{p_i} = k\, c_{\mathcal{P}}^{\text{eq}}, \tag{64}$$

while the PDE composite uses the count-based scale $s = |\mathcal{P}| = k$,

$$v_{\mathcal{P}}^{\text{PDE}} = s\, c_{\mathcal{P}}^{\text{map}} = k\, c_{\mathcal{P}}^{\text{map}}. \tag{65}$$

Therefore, under the isotropic shared covariance assumption and $\lambda = 0$,

$$v_{\mathcal{P}}^{\text{PDE}} = k\, c_{\mathcal{P}}^{\text{map}} = k\, c_{\mathcal{P}}^{\text{eq}} = v_{\mathcal{P}}^{\text{LDE/GDE}}. \tag{66}$$

This shows that the LDE/GDE carrier, and hence the LDE/GDE composite, can be viewed as a special case of the PDE carrier and composite in this isotropic limit.

## D  USE OF LARGE LANGUAGE MODELS

Large language models are used to refine writing of this paper. LaTeX form of tables and mathematical equations are provided by LLMs with natural language guide given by the authors of the paper. Also, related works are found using LLMs.

