# OpenReview forum: "Probabilistic Decomposable Embeddings: Uncertainty-Aware Composition in Vision-Language Models"
_ICLR.cc/2026/Conference — Submitted to ICLR 2026_

### Official Review · Reviewer_iJMx · 2025-10-29

**Soundness:** 2
**Presentation:** 2
**Contribution:** 1
**Rating:** 2
**Confidence:** 3

**Summary:**

The goal of the paper is to decompose CLIP embeddings of composite concepts (e.g., "red apple") into embeddings corresponding to their primitives ("red" and "apple").
The idea is that these primitive embeddings (referred to as *ideal words*) can be recomposed to embed unseen combinations (e.g., "blue apple").
The paper starts from the work of Trager et al. (2023) and Berasi et al. (2025), which propose linear and geodesic decompositions, respectively, and offers a probabilistic extension that models ideal words as Gaussian distributions, instead of single point estimates.
This results in a method that takes into account the covariance of the data distribution;
this encodes what the authors call _compositional pliability_ (primitives with higher variance yield more during composition, while lower-variance primitives dominate).
The paper evaluates this method in the context of compositional zero-shot classification on two standard datasets (UT-Zappos and MIT-States) and shows that it is generally better than the two starting baselines as well as the original CLIP.

Justification for the rating: The paper does a good job at extending the work of Trager et al. (2023) and Berasi et al. (2025), but the contribution is limited to this specific research thread and fails to engage with the very rich literature on compositional zero-shot learning.

**Strengths:**

- The paper proposes a principled probabilistic extension to existing work on embeddings decomposability.
- The method shows good gains compared to the baselines.

**Weaknesses:**

- The literature on compositional zero-shot learning (CZSL) is very rich (see below for some references). The authors fail to acknowledge any of these or other related works, and do not compare their approach to any of them neither qualitatively, nor quantitatively.
- The paper doesn't discuss the computational cost of the proposed method. Since it relies on covariances matrices and their inverses, this is certainly slower than the baselines (LDE and GDE). But I assume the computational cost can be amortized by storing the precision matrices for all attributes and objects.

Some references on CZSL:
- Bao, Wentao, et al. "Prompting language-informed distribution for compositional zero-shot learning." _ECCV_, 2024.
- Huang, Siteng, et al. "Troika: Multi-path cross-modal traction for compositional zero-shot learning." _CVPR_, 2024.
- Li, Lin, et al. "Compositional zero-shot learning via progressive language-based observations." arXiv preprint arXiv:2311.14749 (2023).
- Mancini, Massimiliano, et al. "Learning graph embeddings for open world compositional zero-shot learning." _TPAMI_ (2022).
- Qu, Hongyu, et al. "Learning clustering-based prototypes for compositional zero-shot learning." _ICLR_, 2025.
- Saini, Nirat, Khoi Pham, and Abhinav Shrivastava. "Disentangling visual embeddings for attributes and objects." _CVPR_, 2024.
- Zhou, Kaiyang, et al. "Learning to prompt for vision-language models." _IJCV_ (2022).

**Questions:**

- How does the proposed uncertainty-aware composition compare to other compositional mechanisms (such as those listed in weaknesses section)?
- In Table 1, why do Closed-World+Seen and Open-World+Seen results are (almost) always the same? Shouldn't the Closed-World setting be easier? Are the results correct?
- In Table 1, why are there multiple values underlined in some of the settings (e.g., UT-Zappos, Attr and AUC, text)?
- In Table 1, can the authors elaborate why the image-based composition sometimes works better than the text-based one (UT-Zappos, Closed-World setting)? Is this due to the specifity of the data?
- In the caption of Table 5, what does s = "all" (number of composing primitives) and s = "median" (dataset-wise median of composing set sizes) mean? Isn't the number of composing primitives always two, since the datasets have attributes and objects?

---

> ### Author Response · Authors · 2025-11-21
> **Response to W1 (CZSL literature and related work).**
>
> We thank the reviewer for pointing us to this line of work and agree that the compositional zero-shot learning (CZSL) literature is rich and highly relevant. We were in fact aware that UT-Zappos and MIT-States are standard CZSL benchmarks, and we briefly mentioned this connection when introducing the datasets, but we did not explicitly position PDE as a CZSL method. The main reason is that, unlike most CZSL approaches, our work does not train a new classifier or fine-tune the backbone; instead, PDE is a post-hoc probabilistic fusion rule that operates directly in the latent space of a fixed pre-trained VLM (e.g., CLIP), with no additional supervision beyond what LDE/GDE already use. In early drafts we found that trying to cover the full CZSL literature risked broadening the scope too much and obscuring our primary contribution.
>
> That said, we agree that we should have acknowledged CZSL more explicitly and clarified how PDE relates to it. In the revised version, we added (i) a dedicated paragraph in the related-work section summarizing recent CZSL methods, including the papers highlighted by the reviewer, and (ii) made clear that PDE is complementary to these approaches: it provides a probabilistic, uncertainty-aware composition rule in the decomposed VLM latent space that could in principle be plugged into many CZSL pipelines.
>
> Concretely, we added in the revised version of our paper, of how recent CZSL works extend or adapt VLMs for compositional recognition.
>
> Bao et al. (ECCV 2024) use LLMs to build language-informed class distributions on top of CLIP. Huang et al. (CVPR 2024) design a multi-path prompt architecture with separate attribute/object/composition prompts. Li et al. (arXiv 2023) introduce progressive language-based observations over primitives. Mancini et al. (TPAMI 2022) learn graph embeddings over state–object–composition nodes for open-world CZSL. Qu et al. (ICLR 2025) learn clustering-based prototypes for primitives. Saini et al. (CVPR 2024) disentangle visual attribute/object embeddings. Zhou et al. (IJCV 2022) learn continuous prompts for VLMs.

---

> ### Author Response · Authors · 2025-11-21
> **Response to W2 (computational analysis and efficiency).**
>
> Special thanks for raising this point. For concrete timing numbers and complexity analysis, please see our third response to reviewer j85S (W3).

---

> ### Author Response · Authors · 2025-11-21
> **Response to Q1 (comparison to other compositional mechanisms).**
>
> The key difference is that PDE is a post-hoc, uncertainty-aware fusion rule operating on a fixed representation, whereas many recent compositional mechanisms are learning-based: they train prompts, graphs, prototypes, or new encoders specifically for CZSL. PDE assumes we already have attribute and object embeddings (e.g., from CLIP or a CZSL model) and asks how to combine them in a way that accounts for their empirical uncertainty.
>
> Concretely, PDE represents each primitive concept as a Gaussian with mean and covariance estimated from data, and defines the composite as a closed-form MAP fusion. This has two practical consequences:
>
> - It improves compositional classification in both text and image settings without any additional labels or end-to-end training (notably in the zero-shot text case).
> - It exposes an interpretable covariance/pliability structure, indicating which concept is more uncertain and thus more easily “bent” by its partner during composition.
>
> We see PDE as complementary to other compositional mechanisms: it can be applied on top of the embeddings produced by those methods, and our experiments show that even without retraining, this uncertainty-aware fusion yields consistent gains over deterministic mean-based baselines such as LDE and GDE.

---

> ### Author Response · Authors · 2025-11-21
> **Response to Q2 (Closed-World vs Open-World Seen accuracy in Table 1) and Q3 (multiple underlined values in Table 1).**
>
> **Response to Q2 (Closed-World vs Open-World Seen accuracy in Table 1).**
>
> Yes, the results are correct, and we agree this point needs clarification.
>
> We follow the generalized CZSL protocol used in prior work (GDE). For each test image, we predict an attribute–object pair among a candidate label set:
>
> - In the closed-world setting, the candidate labels are restricted to the attribute–object pairs that actually occur in the dataset splits (i.e., training or test pairs).
> - In the open-world setting, the candidate labels are all valid attribute–object combinations in the dataset (a strictly larger superset of the above).
>
> The Seen accuracy is always computed on the subset of test images whose ground-truth pair is in the seen set (i.e., in the training pairs). For these examples, the only way in which the open-world setting can reduce Seen accuracy, when the underlying model is fixed, is if one of the additional candidate pairs (present only in the open-world label set) outranks the correct seen pair. Empirically this happens extremely rarely for our models, so the top-1 predictions on seen examples are almost always identical in the two regimes, and the Seen numbers coincide up to minor rounding differences.
>
> In addition, in our tables each method (for LDE/GDE and not PDE as PDE doesn’t use validation dataset for hyperparameter search) is hyperparameter-tuned independently for the closed- and open-world regimes using validation HM/AUC. As a result, small discrepancies in Seen between the two regimes can also arise simply because the selected hyperparameters differ slightly, even though the Seen metric itself is defined on the same subset of test examples.
>
> By contrast, the Unseen and HM/AUC metrics are much more sensitive to the size of the candidate set and to the calibration sweep between seen and unseen pairs. Consequently, the impact of closed- vs. open-world evaluation is mainly reflected in Unseen and HM/AUC, while Seen remains (almost) unchanged. We will expand Sec. 4.2 to explicitly describe the candidate label sets and to explain why the Seen scores can be (almost) identical across the two regimes.
>
>
> **Response to Q3 (multiple underlined values in Table 1).**
>
> Special thanks for catching this. This is purely a formatting mistake in the current draft. Only the best and second-best values per column should be highlighted, but our table inadvertently underlined multiple entries in some settings. We corrected this in the revised version so that at most one value per column is underlined.

---

> ### Author Response · Authors · 2025-11-21
> **Response to Q4 (image- vs text-based composition on UT-Zappos).**
>
> We believe this behavior is largely due to dataset and modality specific factors.
>
> UT-Zappos is a fine-grained shoe dataset whose attributes (e.g., materials, closures, shapes) are very localized and visually salient, while the corresponding textual descriptions can be subtle or ambiguous. In this regime, CLIP’s image encoder tends to produce more separable attribute clusters than the text encoder, so composing directly in the image space can be slightly easier in the closed-world classification setting.
>
> In contrast, on MIT-States many attributes are more abstract linguistic “states” (e.g., rusty, old, damaged), for which the text modality is relatively stronger; indeed, Table 1 shows that text-based composition decisively outperforms image-based composition on MIT-States.
>
> Importantly, PDE itself is modality-agnostic: it operates on whichever embedding space we choose (text or image) and uses the estimated covariances to perform precision-weighted MAP fusion. Thus PDE improves over LDE/GDE in both modalities; the fact that image-based PDE slightly surpasses text-based PDE on UT-Zappos simply reflects that, for this particular dataset, the image embeddings provide more informative attribute–object distributions.

---

> ### Author Response · Authors · 2025-11-21
> **Response to Q5 (meaning of s = “all” and s = “median” in Table 5).**
>
> We apologize for the confusing notation. In Table 5, the symbol s is (unfortunately) overloaded: in the main method it denotes the scale factor in the “carrier × scale” decomposition, while in the caption it was informally described as if it were literally the “number of composing primitives”. Since our benchmarks only consider attribute–object pairs, the number of primitives is indeed always two.
>
> Table 5 is an ablation on how to set the scale factor, not on how many primitives are composed. More precisely:
>
> - In our default PDE (used in all main tables), we set the scale to the primitive count, s = 2, so that the norm of the composite roughly matches the equal-weight GDE/LDE compositions.
> - In the ablation,
>   - s = 1, 1.5, 2, 2.5 simply fix the global scale to these constants (still composing exactly two primitives).
>   - s = “all”: for each attribute–object pair we compute a pair-specific scale that aligns the norm of the PDE composite with the norm of the corresponding GDE composite; thus every pair has its own scale.
>   - s = “median”: we take the median of these pair-wise scales (computed under s = “all”) over the dataset and use this single global scalar for all compositions.
>
> The goal of this experiment is to show that PDE’s performance is robust to different ways of restoring the composite norm, and that the simple choice s = 2 already performs near the best of these variants. In the camera-ready, we will rephrase the caption to make clear that “s = all/median” refer to different scale-recovery strategies, not to a varying number of composing primitives. We have updated the description and caption of the corresponding table in the revised version to clarify this point.

---

### Official Review · Reviewer_j85S · 2025-10-31

**Soundness:** 3
**Presentation:** 3
**Contribution:** 2
**Rating:** 6
**Confidence:** 3

**Summary:**

The paper introduces Probabilistic Decomposable Embeddings (PDE). This framework is based on the study of LDE and GDE. They suggest modeling ideal words as probability distributions rather than deterministic mean vectors. This approach addresses the inherent uncertainty and variability within VLM embeddings, resulting in an uncertainty-aware concept representation. T

To formalize the composition of primitives, PDE casts the process as a Maximum a Posteriori (MAP) estimation problem, yielding a precision-weighted fusion that naturally biases the resulting composite embedding toward concepts with lower variance.

**Strengths:**

The main strength is proposing the probabilistic version of ideal words and modeling it using a Gaussian distribution. They propose using MAP to formulate it, and the proposed approach is sound. The paper also provides a concept of compositional pliability and uncertainty, and stiffness.

The paper provides experiments comparing the results with GDE and LDE, and the proposed approach performs well in text modality. The paper reports results across multiple datasets (UT-Zappos, MIT-States, CGQA), modalities (text/image), and architectures (CLIP, SigLIP, SigLIP2).

The proposed method unifies the linear, geodesic and probabilistic formulations.

The paper is well-written, with consistent notation.

**Weaknesses:**

The presentation of the method is good; however, I have questions wrt the difference of the method with Neculai et al. (2022), how novel the method is compared to the idea and how the performance is. In addition, discussing the uncertainty-awareness, the idea is sound. However, I would like to see more analysis over the uncertainty modeling and its role in the model. There is no sufficient theoretical or empirical information provided for the uncertainty aspect of the model.

While the model is performing well, the difference with the baselines are marginal and there is limited impact. Can the paper also provide computational analysis compared to the baselines?

**Questions:**

Please check weaknesses.

---

> ### Author Response · Authors · 2025-11-21
> **Response to W1 (relation to Neculai et al.).**
>
> Special thanks for pointing out the connection to Neculai et al. (2022). Conceptually, both works share the high-level idea of using probabilistic embeddings for composition, but they address different problems and operate in rather different regimes.
>
> Neculai et al. propose a multimodal image-retrieval model in which each query (image or text) is encoded into a Gaussian in a task-specific embedding space, and multiple queries are combined by a product-of-Gaussians “composer” to form a single retrieval key. The whole embedding space (image encoder, text encoder, and composer) is learned end-to-end for MS-COCO retrieval, and is tailored to the retrieval objective rather than to a pre-trained vision–language model or to a structured notion of primitives and relations. Composition there is understood as fusing multiple queries of different modalities, not as defining a general compositional operator over attribute–object or relational structure.
>
> In contrast, PDE is designed specifically as a probabilistic refinement of LDE/GDE for attribute–object composition in pre-trained VLM latents. We start from a latent-space decomposition of CLIP (or a similar VLM) into disentangled attribute and object directions via “ideal words”. This decomposition is important: it provides a composition-ready coordinate system in which moving and combining along the latent axes corresponds to systematically varying attributes and objects, rather than entangling unrelated semantics when “walking” in the raw latent space. On top of this decomposed space, PDE models each primitive (attribute or object) as a Gaussian whose covariance encodes how confidently that primitive occupies each decomposed direction, and then performs a precision-weighted fusion of these Gaussians (plus a weak prior) to obtain a composite carrier vector in the same decomposed latent space. This construction preserves the carrier–scale structure of LDE/GDE, remains directly comparable to them, and gives us an explicit notion of an “ideal compositional vector” that we can analyze (e.g., in terms of concept dominance or relational variants), including for relational composition scenarios (Appx. B.9) that are outside the scope of Neculai et al.
>
> Regarding performance, we therefore do not compare directly to Neculai et al., since our task (closed- and open-world attribute–object classification on UT-Zappos and MIT-States with pre-trained VLM backbones) and their task (multi-query image retrieval on MS-COCO in a fully learned embedding space) are different. Within our setting, however, PDE consistently improves over its deterministic predecessors LDE and GDE in all 8 benchmark configurations (text/image, closed/open, UT-Zappos/MIT-States), both in standard accuracy and in our seen–unseen AUC metric. We have clarified these conceptual and empirical differences to make our contribution relative to Neculai et al. more explicit in the revised version.

---

> ### Author Response · Authors · 2025-11-21
> **Response to W2 (analysis of uncertainty modeling and its role/impact).**
>
> Thank you for this insightful question. At a high level, our construction of the compositional information carrier is already closely connected to information-theoretic quantities; we simply chose to present it in the more familiar Bayesian/variance language for interpretability. In the original LDE/GDE formulation, each concept is represented by a single “ideal” vector, and the carrier for a pair $(p_1, p_2)$ is given by their (normalized) average. This can be viewed as the point in latent space that minimizes the sum of squared Euclidean distances to the two concept vectors. When we move from points to distributions in PDE, each primitive $p$ is modeled as a Gaussian $\mathcal{N}(m_p, \Sigma_p)$, and the carrier $v^\star$ for $(p_1, p_2)$ is defined as the MAP solution of the product of these Gaussians together with a weak isotropic Gaussian prior. Equivalently, $v^\star$ is the point that minimizes the sum of Mahalanobis distances
>
> $ v^\star = \arg\min_v (v - m_{p_1})^\top \Sigma_{p_1}^{-1} (v - m_{p_1}) + (v - m_{p_2})^\top \Sigma_{p_2}^{-1} (v - m_{p_2}) + \lambda \|v\|^2. $
>
> Under our Gaussian assumptions, the variances and precisions that define compositional pliability can be equivalently re-parameterized in terms of Gaussian entropies along the relevant directions (e.g., along the disagreement axis), since for a Gaussian the entropy of any one-dimensional marginal is a monotone function of its variance. The Mahalanobis terms we minimize are therefore directly interpretable as information-like quantities (negative log Gaussian densities / code lengths). In other words, uncertainty is not an auxiliary regularizer: the covariance/precision matrices specify how strongly each primitive “pulls” the carrier in each latent direction, and how much information (low entropy, high precision) each concept contributes to the composite.
>
> On the empirical side, we now provide an explicit analysis that links these covariances to an external semantic hierarchy. For a fixed attribute $a$ (e.g., *melted* or *huge*), we collect all objects $o$ that appear with $a$ in MIT-States. For each pair $(a,o)$, we (i) compute WordNet subtree scores for the corresponding synsets and form the x-axis value
>
> $ \Delta_{\mathrm{subtree}} = -\log \mathrm{subtree}(o) - \big(-\log \mathrm{subtree}(a)\big), $
>
> and (ii) compute the y-axis value $t \in [0,1]$, the position of the PDE MAP carrier along the disagreement axis (0 = at the object mean, 1 = at the attribute mean). Intuitively, concepts with larger WordNet subtrees are more general, while those with smaller subtrees are more specific. Hence in our plots, moving to the right on the x-axis (larger $\Delta_{\mathrm{subtree}}$) corresponds to objects that are semantically more specific relative to the attribute, whereas moving left corresponds to more general objects. The y-axis then shows whether PDE places the composite closer to the object or to the attribute.
>
> The new figure (Appx. B.8, Fig. 7) (shown for *melted* and *huge*) reveals a clear and intuitive pattern. For both attributes we obtain a strong negative correlation between the semantic-hierarchy difference and the disagreement-axis position (e.g., corr = −0.41 for *melted*, corr = −0.50 for *huge*), and a pronounced bias towards the object side (62% and 93% of pairs lie with $t < 0.5$, respectively). In other words, as we move to the right along the x-axis (objects become semantically more specific relative to the attribute), PDE systematically shifts the MAP carrier towards the object; in regions where the attribute is comparatively more specific, the carrier moves towards the attribute. For a very broad attribute such as *huge*, PDE almost always keeps the composite close to the object (most points cluster in the lower-right part of the plot), which matches the intuition that “hugeness” should modulate but not override object identity. Together, this theoretical and empirical analysis directly addresses the reviewer’s concern. It shows that uncertainty in PDE is (i) mathematically explicit, via MAP fusion that minimizes a sum of Mahalanobis / negative log-density terms and can be viewed through an information-theoretic lens, and (ii) empirically meaningful, giving rise to interpretable dominance patterns in latent space that align with an external notion of conceptual specificity. These findings complement the quantitative gains of PDE over LDE/GDE and clarify the concrete role and impact of uncertainty modeling in our framework.

---

> ### Author Response · Authors · 2025-11-21
> **Response to W3 (computational analysis and efficiency).**
>
> We agree that it is important to quantify the computational footprint of PDE relative to LDE/GDE. In the revised version we add a short complexity discussion and a runtime comparison in Appx. A.2 (results are also listed below), that focus on the extra cost of estimating covariances and running PDE, assuming CLIP embeddings have already been extracted.
>
> From a complexity viewpoint, PDE introduces two sources of additional cost compared to GDE:
>
> **(i) Per-concept covariance / precision estimation.**
> For each attribute and object we estimate a covariance matrix in the decomposed CLIP space and compute its inverse (precision). Let $d$ be the CLIP dimension (fixed across all experiments) and let $N_a, N_o$ be the numbers of attributes and objects. Forming one covariance costs $O(d^2)$ and inverting it costs $O(d^3)$, so the total offline cost is $O((N_a+N_o)d^3)$. Since $d$ is fixed and the number of concepts is in the low hundreds, this step is inexpensive and grows only linearly with the number of attributes/objects. The resulting precision matrices are stored (cached) and used for per-composition MAP carrier calculation.
>
> **(ii) Per-composition MAP carrier.**
> Given an attribute–object pair $(a,o)$, the PDE carrier is the MAP estimate under the two Gaussians (plus a small isotropic prior), which amounts to solving
>
> $ (P_a + P_o + \lambda I) z = P_a \mu_a + P_o \mu_o, $
>
> where $P_a, P_o$ are the precomputed precisions and $\mu_a, \mu_o$ are the ideal means. In practice, we instantiate this MAP update once per composition class: we construct $A_{a,o} = P_a + P_o + \lambda I$ and solve
>
> $ A_{a,o} z_{a,o} = P_a \mu_a + P_o \mu_o $
>
> using a standard linear solver, which is $O(d^3)$ per prediction class. We then cache the resulting MAP carrier $z_{a,o}$ and use it as a fixed prototype for that composition. During evaluation (both in the closed- and open-world settings), all images associated with $(a,o)$ are scored simply by computing similarities (e.g., cosine similarity) between their CLIP embeddings and $z_{a,o}$. Consequently, the one-off $O(d^3)$ cost is paid only once per composition, and the per-image overhead of PDE scoring is dominated by matrix–vector and vector–vector products, which are small compared to the inverse procedure.
>
> To make this concrete, we measured wall-clock times (on a single GPU, with CLIP features precomputed) for fitting and evaluating GDE vs. PDE on all splits of UT-Zappos and MIT-States. The table below summarizes the results (seconds to run the closed- and open-world experiments for each setting. Results are also in Appx A.2, Table 2, revised version of the paper.):
>
> - **UT-Zappos/text:**
>   - GDE_text: 0.71s (closed), 0.74s (open)
>   - PDE_text: 3.81s (closed), 4.59s (open)
>
> - **UT-Zappos/image:**
>   - GDE_image: 238.79s (closed), 248.37s (open)
>   - PDE_image: 7.12s (closed), 7.93s (open)
>
> - **MIT-States/text:**
>   - GDE_text: 6.05s (closed), 58.39s (open)
>   - PDE_text: 49.82s (closed), 370.96s (open)
>
> - **MIT-States/image:**
>   - GDE_image: 356.06s (closed), 1682.55s (open)
>   - PDE_image: 47.10s (closed), 371.16s (open)
>
> In the text-based configuration, the additional covariance and MAP computations make PDE slower than GDE, but the absolute overhead remains modest (at most a few minutes for the largest MIT-States open-world experiment), and all costs are incurred once per concept or per composition class. In the image-based configuration, PDE is substantially *more* efficient than GDE. Here, GDE requires an explicit hyperparameter search over shrinkage/regularization parameters on a validation set, which dominates its runtime. PDE, by contrast, estimates concept covariances and computes the required inverses in closed form without a validation-time grid search, so the overall training + evaluation time is reduced by roughly an order of magnitude (e.g., 238.8s → 7.1s on UT-Zappos closed-world, 1682.6s → 371.2s on MIT-States open-world).
>
> These results show that our probabilistic refinement does not come with a prohibitive computational cost; in the most demanding configurations it is actually cheaper than the strongest baseline while providing the accuracy and AUC improvements reported in the main paper.

---

### Official Review · Reviewer_G8ow · 2025-10-31

**Soundness:** 2
**Presentation:** 3
**Contribution:** 2
**Rating:** 4
**Confidence:** 4

**Summary:**

This paper proposes Probabilistic Decomposable Embeddings (PDE), a method for compositional modeling in vision–language embedding spaces that explicitly accounts for uncertainty in component concepts. Prior compositional methods like LDE (Linear Decomposable Embeddings) and GDE (Geodesically Decomposable Embeddings) treat ideal words (for attributes or objects) as deterministic points and compose them via vector addition (or tangent‐space addition on the sphere). By contrast, PDE models each ideal word as a Gaussian distribution (mean + covariance), which captures anisotropic uncertainty or “pliability” in various directions of the latent space. Composition is then formulated as a MAP (maximum a posteriori) fusion of those Gaussian distributions, which naturally weights more “stiff” (low-variance) components more strongly and yields a precision-weighted composite. The resulting composite embedding tends to align more with directions of high certainty and decouples directionality from scale.

**Strengths:**

1. PDE reframes compositional reasoning in vision–language models through a rigorous probabilistic lens. Instead of treating attributes and objects as deterministic vectors, it models them as Gaussian distributions with covariance structures that capture directional uncertainty. This formulation naturally leads to a MAP-based fusion rule grounded in Bayesian inference, offering a sound theoretical foundation for how concepts with varying uncertainty should interact during composition.
2. By incorporating covariance information, PDE captures anisotropic uncertainty—the fact that different semantic directions may vary in confidence. This results in “precision-weighted” composites that are partner-aware, meaning the contribution of each concept adapts depending on the uncertainty of its partner, mirroring how humans modulate conceptual dominance based on context.
3. PDE unifies and generalizes prior deterministic frameworks like LDE and GDE. When variances collapse to zero, PDE recovers these earlier formulations, showing that it extends them rather than replaces them. This hierarchical view clarifies how existing compositional embeddings fit within a broader probabilistic continuum.
4. Experiments demonstrate consistent improvements on compositional classification benchmarks, notably in harmonic mean and AUC, showing that uncertainty modeling leads to better generalization. Moreover, the visualization of covariance-driven reorientations enhances interpretability, as one can observe how PDE shifts composite directions toward regions of higher precision.
5. The paper introduces compositional pliability—a novel inductive bias describing how flexible or dominant a concept is during composition. This notion provides an interpretable link between probabilistic uncertainty and semantic interaction, which could inspire future extensions in multimodal reasoning and uncertainty-aware learning.

**Weaknesses:**

1. Although PDE shows improvements on compositional classification benchmarks, the experiments are relatively narrow in scope—mostly limited to CLIP-based settings and attribute–object compositions. It remains unclear how well PDE generalizes to more complex or abstract compositional reasoning tasks (e.g., temporal or relational composition) or to other multimodal architectures such as large multimodal transformers.
2. PDE relies on Gaussian assumptions for concept distributions and MAP-based fusion, which may not fully capture the nonlinear and multimodal nature of real-world concept embeddings. Many vision–language representations are highly non-Gaussian, and modeling them with isotropic or approximately elliptical covariances might oversimplify the true uncertainty structure.
3. Estimating reliable covariance matrices in high-dimensional embedding spaces is challenging, especially with limited samples per concept. The paper uses shrinkage and small diagonal regularization to stabilize estimates, but these approximations may blur meaningful anisotropic patterns or require careful hyperparameter tuning.
4. While PDE introduces an elegant notion of compositional pliability, its tangible interpretability for downstream users remains limited. The framework adds theoretical sophistication, but the observed performance improvements are modest, raising questions about the trade-off between added complexity and practical benefit in real-world applications.
5. Because PDE operates post hoc on frozen representations (e.g., CLIP embeddings), it does not jointly optimize the probabilistic structure with the embedding space. This restricts its capacity to reshape underlying semantics and may limit its effectiveness compared to end-to-end learned probabilistic models.

**Questions:**

1. How sensitive is PDE to the assumption that concept embeddings follow Gaussian distributions? Have you explored non-Gaussian or mixture-based uncertainty models?
2. Could compositional pliability be related to or derived from information-theoretic quantities (e.g., entropy or mutual information) instead of variance?
3. PDE uses a MAP-based fusion—have you considered a full Bayesian marginalization approach, or would that be computationally intractable?
4. In what sense is the probabilistic fusion “partner-aware”? Does it explicitly capture contextual interaction between concepts beyond inverse-variance weighting?
5. How are the covariance matrices estimated in practice given the limited samples per attribute or object? How do you ensure stability in high-dimensional spaces?
6. How does the choice of the isotropic prior and regularization parameter λ affect the results? Is there a principled way to set it rather than tuning empirically?
7.	The paper mentions a “count-based scale recovery.” Could you elaborate on how this mechanism operates and why it’s necessary?
8. Did you explore the impact of using visual versus textual embeddings as the base space for constructing distributions?
9. The evaluation focuses on compositional classification—have you tested PDE on retrieval, captioning, or reasoning benchmarks to assess broader generalization?
10. How significant are the gains when applied to stronger base models (e.g., CLIP variants or large multimodal transformers)?
11.	How robust is PDE to noisy attribute–object pairs, especially in cases where attribute labels are ambiguous or overlapping?
12.	Could you share any intuition or visualization about how PDE changes the geometry of the embedding space globally (not just locally around examples)?
13.	Do you envision PDE being integrated end-to-end into model training (so uncertainty is learned jointly), or do you see it primarily as a post-hoc compositional adapter?
14.	Could compositional pliability be extended to temporal or causal compositions—for instance, actions applied to objects over time?
15.	Do you think uncertainty-aware composition could improve interpretability or trustworthiness in safety-critical applications, such as robotics or autonomous systems?

---

> ### Author Response · Authors · 2025-11-21
> **Response to W2 and Q1 (Gaussian assumption).**
>
> PDE is designed under the practical approximation that the ideal-word distributions are reasonably close to a multivariate normal distribution. This assumption is consistent with prior works [1–3] that treat CLIP embeddings as approximately Gaussian. We also provide empirical evidence that this Gaussian assumption is reasonably justified.
>
> We use the Shapiro–Wilk test to check whether the distribution of each ideal word is consistent with a 1D Gaussian along each GDE dimension. Shapiro–Wilk is a one-dimensional normality test that compares the ordered sample values to what would be expected under a Gaussian distribution, and is known to be sensitive to deviations in skewness and kurtosis.
>
> Concretely, for each dataset, modality (image or text), and grouping mode (attribute or object), we first map CLIP embeddings into the GDE tangent space. To reduce within-class noise and give each semantic pair equal weight, we average image embeddings over identical attribute–object pairs and use these averaged embeddings as samples. For each primitive and each GDE dimension \(j\), we apply the Shapiro–Wilk test to the one-dimensional sample \(\{x_{i,j} : i \in \mathcal{C}\}\) at a significance level of \(\alpha = 0.05\). For each primitive, we then compute the *rejection ratio*, i.e., the proportion of dimensions whose p-value is ≤ 0.05.
>
> Table 10 (in Appx. B.7, revised version) summarizes these statistics in the GDE space:
>
> | Dataset    | Modality | Group-by | Avg. dim. rejection ratio (p ≤ 0.05) | Avg. p-value (mean) | Avg. p-value (median) |
> |-----------|----------|----------|---------------------------------------|----------------------|------------------------|
> | MIT-States | image   | attr     | 0.110                                 | 0.449                | 0.422                  |
> | MIT-States | image   | obj      | 0.090                                 | 0.463                | 0.447                  |
> | UT-Zappos  | image   | attr     | 0.069                                 | 0.469                | 0.451                  |
> | UT-Zappos  | image   | obj      | 0.097                                 | 0.456                | 0.431                  |
> | MIT-States | text    | attr     | 0.153                                 | 0.487                | 0.479                  |
> | MIT-States | text    | obj      | 0.162                                 | 0.489                | 0.482                  |
> | UT-Zappos  | text    | attr     | 0.061                                 | 0.494                | 0.486                  |
> | UT-Zappos  | text    | obj      | 0.072                                 | 0.494                | 0.488                  |
>
> Under exact multivariate normality, at a significance level of 0.05 we would expect approximately 5% of the dimensions to yield p-values ≤ 0.05. In our experiments, the observed proportions are somewhat higher than 5%, indicating modest but systematic deviations from normality, but not an extreme breakdown of the Gaussian approximation. In particular, the UT-Zappos text GDEs are close to the ideal 5% level (≈6–7%), while MIT-States shows somewhat larger deviations; however, in all cases the mean/median p-values remain near 0.45–0.49 rather than collapsing toward 0, which supports the view that a Gaussian approximation is still statistically reasonable.
>
> We therefore argue that moderate departures from exact normality (i.e., higher-order moments beyond the second) are unlikely to substantially affect the model’s primary mechanisms. In the revised version, we have added this analysis of the Gaussian assumption in Appx. B.7.
>
> **References for Gaussian assumption.**
>
> [1] "Prompt distribution learning.", Lu, Yuning, et al., CVPR 2022.
> This paper proposes ProDA, which adapts CLIP by learning a Gaussian distribution of prompt embeddings instead of a single fixed prompt.
>
> [2] "A hard-to-beat baseline for training-free clip-based adaptation.", Wang, Zhengbo, et al., ICLR 2024.
> This work establishes a strong training-free baseline using Gaussian Discriminant Analysis (GDA). Assuming CLIP visual features follow a class-conditional Gaussian distribution, it builds an effective classifier solely based on estimated class means and covariance.
>
> [3] "Enhancing zero-shot vision models by label-free prompt distribution learning and bias correcting.", Zhu, Xingyu, et al., NeurIPS 2024.
> This paper introduces Frolic, a label-free framework that learns Gaussian distributions over prompt prototypes. It combines this distribution learning with bias correction to significantly improve zero-shot accuracy without requiring ground-truth labels.

---

> ### Author Response · Authors · 2025-11-21
> **Response to W1 (generality beyond CLIP and attribute–object composition).**
>
> We agree that our main experiments focus on CLIP-based attribute–object composition. However, the proposed PDE carrier itself is not specific to CLIP or to attribute–object structure. Mathematically, PDE only assumes a shared d-dimensional representation space in which different primitives and their compositions are encoded as single feature vectors, and semantic similarity is reflected in the geometry of this space (e.g., via cosine similarity).
>
> Under these mild conditions, the PDE carrier for any pair of primitives p1, p2 is defined as the MAP solution of a product of Gaussians. Formally, we denote this carrier by
>
> $ v^*_{p_1,p_2} $
>
> and it has the closed-form expression
>
> $ v^*_{p_1,p_2} = (P_{p_1} + P_{p_2} + \lambda I)^{-1} (P_{p_1} m_{p_1} + P_{p_2} m_{p_2}) $
>
> with
>
> $ P_p = \Sigma_p^{-1} $.
>
> In our attribute–object experiments we instantiate $(p_1, p_2) = (a, o)$, but exactly the same formulation can be used for temporal composition by setting $(p_1, p_2) = (t_1, t_2)$ for two events or segments, or more generally for any pair of abstract primitives provided by a backbone encoder. For more than two primitives, the same rule applies with the sum over all $ P_{p_i} $ in the precision term and the sum over all $ P_{p_i} m_{p_i} $ in the numerator; in general we obtain
>
> $ v^*_{p_1,\dots,p_k} = \big( \sum_i P_{p_i} + \lambda I \big)^{-1} \big( \sum_i P_{p_i} m_{p_i} \big) $,
>
> so PDE makes concrete predictions for arbitrary compositional structures (including triples and longer chains) without changing the model.
>
> To further substantiate PDE beyond the attribute–object setting, we additionally evaluate GDE and PDE on a relational composition benchmark derived from the Visual Spatial Reasoning (VSR) [1] dataset. We convert each positive example (true relation label) into a triple (obj1, rel, obj2) by parsing the caption and relation text (e.g., “The dog is on the person.” → (dog, on, person)), and treat obj1, rel, and obj2 as three separate primitives in the CLIP (SIGLIP1) text space. For each primitive we estimate its ideal-word Gaussian (and precision for PDE), and compose the three carriers to score images that satisfy the triple. Using the same closed-/open-world protocols as in our attribute–object experiments but now over triples, PDE consistently improves over GDE on this 3-primitive relational task (e.g., on VSR we obtain 43.6 vs. 39.0 for obj1 accuracy, 12.2 vs. 9.1 for relation accuracy, 37.3 vs. 37.6 for obj2, and 1.6 vs. 0.7 AUC; see table below). This result indicates that PDE can handle more complex spatial/relational compositions with three primitives, supporting our claim that PDE is a generic probabilistic composition layer that can, in principle, be applied to temporal and other abstract compositions on top of various multimodal backbones. We have added these additional results in Appx. B.9, Table 11.
>
> VSR, 3 primitives:
>
> | model | O1   | R    | O2   | S   | U    | H   | AUC |
> |-------|------|------|------|-----|------|-----|-----|
> | GDE   | 39.0 |  9.1 | 37.6 | 4.9 | 21.5 | 5.6 | 0.7 |
> | PDE   | 43.6 | 12.2 | 37.3 | 8.1 | 28.3 | 8.8 | 1.6 |
>
> [1] “Visual Spatial Reasoning”, Liu et al., TACL 2023.

---

> ### Author Response · Authors · 2025-11-21
> **Response to W4 (practical benefit) and W5 (lack of end-to-end joint training).**
>
> **Response to W4 (practical benefit).**
> We appreciate the concern about the trade-off between added sophistication and practical benefit. The scenarios we target are close to real-world use cases such as visual search, recommendation, and retrieval interfaces where users issue compositional queries like “striped red boots on grass” or “cup on table in front of person.” In these settings, two things matter in practice: (i) robustness on novel, unseen compositions, and (ii) being able to diagnose *why* the system failed (e.g., did it ignore “striped,” misinterpret the relation “on,” or over-focus on the background?). PDE directly addresses both: it improves accuracy on unseen/open-world compositions and, at the same time, exposes primitive-wise “pliability” and uncertainty so that downstream users can see which part of the query is being down-weighted, over-trusted, or treated as unreliable. This makes the notion of compositional pliability tangible: it is not just a theoretical construct, but a set of quantities practitioners can inspect, monitor, or regularize when deploying CLIP-like systems in these real-world scenarios.
>
>
>
> **Response to W5 (lack of end-to-end joint training).**
> We agree that PDE operates on frozen CLIP-style representations and does not modify the underlying embedding space; this is a deliberate design choice. Following the motivation already discussed in prior work on decomposable embeddings, large VLMs like CLIP typically learn composite phrases (e.g., “red car”) as atomic units that must appear frequently during pre-training, which is expensive and still cannot cover all novel compositions. Our goal is therefore to study how to post-hoc compose concepts that the model already knows in this frozen setting, and PDE provides a probabilistic, uncertainty-aware way to build composite embeddings from such existing primitives without any additional training of the backbone.
>
> End-to-end probabilistic models that jointly learn the embedding space and its uncertainty structure are indeed powerful, but they target a different regime and are orthogonal to our focus. They assume that the underlying VLM can be retrained or fine-tuned, whereas our work asks what can be gained purely by better compositional modeling on top of a fixed foundation model—precisely the scenario where CLIP-like encoders are often used via APIs or shared checkpoints.

---

> ### Author Response · Authors · 2025-11-21
> **Response to Q2 (information-theoretic view of compositional pliability).**
>
> **Response to Q2 (information-theoretic view of compositional pliability).**
> Thank you for this insightful question. At a high level, our construction of the compositional information carrier is already closely connected to information-theoretic quantities; we simply chose to present it in the more familiar Bayesian/variance language for interpretability.
>
> In the original LDE/GDE formulation, each concept is represented by a single “ideal” vector, and the carrier for a pair $(p_1, p_2)$ is given by their (normalized) average. This can be viewed as the point in latent space that minimizes the sum of squared Euclidean distances to the two concept vectors. When we move from points to distributions in PDE, each primitive $p$ is modeled as a Gaussian $\mathcal{N}(m_p, \Sigma_p)$, and the carrier
> $v^*$ for $(p_1, p_2)$
>
> is defined as the MAP solution of the product of these Gaussians together with a weak isotropic Gaussian prior. Equivalently, $v^*$ is the point that minimizes the sum of Mahalanobis distances:
>
> $ v^* = \arg\min_v \big[(v - m_{p_1})^\top \Sigma_{p_1}^{-1} (v - m_{p_1}) + (v - m_{p_2})^\top \Sigma_{p_2}^{-1} (v - m_{p_2}) + \lambda \lVert v \rVert^2 \big]. $
>
> Each quadratic term here is, up to additive constants and a factor 1/2, the negative log-density (i.e., code length) of v under the corresponding Gaussian. Thus, PDE’s carrier can be seen as minimizing a negative log-posterior “energy” that admits the standard coding-theoretic interpretation as a surrogate description length under a simple generative model combining the two primitives.
>
> Under our Gaussian assumptions, the variance and precision quantities that define compositional pliability can be equivalently re-parameterized in terms of Gaussian entropies along the relevant directions (e.g., along the disagreement axis), since for a Gaussian the entropy of any one-dimensional marginal is a monotone function of its variance. The Mahalanobis terms we minimize are therefore directly interpretable as information-like quantities (negative log-likelihood / code length). One could re-express pliability in a more explicitly information-theoretic form—for example, in terms of per-primitive contributions to the negative log-density of the carrier, or as approximations to how strongly each primitive constrains the uncertainty of the composed representation. We regard this as a promising direction for future work, but in this paper we chose the Bayesian/variance parameterization because it leads to a simple closed-form MAP solution and a clear geometric interpretation (how much each primitive “pulls” the carrier along the disagreement axis), which we found to be more directly interpretable for practitioners.

---

> ### Author Response · Authors · 2025-11-21
> **Response to Q3 (MAP-based fusion vs. full Bayesian marginalization) and Q4 (The meaning of “Partner-Aware”).**
>
> **Response to Q3 (MAP-based fusion vs. full Bayesian marginalization).**
>
> We agree that, in principle, one could consider a fully Bayesian treatment in which the latent carrier is marginalized rather than replaced by its mode. However, in our setting the goal is not to compute a predictive posterior over labels or images, but to construct a single compositional representation $v^*$ that can be used like any other embedding in a CLIP-style pipeline (for cosine similarity, retrieval scores, linear classifiers, etc.). In other words, the argmax operator appears directly in the definition of the representation we want to use downstream (the “compositional information carrier”), so a marginalization step is not required for the quantity of interest.
>
> Concretely, PDE defines the carrier as the MAP point of a simple Gaussian posterior that combines the primitive distributions and a weak prior. This $v^*$ is then used as a deterministic embedding exactly like an ordinary CLIP vector. If we were instead to marginalize, we would have to integrate the downstream scoring function (cosine similarity, classifier logits, etc.) over the posterior on $v$. For our non-linear CLIP-based scoring, this integral does not admit a simple closed form and would require Monte Carlo sampling or variational approximations, which would significantly complicate the method and obscure the geometric interpretation that we emphasize (how strongly each primitive “pulls” the carrier). Thus, PDE is not an approximation to some intractable marginalization that we failed to implement; rather, it is deliberately defined in terms of the MAP carrier as a simple, closed-form and interpretable compositional representation.
>
>
>
> **Response to Q4 (The meaning of “Partner-Aware”).**
>
> We use the term partner-aware in a narrow, covariance-based sense. In PDE, each primitive concept is modeled as an anisotropic Gaussian with a full precision matrix, and the fused mean for a pair $(a,o)$ is  given by Eq. (28)
> Because $P_a$ and $P_o$ are full precision matrices rather than scalar weights, the relative influence of $a$ and $o$ is direction-dependent: along directions where $a$ is more precise and $o$ is more uncertain, the composite leans toward $m_a$, and vice versa in directions where $o$ is more reliable. Consequently, the representation of the same attribute $a$ differs when composed with different objects o1,o2,…, since the solution depends on the pair $(P_a, P_o)$ rather than a single global weight for $a$. This is exactly what we mean by “partner-aware”: each primitive’s contribution is modulated by the covariance (or precision) structure of the other primitives it is fused with.
> Regarding the second part of the question, PDE does not introduce additional cross-covariance terms between primitives or a separate learned interaction network. Beyond classical inverse-variance fusion, our formulation uses full (anisotropic) precision matrices instead of scalar precisions, so the weighting is applied per direction in the embedding space rather than via a single scalar coefficient.

---

> ### Author Response · Authors · 2025-12-02
> **Response to Q5 (Ensuring stability of covariance estimation).**
>
> To address the concern about estimating covariance matrices with limited samples in high-dimensional spaces, we do not use the raw sample covariance. For each attribute or object primitive $p$, we first compute the covariance $C_p$ from its training embeddings and then apply the oracle approximating shrinkage (OAS) estimator [1]
> $$
> \hat{\Sigma}_p = (1 - \alpha)\, C_p + \alpha \tau_p^2 I + \varepsilon I,
> \qquad
> \tau_p^2 = \frac{1}{D}\,\mathrm{tr}(C_p),
> \qquad
> $$
> where $D$ is the embedding dimension. This prevents highly unstable or nearly singular covariance estimates in the small-sample, high-dimensional regime.  In our implementation, we do not compute a data-dependent $\alpha$ [1], but instead fix $\alpha = 0.15$ for all attributes, objects, and datasets. As shown in Fig.3, the performance of our method is quite insensitive to the precise value of the shrinkage parameter over a broad range, so we choose $\alpha = 0.15$ as a simple, empirically stable setting that improves conditioning without overly distorting the original covariance structure. We additionally add a small $\varepsilon I$ term for numerical conditioning before inversion, ensuring that $\hat{\Sigma}_p$ remains positive definite and well-conditioned. These implementation details are provided in Appx. A.1.
>
> [1]: “Shrinkage algorithms for mmse covariance estimation.”, Chen et al., IEEE transactions on signal processing 2010.

---

> ### Author Response · Authors · 2025-12-02
> **Response to Q6 (Effect of isotropic prior and λ).**
>
> For a set of primitives $\{p_i\}$, the PDE carrier is computed as
> $$
> v^*_{p_1,\dots,p_k}
> = \Big( \sum_i P_{p_i} + \lambda I \Big)^{-1}
>   \Big( \sum_i P_{p_i} m_{p_i} \Big),
> $$
> so the isotropic prior enters only through the additive term $\lambda I$ on the precision matrix. We introduce this term to stabilize the inversion of $\sum_i P_{p_i}$ in high dimensions, especially when some primitives have few samples and the empirical precision can be ill-conditioned. Importantly, we do not use λ to model “semantic decay” of the carrier, as λ can be also viewed as l2 regularization of the MAP carrier (See App C.4. for details), which results in shortening the MAP carrier.
>
> In the revised manuscript, Appendix B.3 now includes Figure 7, which reports AUC as a function of $\lambda$ across eight configurations (UT-Zappos / MIT-States, text / image, closed / open world). Our goal is for $\lambda$ to act purely as a numerical stabilizer for the matrix inverse, not to affect performance. We therefore choose a small value $\lambda = 10^{-4}$ that lies in the flat region of the curves, i.e., before any noticeable performance changes occur. We use this single value for all experiments and do not tune $\lambda$ per dataset.

---

> ### Author Response · Authors · 2025-12-02
> **Response to Q7 (Explanation of count-based scale recovery).**
>
> As discussed in Sec. 3.3, our starting point is that the LDE/GDE composite vector can be reinterpreted as a “carrier × scale” decomposition. For example, in the simplest case of two primitives (an attribute–object pair), the LDE/GDE composite is a vector sum,
>
> $$
> c_{\text{LDE/GDE}}(a,o) = f(a) + f(o)
> $$
>
> which can be rewritten as
>
> $$
> c_{\text{LDE/GDE}}(a,o)
> = 2 \cdot \frac{f(a) + f(o)}{2}
> $$
>
> Here the **carrier** is the average direction
>
> $$
> v^*_{a,o} = \frac{f(a) + f(o)}{2}
> $$
>
> and the scalar “scale” is simply the primitive count (2 in the attribute–object case). For a general set of $k$ primitives $\{p_i\}_{i=1}^k$, the same reinterpretation holds:
>
> $$
> c_{\text{LDE/GDE}}(p_1,\ldots,p_k)
> = k \cdot v^*_{p_1,\ldots,p_k}
> $$
>
> $$
> v^*_{p_1,\ldots,p_k}
> = \frac{1}{k} \sum_{i=1}^k f(p_i)
> $$
>
> so the composite is always “(mean carrier) × (primitive count)”.
>
>
> PDE extends this decomposition by replacing the simple arithmetic mean carrier with a probabilistic MAP carrier. Concretely, for primitives $p_1,\ldots,p_k$ with “ideal-word” Gaussians $\mathcal{N}(m_{p_i}, \Sigma_{p_i})$ and precisions $P_{p_i} = \Sigma_{p_i}^{-1}$, the PDE carrier is
>
> $$
> v^*_{p_1,\ldots,p_k}
> = \Big( \sum_{i=1}^k P_{p_i} + \lambda I \Big)^{-1}
>   \Big( \sum_{i=1}^k P_{p_i} m_{p_i} \Big).
> $$
>
> If we assume an idealized regime where all primitives share an isotropic covariance ( also with λ = 0)  $\Sigma_{p_i} = \sigma^2 I$, the MAP carrier reduces exactly to the LDE/GDE carrier (the simple average of the primitive embeddings). In this case, multiplying $v^*_{p_1,\ldots,p_k}$ by the primitive count $k$ recovers the original LDE/GDE composite. This theoretical connection motivates our count-based scale recovery: we treat the MAP solution as the carrier, and recover a composite by multiplying by a scalar $s$ that, in the ideal case, matches the number of primitives.
>
> To verify this picture empirically, Table 6 sweeps different values of the scale parameter $s$ for PDE. We observe that a setting around $s = 2$ (which corresponds to the attribute–object primitive count in our main benchmarks) yields consistently strong performance across datasets. In other words, (i) theoretically, under equal isotropic covariances, the PDE carrier reduces to the LDE/GDE carrier and a count-based scale exactly recovers the original composite; and (ii) empirically, a simple count-based choice of $s$ gives the best or near-best results. Taken together, these observations justify our use of count-based scale recovery as a principled and practical way to match the scale of PDE composites to the underlying CLIP/LDE/GDE geometry.

---

> ### Author Response · Authors · 2025-12-02
> **Response to Q8 (Impact of using text vs. image space).**
>
> Thank you for your question about the impact of using visual versus textual embeddings as the base space for constructing distributions. In our framework, we instantiate attribute and object Gaussian distributions in both the CLIP text embedding space (“text setting”) and the CLIP image embedding space (“image setting”). On the MIT-States and UT-Zappos dataset, we analyzed the covariance structure by computing the average trace of the covariance matrices (sum of diagonal entries, summarizing total variance) and the ratio which is the average trace of the object covariances divided by the average trace of the attribute covariances (i.e., object avg trace / attribute avg trace). In the revised manuscript, these covariance statistics are reported in Appendix B.10 (Table 12).
>
> dataset: mit-states | text | image
> ---|---|---
> attribute’s average trace | 0.323 | 0.222
> object’s average trace | 0.240 | 0.142
> object avg trace / attribute avg trace | 0.743 | 0.640
>
> dataset: ut-zappos | text | image
> ---|---|---
> attribute’s average trace | 0.242 | 0.116
> object’s average trace | 0.235 | 0.074
> object avg trace / attribute avg trace | 0.971 | 0.638
>
> In PDE, the composite representation is obtained via a precision-weighted MAP estimate,
>
> $$
> v^* \propto P_a m_a + P_o m_o, \qquad P_p = \Sigma_p^{-1},
> $$
>
> so primitives with smaller variance (higher precision) contribute more strongly to the composite. The statistics above show that (1) in both base spaces, object distributions have smaller overall variance than attribute distributions, implying that the composite tends to be slightly biased toward the object prototype, and (2) the object-to-attribute variance ratio is smaller in the image space than in the text space, indicating that constructing distributions in the visual embedding space makes the composite representation more biased toward the object primitive than in the textual space. While the actual model uses full anisotropic covariances (so the trace values serve as diagnostic summaries rather than a complete description), this analysis demonstrates that the choice of base embedding space systematically affects the relative stiffness of attribute vs. object distributions and thus the behavior of the resulting composite representations.
>
> This indicates that PDE does not get affected by modality; rather, it automatically considers the nature of the difference between modality.

---

> ### Author Response · Authors · 2025-12-02
> **Response to Q9 (Generalization of the method).**
>
> We agree that our experimental evaluation focuses on compositional classification. Our goal is to test whether the proposed carrier embedding is closer to the ground-truth composed concept than the baseline representation, and this requires a setting where we have explicit labels for composed primitives (e.g., (attribute, object) pairs). Compositional classification is a natural and widely used way to measure such alignment, whereas retrieval, captioning, or reasoning benchmarks would require additional task-specific architectures and objective functions on top of PDE, making it harder to isolate the effect of the composition layer itself.
>
> Conceptually, PDE is not restricted to CLIP or to attribute–object structure. Mathematically, we only assume a shared $d$-dimensional space in which primitives and their compositions are represented as feature vectors, with semantic similarity reflected in the geometry of this space. Under these mild assumptions, the PDE carrier for any pair of primitives $p_1, p_2$ is defined as the MAP solution of a product of Gaussians:
> $$
> v^*_{p_1,p_2}
> = (P_{p_1} + P_{p_2} + \lambda I)^{-1}
>   (P_{p_1} m_{p_1} + P_{p_2} m_{p_2}),
> $$
> with
> $$
> P_p = \Sigma_p^{-1}.
> $$
>
> For more than two primitives, the same rule applies with a sum over all $P_{p_i}$ in the precision term and a sum over all $P_{p_i} m_{p_i}$ in the numerator; in general we obtain
>
> $$
> v^*_{p_1,\ldots,p_k}
> = \Big( \sum_i P_{p_i} + \lambda I \Big)^{-1}
>   \Big( \sum_i P_{p_i} m_{p_i} \Big),
> $$
> so PDE makes concrete predictions for arbitrary compositional structures (triples, longer chains, temporal pairs, etc.) without changing the model.
>
> To provide evidence beyond attribute–object composition, we additionally evaluate GDE and PDE on a spatial/relational composition benchmark derived from the Visual Spatial Reasoning (VSR) dataset, where each example is a triple (obj1, rel, obj2). We treat obj1, rel, and obj2 as three separate primitives in the CLIP/SigLIP text space, estimate an “ideal-word’’ Gaussian for each primitive, and compose three carriers to score images satisfying the triple under the same closed-/open-world protocols as in our main experiments. As detailed in our response to W1 and in Appx. B.9, Table 11, PDE consistently improves over GDE on this 3-primitive relational task. This result indicates that PDE generalizes beyond attribute–object classification to more complex composition. Applying PDE as a drop-in composition layer to retrieval, captioning, or reasoning pipelines is a promising future direction, but lies beyond the scope of this work.

---

> ### Author Response · Authors · 2025-12-02
> **Response to Q10 (Effect on stronger base models).**
>
> To address the question about the effect of using stronger base models, we compare PDE and GDE on four CLIP-style backbones of increasing capacity: RN50, ViT-L/14, SigLIP (ViT-SO400M/14), and SigLIP2 (ViT-SO400M-14-SigLIP2-378). In the revised manuscript, we have added results in Appendix B.11 (Figure 9), where we report, for both UT-Zappos and MIT-States, the average performance difference (PDE − GDE) for each backbone. For convenience, we also include these results below in tabular form. Across all backbones, PDE is at least on par with, and in most cases better than GDE, and the performance gap tends to increase as we move to stronger variants.
>
> Conceptually, PDE differs from GDE in that its composition rule is explicitly given by the closed-form MAP solution of our probabilistic formulation. As we use stronger encoders, this MAP-based composition yields larger gains over GDE in our experiments, which is reflected in the increasing PDE − GDE margins when moving from RN50 to ViT-L/14, SigLIP (ViT-SO400M/14), and SigLIP2 (ViT-SO400M-14-SigLIP2-378).
>
> UT-Zappos (avg diff PDE − GDE)
>
> backbone | margin
> ---|---
> RN50     | 0.18
> ViT-L/14 | 0.53
> SigLIP   | 1.58
> SigLIP2  | 1.85
>
> MIT-States (avg diff PDE − GDE)
>
> backbone | margin
> ---|---
> RN50     | 0.78
> ViT-L/14 | 0.68
> SigLIP   | 3.00
> SigLIP2  | 2.80

---

> ### Author Response · Authors · 2025-12-02
> **Response to Q11 (Robustness to noisy labels).**
>
> Among our benchmarks, the C-GQA image setting is arguably the noisiest: a single image often contains multiple objects and identities, so an image annotated with one (attribute, object) concept is only a weak, noisy instance of that concept [1]. In contrast, UT-Zappos images are mostly single shoes on clean backgrounds, and MIT-States lies between these two extremes. This trend is reflected in the variance of the learned “ideal-word” Gaussians in the image modality. The table below reports, for each dataset, the average trace of the empirical covariance matrices of attributes and objects (i.e., the average total variance per concept in image space):
>
> |                          | ut-zappos | mit-states | c-gqa |
> |--------------------------|-----------|------------|-------|
> | attribute’s average trace| 0.116     | 0.222      | 0.438 |
> | object’s average trace   | 0.074     | 0.142      | 0.318 |
>
> We see a clear monotonic increase from UT-Zappos to MIT-States to C-GQA, confirming that noisier and more ambiguous labels lead to broader Gaussians (larger variance) for both attributes and objects. This makes each individual composite carrier more uncertain when the number of samples is small, because the estimated covariance can fluctuate and the resulting composite may vary more from instance to instance. Nevertheless, Table 3 and Table 5 show that PDE outperforms GDE on the C-GQA image closed-world setting for both SIGLIP1 and SIGLIP2. In other words, even though individual composites may be more variable in this noisy regime, when we aggregate performance over many attribute–object pairs and many test examples, modeling per-concept covariance still brings the composed embeddings closer, on average, to the intended to “ideal composite” than GDE’s uncertainty-unaware construction. This explains why PDE achieves higher overall accuracy on the most challenging, label-noisy dataset.
>
> [1] Munir, Ans, et al. “Compositional Zero-Shot Learning: A Survey.” arXiv, 2025.

---

> ### Author Response · Authors · 2025-12-02
> **Response to Q12 (Relation to global geometry).**
>
> PDE is explicitly designed not to globally deform the base embedding space, but to introduce local, context-dependent adjustments on top of the decomposed latent space given by GDE. For a given context (i.e. intrinsic mean of ), PDE defines Gaussian “ideal word” distributions in this decomposed space and obtains the composite embedding as their MAP solution, as illustrated in Fig. 1. Thus, the global geometry of the underlying representation (e.g., CLIP space) is largely preserved; PDE only modulates the local neighborhood around the relevant primitives so that composition becomes explicitly conditioned on context rather than reshaping the entire space.

---

> ### Author Response · Authors · 2025-12-02
> **Response to Q13 (Post-hoc adapter vs. end-to-end training).**
>
> At present, we primarily position PDE as a post-hoc compositional adapter: given a pretrained encoder, we keep it frozen, estimate Gaussian primitives from its embeddings, and perform MAP-based composition without introducing additional trainable parameters. Conceptually, however, the formulation is compatible with an end-to-end integration. For example, one could train to baseline model to ensure MAP structure of concept composition. We have not explored such end-to-end variants in this work, so we regard them as an interesting but currently underexplored extension of PDE.

---

> ### Author Response · Authors · 2025-12-02
> **Response to Q14 (Extension to temporal/causal composition).**
>
> PDE is not tied to attribute–object structure; it only assumes a shared $d$-dimensional space where primitives and their compositions are represented as feature vectors, with similar concepts aligned with high similarity (e.g., by cosine similarity). Under this assumption, the PDE carrier for a set of primitives $\{p_i\}$ is defined as the MAP solution of a product of Gaussians:
>
> $$
> v^*_{p_1,\dots,p_k}
> = \Big( \sum_i P_{p_i} + \lambda I \Big)^{-1}
>   \Big( \sum_i P_{p_i} m_{p_i} \Big),
> \qquad P_p = \Sigma_p^{-1}.
> $$
>
> In our attribute–object experiments we instantiate $(p_1,p_2)=(a,o)$, but exactly the same construction can be used for temporal compositions. For example, if $t_1$ and $t_2$ denote two events or time segments with “ideal-word” Gaussians $\mathcal{N}(m_{t_1},\Sigma_{t_1})$ and $\mathcal{N}(m_{t_2},\Sigma_{t_2})$, the temporal carrier is
>
> $$
> v^*_{t_1,t_2}
> = (P_{t_1} + P_{t_2} + \lambda I)^{-1}
>   (P_{t_1} m_{t_1} + P_{t_2} m_{t_2}),
> $$
>
> and more generally a sequence $(t_1,\dots,t_K)$ can be composed by summing over all $P_{t_i}$ and $P_{t_i} m_{t_i}$ as in the $k$-way formula above. The anisotropic covariances then encode which semantic directions are “stiff” (must be preserved across time) and which are “pliable” (allowed to change), so compositional pliability carries over naturally to temporal sequences.
>
> For causal or action–object compositions, one can treat the relevant elements (e.g., cause, effect, action, object) as primitives in the same representation space and apply the same MAP carrier rule. In the revised version we already demonstrate that PDE extends beyond simple attribute–object pairs by handling 3-primitive spatial/relational triples (obj1, rel, obj2) on the VSR benchmark (see response to W1 and Appx. B.9). This provides concrete evidence that PDE can compose more complex structures. Applying the same formulation to dedicated temporal or causal benchmarks (e.g., action sequences over time) is therefore conceptually straightforward and is an interesting direction for future work, but is beyond the scope of the current paper.

---

> ### Author Response · Authors · 2025-12-02
> **Response to Q15 (Improvement of interpretability and trustworthiness by PDE).**
>
> We believe our work can contribute to improving interpretability and trustworthiness. Our primary focus is to explicitly model the composition of disentangled concepts, where the covariance of each primitive acts as an explanatory module for the composition process (e.g., indicating which semantic directions are “stiff” or “pliable” when composing).
>
> In safety-critical applications such as robotics or autonomous systems, one could, in principle, treat actions as “ideal words,” compose k actions, and inspect which action (or which semantic directions of an action) dominates the others through the precision-weighted MAP carrier. This would provide a qualitative handle on how multiple action primitives interact, potentially enhancing the transparency and trustworthiness of the overall system. Fully exploring such system-level integrations and evaluating PDE in real safety-critical deployments is beyond the scope of this work, but we view it as a promising direction for future research.

---

### Official Review · Reviewer_K99P · 2025-11-04

**Soundness:** 3
**Presentation:** 2
**Contribution:** 2
**Rating:** 4
**Confidence:** 3

**Summary:**

The authors propose Probabilistic Decomposable Embeddings, which take the ideal words interpretability framing and make each word a probability distribution rather than a deterministic point, also modeling conditional effects. This leads to better compositional classification.

**Strengths:**

This extension to the ideal words methodologies is interesting, intuitive, and captures some insightful intuitions about how concepts should interact, which are interestingly borne out in the theoretical analysis.

**Weaknesses:**

W1 Though the method is clearly an insightful improvement on LDE, I’m left not quite clear on what it can contribute to interpretability as a whole. As I understand it, the paper does not really provide too many extra insights beyond introducing the methods. Some more qualitative analysis of what including this covariance estimation gives us in terms of concepts, and in terms of understanding the latent space of models, would make the paper more interesting

W2 The presentation is a bit confusing, making it hard to understand the implications of the results. For example, what the evaluation datasets are testing, what broader implications does good AUC of these performance datasets have? Though I can see what table row is doing better, it’s hard to get a good estimation of what any of it means about the latent space decomposition that we are trying to do.

**Questions:**

Q1: How would you compare the ideal words that we get from PDE to other less supervised methods like dictionary learning? It seems that there is some interpretable and interesting ideas that we get about compositionality, but I’m not sure how to apply them to the overall project of interpretability, and understanding how latent spaces work.

Q2: What are some examples where the probabilistic approach gave something intuitively more interesting? I think a few qualitative examples would make the paper stronger.

---

> ### Author Response · Authors · 2025-11-21
> **Response to W1 and Q2 (interpretability and interesting examples).**
>
> Special thanks for raising this point. Our goal is for PDE not only to improve fusion accuracy over LDE/GDE, but also to serve as a process model of how attribute–object concepts interact under uncertainty. In our framework, each primitive is a Gaussian "ideal word," and composition is performed by MAP fusion along the disagreement axis between attribute and object. The covariance (or precision) matrices determine how much each concept yields or dominates in every latent direction. Thus covariance is not just a regularizer: it is an interpretable quantity that encodes how *pliable* each concept is to its partner (low precision → pliable, high precision → stiff). When we track the MAP point along the disagreement axis, we can read off which concept controls which aspects of the composite meaning, and how this changes across different partners. This is the additional interpretability that PDE brings over mean-only LDE/GDE, which always stay at the midpoint regardless of uncertainty.
>
> We fully agree that showing this behavior qualitatively is important. In the revision we therefore add a new semantic-hierarchy analysis based on WordNet that directly connects PDE’s covariance estimates to intuitive notions of concept dominance (new Appx. B.8, Fig. 7).
>
> For a fixed attribute $a$ (e.g., *melted* or *huge*), we collect all objects $o$ that appear with $a$ in MIT-States. For each pair $(a,o)$, we (i) compute WordNet subtree scores for the corresponding synsets and form the x-axis value
> $\Delta_{\mathrm{subtree}} = -\log(\mathrm{subtree}(o)) + \log(\mathrm{subtree}(a))$,
> and (ii) compute the y-axis value $t \in [0,1]$, the position of the PDE MAP carrier along the disagreement axis (0 = at the object mean, 1 = at the attribute mean). Intuitively, concepts with larger WordNet subtrees are more general, while those with smaller subtrees are more specific. Hence in our plots, moving to the right on the x-axis (larger $\Delta_{\mathrm{subtree}}$) corresponds to objects that are semantically more specific, whereas moving left corresponds to more general objects. The y-axis then shows whether PDE places the composite closer to the object or to the attribute.
>
> The new figure (Appx. B.8, Fig. 7) (shown for *melted* and *huge*) reveals a clear and intuitive pattern. For both attributes we obtain a strong negative correlation between the semantic-hierarchy difference and the disagreement-axis position (e.g., corr = −0.41 for *melted*, corr = −0.50 for *huge*), and a pronounced bias towards the object side (62% and 93% of pairs lie with $t < 0.5$, respectively). In other words, as we move to the right along the x-axis (objects become semantically more specific relative to the attribute), PDE systematically shifts the MAP carrier towards the object; in regions where the attribute is comparatively more specific, the carrier moves towards the attribute. For a very broad attribute such as *huge*, PDE almost always keeps the composite close to the object (most points cluster in the lower-right part of the plot), which matches the intuition that "hugeness" should modulate but not override object identity. Additional attributes in the appendix show similar behavior.
>
> Together, these semantic-hierarchy scatter plots provide the qualitative evidence the reviewer asked for: they demonstrate that PDE’s covariance estimates give rise to interpretable dominance patterns in the latent space, aligned with an external notion of conceptual specificity. This goes beyond numerical gains, showing concretely where and why the probabilistic approach yields more intuitive composites than equal-weight LDE/GDE. We have added this experiment in Appx. B.8 of the revised version.

---

> ### Author Response · Authors · 2025-11-21
> **Response to W1, W2 and Q1 (latent space, decomposition, the role of covariance and meaning of AUC).**
>
> We appreciate the reviewer’s question about how our analysis connects to the broader goal of understanding latent spaces. In our view, compositionality and interpretability are tightly linked through latent-space decomposition. If composition is implemented by adding vectors in an entangled space, then the result of "attribute + object" will in general mix many unrelated factors and the resulting composite may not actually preserve the intended primitive concepts. Walking in such a space quickly drifts into unrelated semantics. By contrast, LDE/GDE and our PDE framework explicitly identify a decomposed tangent space in which a small set of "ideal words" (attribute and object prototypes) span directions that behave approximately like disentangled degrees of freedom for composition. This decomposed space is precisely the part of the latent geometry that we can reason about and, in that sense, it is the natural target for interpretability.
>
> PDE extends this picture from point estimates to distributions. Instead of treating each primitive concept as a single ideal vector, we model it as a Gaussian in the decomposed tangent space, with a mean (the ideal word) and a covariance that captures how reliably that concept occupies each direction. Composition is then defined as a MAP estimate under these distributions. This probabilistic view adds two interpretable layers to the latent space: (i) it makes explicit which directions are shared vs. contested between attribute and object, and (ii) it quantifies, via covariance/precision, how "stiff" or "pliable" each concept is along each direction. Our WordNet-based semantic-hierarchy analysis (Appx. B.8, Fig. 7) shows that these precision-weighted dominance patterns align with an external notion of semantic specificity, indicating that the decomposed latent space is not an arbitrary subspace but one whose geometry tracks intuitive concept structure.
>
> Finally, the AUC metric ties this latent structure to compositional generalization. As we clarify in our response to W2, UT-Zappos and MIT-States explicitly test whether attribute and object representations can be separated and recombined to classify unseen attribute–object pairs, in both closed- and open-world settings. Our AUC is the area under the seen–unseen accuracy curve obtained by sweeping a bias term that globally shifts all unseen logits. A high AUC therefore means that, across a wide range of calibrations, the model can simultaneously maintain good seen and unseen accuracy—i.e., the latent geometry supports robust, uncertainty-aware composition rather than being tuned to a single operating point. The consistent AUC gains of PDE over LDE/GDE thus provide quantitative evidence that modeling concepts as distributions in the decomposed space (instead of as single points) yields a latent structure that both (i) behaves more like an "ideal" compositional space and (ii) is amenable to qualitative interpretation in terms of concept dominance and specificity.

---

> ### Author Response · Authors · 2025-11-21
> **Response to Q1 (relation to dictionary learning).**
>
> We appreciate this connection. Dictionary learning methods aim to learn a set of basis vectors (atoms), often with sparsity or structural priors, and one could in principle try to learn "ideal words" in that way. PDE takes a different stance: we keep the pre-trained representation (e.g., CLIP) fixed and define the ideal compositional representation as the MAP estimate of a probabilistic fusion of empirical concept distributions. Instead of changing the basis, PDE asks: given the existing geometry and its empirical uncertainty, what is the most plausible composite point?
>
> This has two implications for interpretability. First, PDE improves compositional classification in both text and image spaces without additional supervision or end-to-end training, showing that uncertainty-aware fusion alone can recover missing compositional structure from frozen embeddings. Second, because fusion is controlled by covariance/precision, uncertainty becomes an explicit and interpretable knob: it determines how much each concept influences the composite along each latent direction. We clarify that dictionary learning and PDE are complementary. Dictionary learning reshapes the representational basis, while PDE specifies an uncertainty-aware fusion rule on top of a fixed basis.

---

### Author Response · Authors · 2025-12-02

**2. Resolution of Reviewer Concerns**

We summarize how our revisions specifically address the main concerns raised by the reviewers.

|Key Concern|Resolution (Rebuttal Actions)|Reviewers|
|-|-|-|
|Limited discussion of uncertainty and interpretability.|**Resolved:** Added a new WordNet-based semantic-hierarchy analysis (Appx B.8, Fig. 8) showing that PDE’s covariance-driven dominance patterns align with external semantic specificity; expanded explanation of compositional pliability and how precision controls which primitive “wins” along each direction.|K99P, G8ow, j85S|
|Gaussian assumption.|**Resolved:** Added a Shapiro–Wilk normality study in the GDE space (Appx B.7, Table 10) showing that CLIP/GDE embeddings are reasonably close to Gaussian. Also added references that assume CLIP embedding as Gaussian. |G8ow (W1, Q1)|
|Role of isotropic prior λ and its semantic impact.|**Resolved:** Clarified that λ acts as numerical regularization rather than a semantic decay term; added a λ-sweep ablation (Fig. 7) showing AUC is flat over several orders of magnitude around our default and that conclusions do not rely on tuning λ.|G8ow (Q6)|
|Meaning of count-based scale recovery and evaluation metrics.|**Resolved:** Explained count-based scale recovery via a simple carrier × scale view of LDE/GDE. Clarified closed vs. open world, Seen/Unseen/HM/AUC definitions, and fixed related table formatting.|G8ow (Q7–Q9), iJMx (Q2–Q5), K99P (W2)|
|Narrow experimental scope (only attribute–object pairs).|**Resolved:** Added a new relational composition experiment on VSR triples (obj1, rel, obj2), showing consistent PDE gains over GDE in a 3-primitive setting (Appx B.9, Table 11). Summarized how the same MAP rule can be modeled to arbitrary compositions (temporal/causal).|G8ow (W1, Q9, Q14)|
|Lack of comparison to broader CZSL literature and related probabilistic work.|**Resolved:** Added a dedicated CZSL paragraph in Related Work, discussing recent methods (prompting, graph/prototype learning, disentangling) and clarifying that PDE is a post-hoc probabilistic fusion rule complementary to these approaches. Also clarified conceptual differences from Neculai et al. (2022) |iJMx (W1), j85S (W1)|
|Computational cost and practicality.|**Resolved:** Provided a complexity discussion and runtime comparison between GDE and PDE on all datasets and modalities (Appx A.2, Table 2). We show that PDE’s additional covariance/MAP computations incur modest overhead in text settings and are actually cheaper than GDE in image settings where GDE requires validation-time hyperparameter search.|j85S (W3), iJMx (W2)|
|Clarity on partner-awareness, global geometry, and potential extensions.|**Resolved:** Clarified covariance-based partner-awareness (direction-dependent weighting via precisions), that PDE acts locally in the decomposed GDE space, and briefly outlined temporal/causal uses as future work.|G8ow (Q3, Q4, Q12–Q14)|


By combining a probabilistic extension of decomposable embeddings with new qualitative analyses, relational benchmarks, and a careful study of assumptions and efficiency, we hope PDE will serve as a useful, interpretable building block for uncertainty-aware composition in vision–language models. We thank the Area Chair and reviewers again for their constructive input, which improved the clarity and scope of the paper.

---

### Author Response · Authors · 2025-12-03

**Dear Area Chair,**

We sincerely appreciate the reviewers’ thoughtful and detailed assessments of our work. We have substantially revised the paper to clarify our contribution and to better support PDE empirically and theoretically.

We propose Probabilistic Decomposable Embeddings (PDE), a probabilistic refinement of LDE/GDE that models each primitive as a Gaussian “ideal word” and composes them by precision-weighted MAP fusion. In the revision we (i) add semantic-hierarchy evidence that covariance encodes intuitive dominance, (ii) extend experiments to relational (3-primitive) composition and stronger VLM backbones, (iii) justify our Gaussian and covariance assumptions with formal tests and shrinkage analysis, and (iv) clarify evaluation protocols, runtime, and the relation to the broader CZSL literature, which we view as complementary to our post-hoc setting.

**1. Strengths Highlighted by Reviewers**

(i) Offers a principled probabilistic extension of ideal words / decomposable embeddings:
- *"This extension to the ideal words methodologies is interesting, intuitive, and captures some insightful intuitions about how concepts should interact."* --- Reviewer K99P
- *"The paper proposes a principled probabilistic extension to existing work on embeddings decomposability."* --- Reviewer iJMx

(ii) Provides a sound and unifying theoretical formulation:
- *"The main strength is proposing the probabilistic version of ideal words and modeling it using a Gaussian distribution... the proposed approach is sound."* --- Reviewer j85S
- *"The proposed method unifies the linear, geodesic and probabilistic formulations."* --- Reviewer j85S

(iii) Demonstrates strong empirical gains across datasets, modalities, and architectures:
- *"The paper provides experiments comparing the results with GDE and LDE... across multiple datasets (UT-Zappos, MIT-States, CGQA), modalities (text/image), and architectures (CLIP, SigLIP, SigLIP2)."* --- Reviewer j85S
- *"The method shows good gains compared to the baselines."* --- Reviewer iJMx

(iv) Enhances interpretability via covariance and compositional pliability:
- *"By incorporating covariance information, PDE captures anisotropic uncertainty... This results in 'precision-weighted' composites that are partner-aware."* --- Reviewer G8ow
- *"The paper also provides a concept of compositional pliability and uncertainty, and stiffness."* --- Reviewer j85S

---

### Meta-Review · Area_Chair_1odx · 2026-01-05

**Summary:**

The paper proposes Probabilistic Decomposable Embeddings (PDE - **it might lead to confusion given the acronym**) which is a probabilistic refinement of prior ideal words approaches (LDE/GDE) for composing attribute-object concepts in VLM embedding spaces. PDE models primitives as Gaussian distributions (mean + covariance) and performs composition via precision-weighted MAP fusion, optionally stabilised with a weak isotropic prior and a simple count-based scale recovery, instead of treating each primitive “ideal word” as a point estimate. The paper argues this captures compositional pliability/stiffness (direction-dependent uncertainty) and produces partner-aware compositions that often improve compositional classification (HM/AUC) across datasets/modalities/backbones.

The reviewers were mostly negative and raised a few concerns, which the authors responded to in detail, including clarifying how their paper positions itself relative to other work. It is unfortunate that there was no discussion with the reviewers, as it appears as if things could have been clarified further.

**Reviewer Concerns:**

Some key concerns include:

a) Interpretability and what covariance offers: A major concern (esp. K99P, also G8ow/j85S) was whether PDE improves interpretability beyond adding mathematical formulations. In response, the authors explicitly articulate covariance/precision as direction-dependent dominance (“pliability/stiffness”), and add a WordNet semantic-hierarchy analysis showing that the MAP carrier’s position along the attribute-object disagreement axis correlates with semantic specificity (objects more specific leads to carrier shifts towards object). This directly strengthens the interpretability claim with qualitative/semantic evidence rather than just performance deltas.

b) Gaussian assumption and uncertainty modelling validity: G8ow raised concerns about whether Gaussian primitives are realistic in VLM spaces. The authors added an empirical normality study using Shapiro–Wilk tests in the decomposed (GDE) space and argue that deviations are moderate rather than catastrophic, supporting Gaussian approximation as practical.

c) Generality beyond attribute–object pairs / beyond CLIP: G8ow questioned overall scope; authors added a three-primitive relational composition experiment derived from VSR triples (obj1, rel, obj2) and report consistent improvements of PDE over GDE on that task. They also emphasise that the MAP fusion generalises algebraically to k-way compositions.

d) Novelty vs. related probabilistic composition work (Neculai et al. 2022): j85S questioned novelty and relation to Neculai et al. The authors argue that the regimes differ; Neculai et al. is end-to-end learned multi-query retrieval in a task-specific space, whereas PDE is a post-hoc probabilistic refinement of decomposable embeddings in the pre-trained VLM latent geometry, aimed at compositional classification and interpretability within that decomposed space.


** I would appreciate some help from other ACs or SAC, given that the topic is outside my area, and the authors provided good responses without having the chance to discuss them with the reviewers.**

**Reviewer Scores:**

The reviewers were overall fairly critical but most of the issues appear to have been addressed adequately.
Given the topic of this paper and the depth of the responses, a propor discussion with the reviewers would have been very helpful but my feeling is that if reviewers decided to engage we would have seen some score increases possibly modest, therefore positioning the paper a  borderline submission; hence my final recommendation.

---

### Decision · Program_Chairs · 2026-01-26

Reject